# The last interglacial sea-level record of Aotearoa New Zealand

Deirdre D. Ryan[1*], Alastair J.H. Clement[2], Nathan R. Jankowski[3,4], Paolo Stocchi[5]

[1]MARUM – Center for Marine Environmental Sciences, University of Bremen, Bremen, Germany
[2]School of Agriculture and Environment, Massey University, Palmerston North, New Zealand
[3] Centre for Archeological Science, School of Earth, Atmospheric and Life Sciences, University of Wollongong, Wollongong, Australia
[4]Australian Research Council (ARC) Centre of Excellence for Australian Biodiversity and Heritage, University of Wollongong, Wollongong, Australia
[5]NIOZ, Royal Netherlands Institute for Sea Research, Coastal Systems Department, and Utrecht University, PO Box 59 1790 AB Den Burg (Texel), The Netherlands

*Correspondence to*: Deirdre D. Ryan (dryan@marum.de)

**Abstract:** This paper presents the current state-of-knowledge of the Aotearoa New Zealand last interglacial (MIS 5 *sensu lato*) sea-level record compiled within the framework of the World Atlas of Last Interglacial Shorelines (WALIS) database. Seventy-seven total relative sea-level (RSL) indicators (direct, marine-, and terrestrial-limiting points), commonly in association with marine terraces, were identified from over 120 studies reviewed. Extensive coastal deformation around New Zealand has prompted research focused on active tectonics, the scale of which overprints the sea-level record in most regions. The range of last interglacial paleo-shoreline elevations are significant on both the North Island (276.8 ± 10.0 to -94.2 ± 10.6 m amsl) and South Island (165.8 ± 2.0 to -70.0 ± 10.3 m amsl) and have been used to estimate rates of vertical land movement; however, in many instances lack adequate description and age constraint for high-quality RSL indicators. Identified RSL indicators are correlated with MIS 5, MIS 5e, MIS 5c, and MIS 5a and indicate the potential for the New Zealand sea-level record to inform sea-level fluctuation and climatic change within MIS 5. The Northland Region of North Island and southeastern South Island, historically considered stable, have the potential to provide a regional sea-level curve, minimally impacted by GIA and reflecting near eustatic fluctuations, in a remote location of the South Pacific, across broad degrees of latitude; however, additional records from these regions are needed. Future work requires modern analogue information, heights above a defined sea-level datum, better stratigraphic descriptions, and use of improved geochronological methods.

The database presented in this study is available open-access at this link: http://doi.org/10.5281/zenodo.4590188 (Ryan et al., 2020a).

## 1        Introduction

The New Zealand nation (Aotearoa in the Māori language) is an archipelago comprised of two large 'main' islands, nine outlying island groups, and hundreds of additional islands. The distribution of islands extends southward from the sub-tropics to sub-polar regions in the mid-latitudes of the Southern Hemisphere. The main islands, the North Island (113,729 km$^2$) and the South Island (151,215 km$^2$), are a small area of Zealandia micro-continental crust straddling the convergent boundary of the Australian and Pacific Plates. This geographic setting produces a relatively youthful and complex landscape subject to volcanism, glaciation, and tectonism. This latter characteristic has facilitated the preservation of marine terrace sequences spanning the Pleistocene and into the Holocene that have been the target of much research and discussion of New Zealand geomorphology and geology since the late 19th Century. Importantly, the terraces also serve as the primary source of last interglacial paleo-sea level records.

Resolving a sea-level record from New Zealand marine terraces (and other sea-level indicators) has historically been complicated by the long-term and ongoing coastal deformation (Section 5.2) and difficulties in geochronology (Section 5.3).

Therefore, the impetus behind much of the marine terrace research completed in New Zealand (Pillans, 1990a and references therein; this review) has been to constrain long-term uplift rates using the many preserved sequences; an approach typically used on active margins (Creveling et al., 2015; Simms et al., 2016). Recent advances in sea-level studies have highlighted the need for increased spatio- and temporal density of relative sea-level (RSL) indicators, analyzed using standard definitions and methods, to assist in constraining paleo ice sheet limits and to improve models of glacio- and hydro- isostatic adjustment (GIA) and future predictions of ice-sheet and sea-level responses to regional warming (Dutton et al., 2015; Rovere et al., 2016; Austermann et al., 2017; Barlow et al., 2018; Capron et al., 2019; Golledge, 2020).

We use the framework provided by the World Atlas of Last Interglacial Shorelines Database (WALIS, https://warmcoasts.eu/world-atlas.html) to build a database of previously published last interglacial RSL indicators: their descriptions, geochronological constraint, and associated metadata. The intuitive interface of WALIS provides a standardized template for data entry, clarifying the collection and analytical methods used in identifying and describing previously published and new paleo sea-level proxies and the source of any associated uncertainties for the scientific community. This includes fields for type and description of sea-level indicator, elevation measurement method and uncertainty, geographic positioning method and uncertainty, sea-level datum, and geochronological constraint and uncertainty; e.g. amino acid racemization and luminescence. The database is available open-access at this link: http://doi.org/10.5281/zenodo.4590188 (Ryan et al., 2020a). Database field descriptors are available at this link: https://doi.org.10.5281/zenodo.3961544 (Rovere et al., 2020).

In order to be considered for entry into WALIS as an RSL indicator, a data point must have three characteristics (Rovere et al., 2016): 1) elevation referred to a defined sea-level datum and position (latitude and longitude) referred to a known geographic system; 2) its offset (relative or absolute) from a former sea-level needs to be known; and 3) it must have an established age (relative or absolute). For our review, preference was given to peer-reviewed publications; although student theses heavily referenced in later work were included (e.g. Goldie 1975; Hicks, 1975) if made available. Geological maps, while useful in indicating the extent of marine deposits and general location of a paleo-shoreline, seldom provide precise descriptions, geographical locations, or elevations above sea level for a singular data point; e.g. Nathan, 1975; Begg and Johnston, 2000. However, a publication not having sufficient information for the identification of an RSL indicator, did not preclude its usefulness or inclusion in the discussion. Many publications contain research worthy of discussion in Section 4. Other publications, some geological maps and government reports were behind paywalls or were not found in the course of the literature review.

In total, we reviewed over 120 publications and identified 77 unique RSL indicators of varying quality. Previous reviews of New Zealand sea-level indicators (Gage, 1953; Pillans, 1990a) have recommended, "more accurate height data, careful attention to shoreline datums, and greater use of dating techniques yielding numerical ages" (Pillans, 1990a, p. 227). This compilation summarizes the current knowledge of last interglacial (LIG) sea level in New Zealand by identifying RSL indicators within the literature, provides an overview of the standardized treatment of LIG RSL indicators using WALIS, identifies problems, and makes recommendations for the improvement of the NZ LIG record. Below we first provide an overview of the historical development of sea-level studies within New Zealand to provide context for the outcomes of this compilation. The following sections describe the types of RSL indicators identified in this work (Section 2) and how geographic location, elevation, and associated uncertainty were assigned to each indicator (Section 3). The bulk of this publication (Section 4) discusses the current state of the last interglacial sea-level record within New Zealand and provides detailed descriptions of the RSL indicators identified, dividing the discussion into sub-regions within the North and South Islands. In the last sections of the paper, we address the main sources of uncertainty (Section 5), and provide further details on the Pleistocene and Holocene sea-level record (Section 6), and suggest future research directions (Section 7).

## 1.1 Overview of literature with reference to last interglacial sea level

The legacy of New Zealand sea-level change studies in many ways reflects the evolution of eustatic sea-level and plate tectonic science. By the latter half of the 19th Century, "raised beaches" were well-recognized within the New Zealand landscape and investigations were subject to review and synthesis as early as 1885 (Hutton). Hutton (1885) attributed the differing altitude of the marine terraces to slow retreat of a maximum sea level exceeding 800 ft [~244 m], whereas Henderson (1924) argued for uniform changes in RSL around New Zealand as the result of either retreat of the sea or uplift of the entire archipelago. Subsequent studies revealed regional differences in marine terrace elevations attributed to differential vertical land movement (Jobberns, 1928; King, 1930; 1932). Although the cause for larger, regional variation was not fully understood, correlation with overseas marine terraces to determine age was considered "extremely hazardous" (Gage, 1953, p. 27). However, awareness of regional warping did not dissuade many authors from attempts to determine terrace age by 'matching' the New Zealand terraces with the well-recognized, yet far-field, marine terrace sequences in the Mediterranean (the Monastirian, Tyrrhenian, Milazzian, Sicilian sequence) (e.g. Brothers, 1954; Cotton, 1957; Leamy, 1958; Suggate, 1965) and on the Huon Peninsula, Papua New Guinea (Chappell, 1975; Bishop, 1985; Bull and Cooper, 1986). The former correlation also promoted the widespread use of Mediterranean sea-level nomenclature as a framework for New Zealand marine terraces for decades. These correlations were often attempted by 'counting backwards', with increasing altitude or distance from the modern shoreline. Prior to the development and common use of other geochronological methodologies (e.g. luminescence dating, amino acid racemization), this approach or the similarities between terrace cover beds were commonly applied to correlate distant terrace sequences within New Zealand (Kear and Waterhouse, 1961; Fleming and Suggate, 1964; Chappell, 1970; 1975; Nathan and Moar, 1975; Heine, 1979; 1982; Ota et al., 1984; 1996; Bishop, 1985; Richardson, 1985). Following the publication of marine oxygen-isotope records, age determination moved beyond the constraints of the Mediterranean and Huon Peninsula sequences and New Zealand marine terraces were correlated with specific sea-level intervals reflecting global sea-level and ice-volume (e.g. Beu and Edwards, 1984; Ward, 1988a; Pillans, 1980a; 1994; Suggate, 1992; Berryman, 1993; Ota et al., 1996). These correlations not only provided greater certainty in the interpretation of New Zealand geomorphology, but also lent strength to the newly developed marine oxygen-isotope sea-level proxy records.

With the recognition of regional tectonics playing a large role in the altitude and distribution of the marine terraces, researchers shifted focus, using marine sequences primarily to assess regional long-term vertical land movements (but also to address other research questions such as paleoclimate) and away from attempts to resolve sea-level fluctuations or eustatic sea-level history (e.g. Ghani, 1978; Hesp and Shepherd, 1978; Pillans, 1983; 1986; Bull and Copper, 1986; Ward, 1988a; Suggate, 1992; Berryman, 1993; Ota et al., 1996; Rees-Jones et al., 2000; Begg et al., 2004; Kim and Sutherland, 2004; Litchfield and Lian, 2004; Alloway et al., 2005; Cooper and Kostro, 2006; Wilson et al., 2007; Claessens et al., 2009; Clark et al., 2010; Oakley et al., 2018). This heralded the practice of correlating a marine terrace with a generic age (e.g. 120,000 ka) and height of the highstand (e.g. 5 m), usually derived from the marine oxygen-isotope records, with little effort to provide precise locations, elevations, or stratigraphic descriptions of sea-level indicators. While the development of geochronological methods assisted in constraining the age of marine terraces to the appropriate marine oxygen-isotope stage, common practice to estimate sea level continued to be the use of a eustatic sea level-height determined from various marine oxygen-isotope records. Due to the difficulty in relating spatially distant marine terraces, in part due to difficulties in determining a numerical age, nomenclature for the terraces and their associated sediments was historically regionally specific and has evolved through time with better age constraints (Table 1; Section 4).

In what may be considered a benefit from the prevalence of uplift along the New Zealand coast, numerous terrace sequences have been identified and correlated with the sea-level peak of MIS 5e, and subsequent interstadials MIS 5c and MIS 5a, or a combination of two of the three. Such sequences can be, and have been, used within New Zealand to study sea-level and climate fluctuation within MIS 5 *sensu lato* (e.g. Brothers, 1954; McGlone et al., 1984; Bussell, 1990; 1992; Berryman, 1992;

Shulmeister et al., 1999); however, the majority of these studies were palynological with a focus on resolving climatic fluctuations and not sea level. Two regions of New Zealand have historically been considered tectonically stable (Gage, 1953; Pillans, 1990a; Beavan and Litchfield, 2012): the Northland Region of the North Island and portions of the southern Otago and eastern Southland regions of the South Island. Although these regions provide the best opportunity to identify a New Zealand sea-level record that can contribute to the discussion of MIS 5e sea level in a global context, only a few RSL indicators have been identified in these areas (Section 4).

This review of the LIG sea-level indicators in New Zealand, like the preceding reviews by Gage (1953) and Pillans (1990a), found considerable lack of necessary detail in the published literature for the identification of robust sea-level indicators defined by the WALIS framework; including that published post-1990. Many locations are referred to broadly, with the only indication of place a large-scale location map from which a field site cannot be precisely determined. Marine terrace descriptions are often limited to a range of altitude heights in meters without a defined sea level datum - commonly the reader is left to assume above mean sea level. In many instances where a stratigraphic description is provided, it is logged as meters depth of burial and the height above sea level cannot be determined. The use of altitude to determine age, whether above an undefined sea level datum or between successive terraces, is prevalent. On the North Island, tephrochronology is commonly used both as a regional stratigraphic marker and to assist in constraining marine terrace age (e.g. Pullar and Grant-Mackie, 1972; Chappell, 1975; Pain, 1976; Iso et al., 1982; Ota et al., 1989; Berryman, 1993; Wilson et al., 2007; Claessens et al., 2009). For the purposes of determining last interglacial age the tephras most commonly referred to are the Rotoehu Tephra at ~47 ka (Danišík et al., 2012; Flude and Storey, 2016), providing a minimum age, and the Hamilton Ash at ~340 ka (Pillans et al., 1996; Lowe et al., 2001) providing a maximum age (Section 5.3.2). The most commonly applied geochronological methods for determining numerical age are calibrated amino acid racemization and various techniques of luminescence dating, both of which have complicated histories in New Zealand (Sections 5.3.3 and 5.3.4). In summary, in New Zealand there are few well-described and constrained RSL indicators in proportion to the sizable quantity of literature published concerning the physical record of last interglacial (*sensu lato*) sea level.

It must be stressed that the challenge in precisely reconstructing last interglacial sea level from this body of literature stems from the early recognition by the New Zealand Quaternary community that the sea-level record was strongly overprinted by the active landscape; hence much of the earlier work has focused on understanding the tectonics and vertical land movements of the archipelago. Furthermore, until late into the 20[th] Century studies had to contend with difficulties in correlating distant sequences due to limited geochronological methods and still must grapple with the tectonically active nature of the islands and difficult topography.

## 2    Types of sea-level indicators

The majority of LIG RSL indicators identified within New Zealand are associated with a marine terrace. The marine terrace terminology used here follows that established by Pillans (1990a; b) (Figure 1). Briefly described, a marine terrace is a gently sloping, sub-planar landform formed either through the deposition of sediments in shallow-water and subaerial coastal environments (sometimes referred to as a marine-built terrace), or by marine erosion processes that produce a basal, sub-horizontal shore platform, subsequently overlain by cover beds of marine and/or terrestrial sediments. Terrace risers, identified by their steeper slopes, are present at the inland and seaward margins of the terrace. The inland terrace riser is the surface expression of a fossil sea cliff, formed at the peak, or sustained height, of the marine transgression. This is also the location of the shoreline angle (or strandline), a wave erosion feature at the intersection of the basal shore platform and the base of the fossil sea cliff, which serves as the best indicator of the height and timing of peak sea-level; although it is often covered with colluvium and seldom exposed. The height of the base of the inland terrace riser or the terrace surface is often used as an

alternative; however, because the thickness of cover beds is often unknown these elevations are not necessarily a good indicator of sea level (Pillans, 1983). The seaward terrace riser identifies the sea cliff of a subsequent sea-level highstand.

Marine terraces have a similar morphology to rocky shore platforms, resulting in shared terminology. However, shore platforms are distinguished from marine terraces by their exposed bedrock surface and relative lack of overlying sediment reflecting active erosional and weathering processes (Griggs and Trenhaile, 1994). The relative roles of marine and subaerial processes in shaping rocky shore platforms remains equivocal (e.g. Stephenson, 2000; Stephenson and Kirk, 2000; Trenhaile, 2008), and shore platforms do not form at uniform elevations because of sea-level alone (e.g. Kennedy and Dickson, 2007; Kennedy et al., 2011; Stephenson and Naylor, 2011). A number of publications from the West Coast and Southland Regions (e.g. Wellman and Wilson, 1964; Bishop, 1985; Bull and Cooper, 1986; Section 4.2.1) allude to landforms which may possibly be features of rocky shore platforms; however, sufficient morphological description to identify a RSL indicator from a rocky shore platform was only provided by Kim and Sutherland (2004; Section 4.2.1). To avoid confusion within Section 4 between rocky shore platforms and the basal shore platforms underlying marine terraces, the latter is referred to as basal platform within Section 4 where it is apparent that the platform is overlain by marine sediments.

In this review, depending on whether the elevation for the marine terrace RSL indicator came from the basal platform, marine cover beds or terrestrial cover beds, the RSL indicator is identified as either a marine-limiting point, a direct indicator, or terrestrial-limiting point, respectively. Limiting data points are derived from landforms that cannot be directly related to sea level and therefore, it is not possible to quantify with useful accuracy its formation relative to mean sea level (Rovere et al., 2016). These data points only inform whether sea level was above (marine-limiting) or below (terrestrial-limiting) their elevation at the time of their formation. For example, marine terrace sediments for which the depositional environment hasn't been constrained and/or are located an undefined distance from the inland terrace riser (paleoshoreline), are considered marine-limiting. Marine terrace sediments with evidence of wave action can be constrained to a tidal range and possibly depositional environment (e.g. beach deposit) and related directly to sea level.

New Zealand RSL indicators have also been identified at depths below modern sea level. These indicators, from within sediment and well cores, are described as sediment packages that vary by location. The depositional environment with which those sediment facies were correlated dictate how the datapoint is identified in this review; i.e. a marine- or terrestrial-limiting point or a direct sea-level indicator. For example, the Bromley Formation (Brown et al., 1988) is considered terrestrial limiting due to its depositional environment described as beach, lagoonal, dune, and coastal swamp sediments associated with rising and high sea level.

Determining where a paleo-RSL indicator formed with respect to elevation above or below a tide level is the most fundamental attribute necessary for paleo-RSL reconstructions. This relationship is established by the indicative meaning of a modern analog (Shennan, 1982; van de Plassche, 1986; Rovere et al., 2016), which basically assumes that the relationship of the equivalent modern landform to a tidal level is representative of the same relationship for the paleo-landform. The indicative meaning consists of two parts: the indicative range (IR) defines the elevation range over which an indicator may form, i.e. the upper and lower limits; and the midpoint of that range is the reference water level (RWL), which is defined by its elevation above or below a geodetic datum or tidal level (Table 2). The observed RWL of the modern analog is used to calculate the height of paleo sea level with respect to the paleo-RSL indicator by simply subtracting the RWL from the elevation of the paleo-RSL indicator (Figure 1). Uncertainty in this measurement is derived from the precision of the measurement method used and the IR – the greater the IR, the greater the uncertainty. Modern analogs should be recorded locally to reflect differing coastal geomorphology and oceanographic patterns.

Modern analog information was provided in only one study from the Northland Region (Nichol, 2002). For all other indicators, the indicative meaning of the modern analog was quantified using the IMCalc tool (Lorscheid and Rovere, 2019). IMCalc

determines indicative meaning for any point along the global coastline using hydro- and morphodynamic equations with inputs from global wave and tide datasets to determine the indicative range of coastal landform features from which a reference water level is derived (the midpoint). The only required inputs to IMCalc to determine indicative meaning for a modern datapoint are the latitude and longitude and type of coastal landform feature; e.g. beach deposit.

Due to the prevalence of coastal deformation around New Zealand, the measured elevation of the RSL indicators is provided in Section 4 rather than the paleo sea-level interpretation, which in many instances would be off by tens of meters from what can realistically be expected. However; the indicative meaning and inferred paleo sea level is available in the database (http://doi.org/10.5281/zenodo.4590188 Ryan et al., 2020a).

**Figure 1: Illustration of marine terrace terminology (adapted from Pillans, 1990b). The indicative range of a marine terrace landform is marked by the storm wave swash height (upper limit) and breaking depth (lower limit; Rovere et al., 2016). (See Table 2 for indicative range of other RSL indicators.) Marine terrace cover beds can be terrestrial-limiting (terrestrial sediments) or either marine-limiting or a direct indicator of sea level depending upon the availability of stratigraphic or sedimentologic information with which to relate to sea level. In this analysis, basal shore platforms underlying marine sediments were considered marine-limiting when elevations were taken from the outer terrace riser or an undefined distance from the inner margin and relationship with paleo-sea level could not be constrained. The inset provides a worked example of the paleo sea-level calculation for the MIS 5 fossil sea cliff (30 ± 5 m amsl) of the Tokomaru Terrace near Otaki (Palmer et al., 1988; Section 4.1.6).**

## 3    Location and elevation measurements

The latitude and longitude for over half of the RSL indicators was determined using Google Earth to match locations from a publication map. The location for each well core described by Brown et al. (1988) was acquired from the Canterbury Regional Council well database (www.ecan.govt.nz/data/well-search; Environment Canterbury, 2020), which also provides accuracy of well elevation measurement above mean sea level. This value was incorporated into the elevation uncertainty provided by Brown et al. (1988). The other primary method used in publications for assigning location information has been New Zealand Map Grid coordinates. In such instances, New Zealand map grid coordinates were converted to WGS84 (G1762) via the Land Information New Zealand online tool for coordinate conversions (https://www.geodesy.linz.govt.nz/concord/index.cgi; LINZ, 2020a). Oakley et al. (2017) determined locations with handheld GPS. Although Wilson et al. (2007) measured elevation using real-time kinetic GPS, locations were provided as map grid coordinates.

Three publications (McGlone et al., 1984; Rees-Jones et al., 2000; Kennedy et al., 2007) referred elevation data to a tidal datum other than mean sea level (msl). All other publications either referred to present or mean sea level or did not define the sea-level datum and only referred to sea level, in which case mean sea level has been assumed. The mean sea level definition has not been differentiated in the below identification (Section 4) of RSL indicators, but is shown in the database (http://doi.org/10.5281/zenodo.4590188 Ryan et al., 2020a). The most commonly used method of elevation measurement has been altimetry, followed closely by metered tape or rod. In almost all cases of altimetry, the accuracy of measurement is considered to be within <5 m. The accuracy of the metered method, most commonly applied to core sediments, ranges from within <0.1 m to <5 m (Brown et al., 1988; Shulmeister et al., 1999). Less common means of determining elevation include differential GPS (Wilson et al., 2007; Oakley et al., 2017), total station (Kennedy et al., 2007), and a combination of topographic maps and digital elevation models (Wilson et al., 2007). In one instance, depression of the sea horizon in relation to the location of measurement was used (Ghani, 1978).

Similar to the sea level datum, in many instances the elevation measurement method is not stated. The intended approach for assigning elevation uncertainty with entry of an RSL indicator into WALIS was to determine elevation uncertainty by calculating the square root of the sum of the sea level datum error and adding a percentage of uncertainty of the elevation measurement depending upon the precision of the method used. If the sea level datum error was not provided, then the elevation

uncertainty would be determined as 20% of the elevation. However, these approaches were not possible due to the lack of stated sea level datum and elevation measurement method in many studies. Furthermore, given the variable rates of uplift along the New Zealand coast, applying 20% of the elevation would result in uncertainty in the range of tens of meters in some instances. Instead, if an uncertainty for a described elevation method was provided in the original publication, this was accepted and applied with entry of the RSL indicator into WALIS. All other instances were assessed on a case-by-case basis and uncertainty was defined dependent upon the quality of the indicator description provided within the original publication. None of the elevations provided here have been adjusted for the effects of uplift, GIA, or dynamic topography.

## 4    Relative sea-level indicators

The approach here for describing RSL indicators will continue the practice of dividing New Zealand between the North and South Islands, with further subdivision based roughly upon the government regions and the dominant tectonic regimes (Figure 2). This approach was chosen for clarity in discussion and in recognition that coastal deformation in New Zealand is driven by the tectonic regimes resulting from the position of the archipelago over the active boundary between the Australian and Pacific Plates. The obliquely convergent plate boundary can be subdivided into three distinct components from north to south: (1) the obliquely westward-subduction of the Pacific Plate at the Hikurangi Subduction Zone, (2) a transitional zone from subduction to dextral transpression and oblique continental collision along the Marlborough Fault System and the Alpine Fault, and (3) the northeastward-subduction of the Australian Plate at the Puysegur Subduction Zone extending south of the South Island (see Nicol et al., 2017 for a detailed review of Quaternary tectonics of New Zealand, and Litchfield et al., 2014 for a detailed review of active faulting in New Zealand).

The WALIS database identifiers for RSL indicators (e.g. WID 43) will be used throughout this section to assist in correlation between this publication and the database (http://doi.org/10.5281/zenodo.4590188 Ryan et al., 2020a), within which they are found under the heading "WALIS RSL_ID". Unique identifiers for amino acid racemization and luminescence samples are as provided in original publication, but can also be correlated with a unique WALIS identifier within the database. Figures 3 and 4 provide spatial context for RSL indicators and illustrate measured elevation with relation to sea level datum and age attribution.

**Figure 2**: **The New Zealand North and South Islands illustrated with government regions and dominant tectonic regimes (discerned by color-shading) drawn after Ballance (2017) and Villamor et al. (2017). Also shown are the primary faults of the North Island Dextral Fault Belt, the Marlborough Fault System, and the Alpine Fault (Langridge et al., 2016). Diamonds indicate locations of RSL indicators within the WALIS database and described here. Hill shade data sourced from the LINZ Data Service and licensed for reuse under CC BY 4.0 (LINZ, 2020b).**

### 4.1    North Island

The North Island features distinctive tectonic and geomorphic domains manifested by the continental collision processes associated with the subduction of the Pacific Plate along the Hikurangi Subduction Zone (Figure 2). The entire eastern North Island constitutes the Hikurangi Margin that is being compressed by the collision of the Pacific and Australian Plates (Nicol et al., 2017). The margin structure from east to west is composed of an accretionary wedge (outer forearc, mostly located offshore), a forearc basin (inner forearc), and the Axial Ranges (frontal ridge) bisected by the North Island Dextral Fault Belt (NIDFB) (Berryman, 1988). The majority of the eastern North Island is being uplifted as the forearc is compressed with the exception of parts of Hawke's Bay: the Heretaunga Plains, a subsiding tectonic depression on the margin of the convergent plate boundary (Lee et al., 2011), and the coast near Wairoa, which experienced late Quaternary co-seismic subsidence (Ota et al., 1989a; Litchfield, 2008). The oblique subduction of the Pacific Plate is rotating the eastern North Island clockwise producing a backarc rift system and volcanic arc: the Taupo Volcanic Zone (TVZ). The Whakatane Graben, at the northern

end of the TVZ, is a subsiding tectonic depression bounded by normal faults and infilled with up to 2 km of late Quaternary sediments (Wright, 1990; Beanland and Berryman, 1992). South of the TVZ lies the Whanganui Basin, a proto-back-arc sedimentary basin infilled with 4-5 km of dominantly marine Plio-Pleistocene sediment (Anderton, 1981). The western and northern North Island is a zone of backarc extension that normal faults have segmented into reticular blocks of uplifted basement rock and sedimentary basins. The backarc region also features extinct volcanic arcs west of the TVZ left relict by the clockwise rotation of the Hikurangi subduction zone (Wallace et al., 2004).

The North Island is subdivided into nine government regions (Figure 2). Much of the north and west coasts are found within the Northland, Auckland, and Waikato regions. The northeast and east coasts are found within the Bay of Plenty, Gisborne, and Hawke's Bay regions with the southernmost coast of the North Island located within the Wellington region. The government region of Whanganui-Manawatu extends to both the southern and eastern coastline of the North Island, with the eastern coastline colloquially referred to as the Wairarapa within New Zealand. This practice is applied here to avoid confusion over reference to geographic location within the review.

**Figure 3: Position, elevation, and age correlation of the North Island RSL indicators within the WALIS database and described here. Elevations are provided above mean sea level (Section 3) unless indicated otherwise; e.g. WID 36 and 37. Each RSL indicator is identified with their unique WALIS database identifiers (e.g. WID 34) and is correlated in text with their original publication. Symbols of indicators that would stack, extend offshore for clarity. Marine- and terrestrial- limiting points are indicated with up- and down- arrows, respectively. Locations and landscape features mentioned in text are also shown with built-up areas (e.g. towns and cities) indicated by stars. Hill shade data sourced from the LINZ Data Service and licensed for reuse under CC BY 4.0 (LINZ, 2020b).**

### 4.1.1    Northland and Auckland regions

Portions of the Northland and Auckland regions have been considered relatively tectonically stable through the late Quaternary (Gage, 1953; Gibb, 1986; Pillans, 1990a; Beavan and Litchfield, 2012). Arguably the best described and constrained RSL indicator in New Zealand is the regressive nearshore, beach, and foredune sequence exposed at One Tree Point on the southern shoreline of Whangarei Harbour (Figure 3A WID 34; Nichol, 2002). Nichol (2002) traced the contact between the beach and foredune facies (delineated by a heavy mineral sand interpreted as the hightide swash deposit) for a distance of 3.4 km. The contact elevation decreases seawards from a maximum height of +6 m to +3 m above mean sea level (amsl), reflecting relative sea-level fall. Ground penetrating radar across the sequence revealed the swash lamination of the beach face and supports the interpretation of a prograding barrier sequence. Thermoluminescence (TL) samples of overlying foredune sand indicate deposition through MIS 5 (substages 5e to 5a). Given the height distribution of the swash deposit, and under the assumption that no tectonic uplift has occurred, Nichol (2002) argued for deposition during late MIS 5e. The height of the swash deposit associated with the TL sample providing MIS 5e age (Sample OTP1, 115 ± 19 ka) is 4.6 ± 0.92 m amsl.

The One Tree Point RSL indicator is the only one identified in the Northland and Auckland regions. Other remnants of MIS 5 sea level have been recorded, but lack sufficient detail to be used as RSL indicators. A series of Master's theses investigated the Quaternary geology and geomorphology of the Aupouri and Karikari Peninsulas in Northland, where possible MIS 5 marine terraces and estuarine sediments were identified (Goldie, 1975; Ricketts, 1975; Hicks, 1975). However, the correlations were considered tentative in appreciation for a lack of dating methods at the time and the dangers of distant terrace correlation or altimetry to determine age.

The west coast of Northland and Auckland has been a location of extensive sand deposition throughout most of the Quaternary and large volumes of these sands are subject to movement by nearshore currents and longshore drift with considerable erosion and deposition documented in historical record (Ballance and Williams, 1992; Blue and Kench, 2017). It is possible that remnants of the MIS 5 highstand may not exist outside of the more sheltered harbours. On the South Kaipara Peninsula, within

the Kaipara Harbour and protected from the high energy coastline, Brothers (1954) identified and described the estuarine Shelly Beach and Waioneke Formations, the two youngest members of the Kaihu Group. The Shelly Beach Formation occurs as a marine terrace, with the surface recorded at 33 to 40 m amsl on the northern end and eastern side of the South Kaipara Peninsula. The Waioneke Formation is comprised mainly of pumiceous silts and localized sandy facies that infill valleys cut into the Shelly Beach and older formations. The terraces surfaces formed upon Waioneke Formation are typically found at 14 m to 23 m and 4.5 m to 7.5 m amsl. Brothers (1954) initially correlated the Shelly Beach Formation to the mid-Pleistocene and the younger Waioneke Formation to the late Pleistocene. Later assignment of the Shelly Beach Formation to MIS 5 was made by Chappell (1975; Table 1). Richardson (1985) identified the Shelly Beach Formation on the southern shoreline of the Pouto Peninsula (on the northern side of the harbour), often occurring with a 40 m terrace. Chappell (1975) and Richardson (1985) determined the MIS 5 age of the Shelly Beach Formation by correlating it with the Ngarino Terrace in the Whanganui Basin to the south (Section 4.1.2). However, the Ngarino Terrace was later correlated with MIS 7 (Pillans, 1983; 1990b). Further south, Claessens et al. (2009) claim to have identified a MIS 5 terrace at 41.5 m amsl within the Waitakere Ranges, on the coastline north of the Manukau Harbour entrance; however, the stratigraphic description and geochronological constraint are poor and the publication was rejected for inclusion into the database.

### 4.1.2    Waikato, Taranaki and the Whanganui Basin

Chappell (1970) redefined the Kaihu Group, extending its framework to encompass the Waikato region, and correlated the Waiau A and Waiau B Formations in the Waikato with the northern Shelly Beach and Waioneke Formations, respectively, on the basis of elevation and position within the Kaihu Group. The sediments of both Waiau A and Waiau B are described as forming terraces overlying irregular erosion surfaces within coastal harbours and valleys. Waiau A is composed of transgressive marine sediments forming a terrace with a maximum surface elevation of ~40 m amsl; however, a detailed stratigraphic description of the Waiau A beds at a defined location is not provided and an RSL indicator could not be identified. The Waiau A beds are made distinct from the overlying Waiau B aeolian and isolated littoral beds by an erosion surface. The surface of the Waiau B littoral beds is not found above 13 m amsl elevation; however, associated aeolian dunes exceed 106 m amsl. At the Waiau B representative section, aeolian beds are drawn to directly overlie the soil-mantled erosion surface of Waiau A at ~16.7 m amsl (Figure 7 of Chappell, 1970), providing a terrestrial-limiting point (Figure 3A WID 35). Remnants of these terraces, identified as North Taranaki 3 and North Taranaki 2 (Waiau A and Waiau B, respectively) in the northern Taranaki region, are used to correlate the west coast terraces to the marine terraces of the Whanganui Basin further south (Chappell, 1975; Table 1).

The south Taranaki coast, located on the western margin of the Whanganui Basin, is experiencing ongoing, gentle uplift due to crustal flexure as the basin depocenter migrates south (Anderton, 1981). The movement has allowed preservation of a globally significant shallow marine sequence, spanning the entire Quaternary, that has been mapped and described by multiple authors (Fleming, 1953; Dickson et al., 1974; Chappell, 1975), but most extensively by Pillans (1983; 1990b; 2017). Marine terraces, formed over the past ~700 ka, provide a robust framework of Late Quaternary sea-level fluctuations. The marine terraces consist of a basal platform overlain by up to 15 m of marine sediments, grading upwards into non-marine sediments. The total thickness of terrace cover beds generally increases with terrace age and in the westward direction towards the Taranaki volcanoes due to greater tephra and lahar cover bed thickness; however, there can be considerable local variation due to sand dunes. Fleming (1953) provided early, detailed descriptions of the Brunswick and Rapanui Terraces, recognizing both as having formed over multiple cycles of sea-level transgression and regression (Table 1). The Rapanui Terrace was later divided into the older Ngarino Terrace and younger Rapanui Terrace (Dickson, 1974). The MIS 5 terraces have since been designated as the Rapanui Terrace (120 ka, MIS 5e), the Inaha Terrace (100 ka, MIS 5c), and the Hauriri Terrace (80 ka, MIS 5a) with the Ngarino Terrace assigned to MIS 7 and the Brunswick Terrace assigned to MIS 9 (Pillans, 1983). The uplift, which allowed preservation of the terraces, has also resulted in shore-parallel deformation of the terraces. The inferred

strandline elevations (Rapanui Terrace <30-70 m, Inaha Terrace 24-40 m, Hauriri Terrace 16 m) must be viewed with additional caution due to the unknown thickness of the cover beds.

Age constraint for the Whanganui Basin sequence was provided by Pillans (1983; 1990b) using a combination of tephrochronology, specifically the Rangitawa Tephra, and amino acid racemization of wood samples (see Section 5.3.3 for discussion of AAR). RSL indicator elevations for each of the MIS 5 terraces were derived from type sections exposed on the modern coast. Most type sections are kilometers seaward of the inland terrace riser and indicate only minimum sea-level estimates for their respective highstands. The basal platform of the Rapunui Terrace (MIS 5e) is found within a cliff section at Castlecliff at 29 ± 0.3 m amsl and is directly overlain by ~5.5 m of marine cover beds (Bussell et al., 1992; Figure 3B WID 38). An additional 7.5 m of terrestrial cover bed includes dune sand correlated to MIS 5c. It is unclear which depositional environment the marine sediments should be correlated with; therefore, the basal platform is considered as a marine-limiting point. The Rapanui Terrace is again exposed 7.5 km west of Hawera, with the basal platform measured to 12 ± 0.3 m above the high-water mark (HWM; McGlone et al., 1984; Figure 3B WID 37). The type section for the Inaha Terrace (MIS 5c) is found 10 km west of Hawera, where the basal platform, at ~2 m above the HWM is overlain by near-shore marine and beach sands at 2.3 m to 4.17 m above HWM (McGlone et al., 1984; Figure 3B WID 36). Given the proximity of this site to the inland terrace riser (within 2 km), the beach sands are considered a direct RSL indicator. The basal platform of the Inaha Terrace can be traced eastwards within the modern coastal cliff to a height of ~8 m above HWM. The basal platform of the Hauriri Terrace (MIS 5a) is exposed at 2 ± 2 m amsl near Waverley and overlain by ~2 m of fossiliferous marine sand with a basal conglomerate (Pillans, 1990b; Figure 3B WID 245). Again, it is unclear which depositional environment the marine sediments should be correlated with and the basal platform is considered a marine-limiting point.

Chappell (1970; 1975) correlated the marine terrace sequences from the Whanganui Basin to northern Taranaki, Waikato, and Auckland Regions prior to any numerical age constraint and largely depended upon terrace elevation and tephrochronology. Correlation of the Waiau A and Waiau B Formations to the Whanganui sequence was reinforced through use of the Miocene-age Kaawa Shellbed as a chronostratigraphic marker and similarities in climatic and sea-level change signals perceived within the Whanganui sequence and the Kaihu Group (Chappell, 1970). In summary, Chappell (1975) correlated, from north to south: the Shelly Beach Formation, Waiau A, North Taranaki (NT) 3, and the Ngarino Terrace to MIS 5e and implied the Waioneke Formation, Waiau B, North Taranaki (NT) 2, and the Rapanui Terrace as correlatives formed during a later sea-level highstand (Table 1). To determine age Chappell (1975) used the current heights of the Ngarino and Rapanui Terrace surfaces and estimates of uplift rates to correlate the terraces with the Huon Peninsula (Papua New Guinea) sea-level curve and assign an age of ~120 000 years for MIS 5e (Chappell, 1974; Bloom et al., 1974). Pillans (1983; 1990b) later correlated the Ngarino Terrace to MIS 7 and the Rapanui Terrace to MIS 5e. The accuracy of Chappell's terrace correlation across most of the west coast of the North Island otherwise remains untested and has been used for geochronological constraint in subsequent publications, e.g. Alloway et al. (2005).

### 4.1.3 Bay of Plenty and Gisborne regions

The central portion of the Bay of Plenty is defined by a subsiding backarc rift occupied by the Taupo Volcanic Zone (Figure 2). The coastline marginal to the rift is experiencing moderated uplift. To the east, the Raukumara Peninsula, a northwards projection of the Axial Ranges, is subject to steady aseismic uplift, with some intermittent coseismic uplift events (Litchfield et al., 2007; Clark et al., 2010).

Chappell (1975) attempted to correlate the marine terraces of the west coast of the North Island with new and previously described terrace remnants across the Bay of Plenty and Raukumara Peninsula, identified as Bay of Plenty 2 (BOP2) and Bay of Plenty 3 (BOP3; Table 1; Kear and Waterhouse, 1961; Selby et al., 1971; Chapman-Smith and Grant-Mackie, 1971; Pullar and Grant-Mackie, 1972). Briggs et al. (1996; 2006) showed that the Pleistocene terraces identified in the western Bay of

Plenty, with which Chappell (1975) attempted correlation, are non-marine in origin. The marine terraces on the Raukumara Peninsula were first identified as the Otamaroa and Te Papa Members of the Rukuhanga Formation, described at a location near Cape Runaway (Chapman-Smith and Grant-Mackie, 1971). The terrace cover beds are comprised of marine sandstones and conglomerates overlain by tephras. After subtracting 3 m for tephra cover bed thickness, the inner margins of the terraces were measured to 60 m amsl (Otamaroa Terrace) and 30 m amsl (Te Papa Terrace) (Pullar and Grant-Mackie, 1972). The tephra overlying the Otamaroa Terrace was tentatively correlated to the Hamilton Tephra and the tephra overlying the Te Papa Terrace was identified as Rotoehu Tephra. Both tephras were used to argue for the formation of the terraces during interstadials equivalent to the European Brörup (MIS 5c) and later Gottweig interstadials, respectively. Chappell (1975) incorporated the Te Papa Terrace into his Bay of Plenty 2 Terrace, which was correlated with west coast MIS 5 terraces in the Auckland and Waikato regions. The Otamaroa Terrace was incorporated in the Bay of Plenty 3 Terrace and correlated to MIS 7. Per Wilson et al. (2007), the Otamaroa and Te Papa Terraces were later correlated by Yoshikawa et al. (1980) to MIS 5c and MIS 5a, respectively; however, the publication is in Japanese and we were not able to confirm the correlation.

Wilson et al. (2007) extensively mapped the Te Papa and Otamaroa Terraces from Whitianga Bay, eastward around the Raukumara Peninsula to Te Araroa, providing section descriptions and elevations measured to heights above local mean sea level. The position of the described sections within the terrace, in relation to the inner and outer margins, is not provided. The cover beds overlying the basal platform are consistently comprised of marine sediments of sand and/or gravel, never exceeding 6 m thickness, overlain by terrestrial sediments, typically less than 4 m thick, with the Rotoehu Tephra serving as a chronostratigraphic marker. Although the marine sediments are described as "beach" sediments, the locations are an unknown distance from the inner margin and contemporaneous strandline. However, the sediments do exhibit evidence of wave sorting during deposition and the locations are considered direct RSL indicators from a marine terrace landform: the Te Papa Terrace at Waihu Bay (23.9 ± 1.7 m amsl; Figure 3C WID 46);and from the Otamaroa Terrace: Omaio (13.0 ± 1.6 m; Figure 3C WID 43), Waihau Bay (62.4 ± 0.7 m; Figure 3C WID 45), Hicks Bay (131.2 ± 1.8 m; Figure 3C WID 42), and Te Araroa (273.3 ± 10.4 m; Figure 3C WID 44). Wilson et al. (2007) also collected samples for infrared stimulated luminescence (IRSL) dating from terrace sand and loess cover beds to resolve the age of the terraces. The IRSL ages suggest correlation of the Otamaroa Terrace with MIS 5a and the Te Papa Terrace with MIS 3, implying MIS 5e and 5c terraces are not present along the northern Raukumara Peninsula. Wilson et al. (2007) found these results unsatisfactory. The MIS 5a and MIS 3 correlation was considered unlikely given the higher relative sea levels during the earlier highstands and the consistent uplift of the peninsula. They also argue development of the terraces within MIS 5 is consistent with regional loess chronology and geomorphological characteristics of Pleistocene marine terraces. It is noted that, IRSL methods have developed significantly since this publication and it is likely the methods used for this work produced underestimated ages (Section 5.3.4).

### 4.1.4   Hawke's Bay region

Mahia Peninsula, in the Hawke's Bay region, is subject to coastal deformation and uplift due to its position within the accretionary wedge of the Hikurangi Margin forearc and its proximity to the Lachlan Anticline, the axis of which is directly offshore (Figure 2). Berryman (1993) provided a comprehensive description of the peninsula and its seven Pleistocene marine terraces, of which three were correlated to MIS 5: Mahia III (5e), Mahia II (5c), and Mahia I (5a). All three terraces are identified on the northeast part of the peninsula. Mahia III is also found along the southwest coastline and across Portland Island directly to the south. The terraces are composed of basal platforms cut into underlying bedrock with overlying marine sand cover beds. The marine sands grade upward into aeolian dune sands, in turn overlain by sequences of tephra and loess. The cover beds are described as varying in thickness from approximately 4 m for the lower terraces to 20 m for the higher terraces, but which terraces are considered lower and higher is not stated. Berryman (1993) correlated Mahia III to the MIS 5e peak at 124 ka, citing an amino acid minimum age for a wood sample (Pillans, 1990a) and palynological evidence (personal communication with Matt McGlone). The stratigraphy of the tephra-loess cover beds was used to determine the age of the

remaining terraces. Mahia I, II, and III were differentiated from the five other marine terraces on the peninsula by the composition of their cover beds, which include three loess units with intervening deposits of Kawakawa Tephra and Rotoehu Tephra. Berryman (1993) mapped numerous spot altitudes of shoreline angle positions, which were determined by subtracting a mean, undefined, thickness for terrestrial cover beds. Due to the deformation of the peninsula, the elevation measurements from each terrace vary significantly: Mahia I (83-116 m amsl), Mahia II (51-150 amsl), northwest Mahia III (130-150 m amsl), southwest Mahia III (42-132 amsl). The shoreline angle elevations used by Berryman (1993) to calculate uplift rates, chosen for their position approximately parallel to the strike of the Lachlan Anticline, are considered direct RSL indicators: Mahia I (92 ± 10 m amsl; Figure 3C WID 39), Mahia II (124 ± 10 m amsl; Figure 3C WID 40), and Mahia III (147 ± 10 m amsl; Figure 3C WID 41).

The uplifting Cape Kidnappers, on the southern margin of Hawke's Bay and opposite of the Mahia Peninsula, is also within the accretionary wedge of the Hikurangi Margin forearc and is bisected by the Kidnappers Anticline. Cashman and Kelsey (1990) reference the Kidnappers Terrace, described as ~10 m of marine cover beds overlying a basal platform and correlated to MIS 5. The elevation of the terrace surface varies from 200 m amsl at the axis of the Kidnappers Anticline, to 100 m amsl on either side. Additional details are not available; of the references referred to by Cahsman and Kelsey (1990), King (1932) provided insufficient information to define a RSL indicator, the remaining Kingma (1971) and unpublished Masters Theses (Kamp, 1978; Hull, 1985) were not made available in the course of this work.

### 4.1.5    Wairarapa and Wellington regions

The Wairarapa coast, south of Hawke's Bay, is subject to uplift and deformation through folding and faulting as it forms the southern portion of the accretionary wedge of the Hikurangi Margin forearc (Figure 2). Further south, deformation of the Wellington coast is driven by the northeast-trending strike-slip faults of the North Island Dextral Fault Belt that have uplifted basement blocks to form the Axial Ranges with intervening basins developed through differential subsidence (e.g. Wellington Harbour-Port Nicholson and the Hutt Valley) (Begg and Johnston, 2000; Lee et al., 2002). Uplifted marine terraces and subsurface marine sediments identified in well cores have been correlated to MIS 5 in multiple locations; however, few have been sufficiently described to qualify as an RSL indicator.

Ghani (1978) mapped four marine terraces in four locations between Riversdale and Wellington City, referring to the sequence as the Eparaima Marine Benches, subdivided from lowest to highest as EA, EB, EC, and ED. The best-preserved sequence is between Riversdale and Flat Point on the southeast coast. The remaining three locations, Cape Palliser, the lower Ruamahanga Valley, and the upper Ruamahanga Valley, are on the south coast and retain varying extents of the terraces, with EC the best preserved. The terraces are described as consisting of a wave-cut surface with overlying beach deposits and loess of undefined thickness. The age of each terrace is constrained by the number of "soil-units": two soil-units are found on the lower terraces (EA and EB, 80 ka and 84 ka, respectively), three on EC (100 ka), and four on ED (125 ka); however, the two oldest soil-units are not consistently present and are limited in use for spatial correlation of the terraces. Mapping of the terraces was done from ground-truthing of features identified in aerial photographs. Primary spot heights were measured by determining the depression of the sea horizon. Secondary spot heights were determined from vertical angles and distances from 20-chains-to-an-inch maps [1 in = ¼ mile]; overall spot height uncertainty was determined by Ghani (1978) to be ±3 m. The spot heights were then used to map the stranded shorelines and contour the outcrop pattern of each terrace from which vertical crustal movements were derived. None of the spot heights were added to the database due to concerns for precision of location and elevation, unknown thickness of cover beds, method of age constraint, and the extensive deformation of the coastline - numerous faults, anticlines and synclines run parallel and perpendicular to the coast. Between the four sequence locations, terrace surface elevation for each terrace varied in the tens of meters; however, the only surface to exceed 200 m amsl is ED.

Mildenhall (1995) analyzed the pollen profile of coastal sediment sequences retrieved from two drillholes (Petone and Seaview) in the Lower Hutt Valley and identified nine Biozones. Biozone P6 was tentatively correlated to the Last Interglacial as the taxa present are currently restricted to the north of the North Island and indicate a period of maximum warmth. Postglacial radiocarbon ages from the overlying Biozone P9 and the glacial climates indicated from the intervening Biozones P7 and P8 are used to strengthen the correlation to MIS 5. The sediment lithology from both drillholes, inclusive of marine shell, indicates a marine environment; however, proximity to the paleo shoreline is unknown. Biozone P6 is found within the Petone drillhole at 82.0 m to 105.4 m depth of burial from the land surface, which is ~1 m amsl. The biozone may extend higher within the sequence as there is a 6 m sampling gap between this biozone and the overlying Biozone P5. The sediments of Biozone P6 thin and increase in elevation to the east where they are found within the Seaview drillhole at 48 m to 65 m depth of burial from the land surface, ~1m amsl. Biozone P6 was later referred to as the Wilford Shellbed, assigned an age of 128 ka to 71 ka, and used in estimates of subsidence rates within the Lower Hutt Valley (Begg et al., 2004). Due to the uncertainty in depositional environment, Biozone P6 is considered marine-limiting (Figure 3D WIDs 568 and 569).

Heine (1974; 1979; 1982) identified numerous terrace surfaces in and around Wellington City. In the latter two studies, Heine used a method described as, "a micro-survey of the topography to identify any systematic pattern in the elevations" (Heine, 1979 p. 379), to deduce any relationship between coastal terraces and local tectonics. Over 350 elevation surfaces above mean sea level were identified within the city of Wellington using altimetry and from topographic maps with the only criteria that the surface was "near to level". The spot heights were then subdivided into 56 discrete 'levels' within which ten 'major' terrace levels were identified and tentative correlation to other New Zealand marine terraces were made on the basis of terrace altitude. Heine (1982) expanded the spatial extent of this approach to include the entire Wellington Peninsula and suggested levels up to 200 m to be marine in origin with correlations with MIS 5 (18-20 m and 40-42 m), MIS 7 (70 m), and MIS 9 (107 m) highstands. The intermediate levels were attributed to sea-level fluctuations. Although spot heights are provided in publication maps, terrace descriptions as basement rock overlain by Quaternary terrestrial sediments of undefined thickness, and the tentative age correlation precluded the identification of a RSL indicator.

Leamy (1958) described a number of terraces within Porirua Harbour, an embayment in the coast north of Wellington City. Two terraces, described as of "probable" marine origin underlain by gravels, were correlated by height to European Main Monastirian and Late Monastirian, recognized as two stages of the Last Interglacial (*sensu lato*; Table 1). Aneroid barometer measurements of the terrace surfaces placed the Main Monastirian terrace surface at 16.5 m to 15.5 m amsl. The Late Monastirian terrace surface was measured to 5.1 m amsl. These datapoints were not added to the database due to the uncertainty of their origin (fluvial or marine) and age. Webby (1964) described additional sections around the harbour identifying "red-weathering horizons", of which some were correlated with the Last Interglacial. Estuarine sediments underlying a 60 ft [18.3 m] marine terrace correlated with the Last Interglacial are identified in section; however, the stratigraphic column is a composite with the elevation of the sequence surface described as varying between 3.0 and 26.5 m and the location was not included as a RSL indicator. Offshore and slightly to the south of the Porirua Harbour mouth, multiple terraces on Mana Island have been described as marine in origin (Williams, 1978). Although they are attributed to Pleistocene high sea levels, no other age constraint has been provided.

### 4.1.6 Southern Manawatu-Whanganui and northwestern Wellington region

West of the Axial Ranges, the onshore eastern margin of the Whanganui Basin begins as a narrow coastal plain at Paekakariki and broadens northward into the southern Manawatu-Whanganui region to form the Horowhenua lowlands (Begg and Johnston, 2000). Spatially variable deformation within the region is associated with faulting of numerous basement faults of the North Island Dextral Fault Belt and overall regional uplift of the ranges (Sewell, 1991; Begg and Johnston, 2000). Early

studies of the region led to the identification two deposits of interest: the Otaki Formation (Oliver, 1948) and the Tokomaru Marine Terrace (Cowie, 1961), which were not explicitly correlated until 1988 (Palmer et al.).

Oliver (1948) formalized the Otaki Formation (originally Otaki Sandstone) designation for Late Pleistocene marine, beach, and dune sands deposited in the coastal region between Paekakariki and Palmerston North (Figure 3). The flat surface overlying the Otaki Formation was attributed to post-depositional erosion and the surface does not appear to be recognized as a marine
terrace until Palmer et al. (1988). An exposure within the outer, seaward terrace riser allowed for a detailed lithological description of the Otaki Formation by Te Punga (1962); however, with limited paleoenvironmental interpretation and no effort for lateral correlation of the sequence with other deposits in the region. Fleming (1972) summarized the sequence as consisting of a basal platform upon which is emplaced a sequence of transgressive marine beach gravel and sands (basal Otaki) that transition to beach-derived, micaceous dune sands (upper Otaki) deposited as sea level fell. Awatea Lignite was deposited by
swamps ponded within the dunes. The Otaki Formation was correlated to the Last Interglacial at ~80,000 to 120,000 years based upon the minimum radiocarbon age of the lignite (>45 ka; N.Z. $^{14}$C, No. 65) and regional geology (Te Punga, 1962; Fleming, 1972). As described, it is difficult to identify the transition of marine to terrestrial sediments; however, a sponge spicule bearing unit ('d') is considered marine-limiting point at 22.6 ± 2 m amsl (Figure 3D WID 413).

The Tokomaru Marine Terrace has been described as extending north from Levin to Palmerston North and identified within
the intervening Manawatu Valley (Cowie, 1961; Hesp and Shepherd, 1978). Hesp and Shepherd (1978), described the terrace sediments as partly marine and mapped the terrace as cutting into the Rapanui Formation. The terrace was correlated with MIS 5 on the basis that the "upper beds" had been tentatively correlated with the Oturian (MIS 5) Stage by Fleming (1971). Although Hesp and Shepherd (1978) make no mention of the Otaki Formation, it is near certain that the correlation referred to is that of the Otaki Formation section described by Te Punga (1962) and interpreted by Fleming (1971; 1972), who does not
relate the Otaki Formation to the Tokomaru Terrace.

The correlation of the Tokomaru Terrace in the Manawatu Valley with MIS 5 prompted Palmer et al (1988) to identify a marine terrace surface at Otaki as the Tokomaru Terrace underlain by Otaki Formation. The terrace surface is described as being near continuous from Otaki through to Whanganui allowing confident correlation of the Tokomaru Terrace and the Rapanui Terrace. The identification of the terraces as equivalents and the correlation of the Rapanui Terrace with MIS 5e
(Pillans, 1983; 1990b) implies the mapping by Hesp and Shepherd (1978) of the Tokomaru and Rapanui Terraces within the Manawatu Valley is incorrect. Although the basal platform is not visible, Palmer et al. (1988) identify both Otaki beach sand and Otaki dune sand to the north and south of Otaki, inclusive of the terrace riser and sediments described by Te Punga (1962). A cross-section of the inner terrace riser, exposed by the Otaki River, provides the location for the fossil marine cliff, which would have been cut by peak sea level during MIS 5e; however, the shoreline angle is obscured by Otaki dune sand. Height
constraint for the Otaki beach sand is not provided other than described as ~30 m amsl for the area and therefore, the location of the fossil marine cliff is identified as direct sea-level indicator but assigned a large uncertainty, 30 ± 5 m amsl (Figure 3D WID 571).

Detailed stratigraphic and lithological analysis of the Otaki Formation accompanied extensive mapping in the region between Otaki and Tokomaru (Sewell, 1991). Sewell (1991) retained use of the Tokomaru Marine Terrace designation for the basal
platform underlying the sediments of the Otaki Formation and the fossil marine cliff cut at the peak of MIS 5e sea level. The extensive mapping confirmed the presence of two later terraces, designated Post Tokomaru Marine Terrace (PTMT) 1 and PTMT 2, correlated with MIS 5c and MIS 5a respectively (Palmer et al., 1988; Sewell, 1991). The age of PTMT 1 and PTMT 2 was determined from their relationship with the Tokomaru Marine Terrace (an extension of the MIS 5e Rapanui Terrace), the similarity of the terrace succession to the Rapanui, Inaha, and Hauriri Terrace succession to northwest, and comparative
loess stratigraphy overlying PTMT 1 – it is overlain by four loess units similar to other marine terraces correlated with MIS

5c on south North Island. Sewell (1991) referred to the marine and dune sediments of PTMT 1 and PTMT 2 as Otaki Formation. Although the detailed stratigraphic and lithological descriptions allowed for a comprehensive paleoenvironmental interpretation of the region, the described sections are not assigned a height in relation to modern sea level and RSL indicators cannot be derived.

## 4.2 South Island

The South Island is comprised of seven government regions, listed in a clockwise order from the northernmost: Nelson, Tasman, Marlborough, Canterbury, Otago, Southland, and West Coast Regions (Figure 2). The majority of RSL indicators have been identified in the Marlborough and Canterbury Regions (Figure 4), which will be reviewed last. No RSL indicators have been correlated to MIS 5 in the Nelson or Tasman Region; although geological mapping in the region indicates the presence of marine terraces in the area of Nelson City (Johnston, 1979; Rattenbury et al., 1998). Speleothems within a cave correlated with a 60 m amsl terrace within the Tasman region were dated to MIS 5e; however, it was concluded that the cave (and terrace) predated the Last Interglacial (Williams, 1982).

Terrace sequences of the South Island are less continuous and more fragmented than the North Island and attempts at broad correlation of terraces within South Island and New Zealand are fewer. Identification of paleo-shorelines and features on the west and south coast of the South Island is made difficult by proximity to the Alpine Fault (driving coastal deformation) and by dense vegetation (Wellman and Wilson, 1964; Suggate, 1992). These regions also remain remote and difficult to access. A large portion of the Canterbury coast is subject to subsidence and the northeast coastline is positioned within the Marlborough Fault System (Rattenbury et al., 2006).

**Figure 4: Position, elevation, and age correlation of the South Island RSL indicators within the WALIS database and described here. Elevations are provided above mean sea level (Section 3) except for WID 51 and WID 768 (Panel C), for which the sea-level datums are mean low tide and mean higher high water, respectively. Each RSL indicator is identified with their unique WALIS database identifiers (e.g. WID 34) and is correlated in text with their original publication. Symbols of indicators that would stack extend offshore for clarity. Marine- and terrestrial- limiting points are indicated with up- and down- arrows, respectively. . Locations and landscape features mentioned in text are also shown with built-up areas (e.g. towns and cities) indicated by stars. Hill shade data sourced from the LINZ Data Service and licensed for reuse under CC BY 4.0 (LINZ, 2020b).**

### 4.2.1 West Coast and Southland

Extensive mapping of the northern West Coast Region in the mid-20[th] Century identified numerous marine terraces and associated formations (Suggate, 1965; 1992). In one of the earliest studies Suggate (1965) described terrace sequences for three separate segments of the coastline (Hokitika to Greymouth, Greymouth to Fox River, and Charleston to Westport) and designated MIS 5 marine sediments the Awatuna Formation. The best-preserved terrace sequence of the northern West Coast is found between Charleston and Westport, where four marine terrace surfaces are identified; two of which Suggate (1965) correlated to MIS 5. Nathan (1975) mapped six distinct terraces north of Charleston, correlating the marine sands and gravels underlying two of the terraces with the Early and Late Oturi Interglacial (MIS 5): the Virgin Flat Formation (6 m to 15 m amsl) and Waites Formation (also 6 m to 15 m amsl), respectively. In later reassessment of the northern West Coast, Suggate (1992) estimated the age of the terraces from each of the three coastal segments previously described using the stratigraphic relationship of the paleo-shorelines to glacial deposits, previously published radiocarbon ages (the majority of which provide minimum ages), and individually calculated uplift rates for each sequence (with consideration for differential uplift along the coast) to correlate terraces with paleo-sea levels derived from the Huon Peninsula sea-level curve (Chappell and Shackleton, 1986). The Awatuna Formation was subdivided and correlated with MIS 5e and MIS 5c; the younger deposits retaining the Awatuna designation and the older MIS 5e deposits renamed Rutherglen Formation (Table 1; Suggate, 1985; 1992). The Virgin Flat and Waites Formations (Nathan, 1975) were correlated with the Rutherglen Formation (Suggate, 1985); however,

insufficient information is present in either publication for an RSL indicator north of Charleston. The type sections for the Awatuna Formation and the Rutherglen Formation are identified as direct indicators (Figure 4A WIDs 824 and 825; Suggate, 1965; 1985; 1992), both within 20 km proximity to Greymouth. The formations consist of rusty, cemented marine sands and gravels located near to former marine cliffs. The surface elevations of the Awatuna and Rutherglen Formations are 52 m and 67 m amsl, respectively, with surface elevation decreasing to the south. Preusser et al (2005) applied multiple luminescence dating techniques to polymineral fine-grain and K-rich feldspar sediment samples taken from within the Awatuna Formation section and correlated the formation with MIS 5. Due to concerns with method expressed by Preusser et al (2005) and discussed below (Section 5.3.4), both the Awatuna and Rutherglen Formations are correlated in WALIS with MIS 5, *sensu lato*.

To the south, between the Moeraki and Haast Rivers, Nathan and Moar (1975) identified and briefly described three terraces: Sardine-2, Sardine-1, and Knights Point. The Sardine-1 Terrace is interpreted as fluvial and its surface is found at an intervening height (56.4-59.4 m amsl) to the marine Sardine-2 (24.4-32 m) and Knights Point (140-147 m) Terraces. The results of radiocarbon analysis (mostly of wood) were considered ambiguous with a mixture of ages implying contamination by either older or younger carbon; therefore, age was determined primarily by correlation with the glacial-interglacial sequences described in the north by Suggate (1965). Sardine-2 was correlated broadly to MIS 5, Sardine-1 to MIS 7 or possibly MIS 6 due to the harsh climatic conditions indicated by pollen within the sediments, and Knights Point Terrace was considered to pre-date MIS 7. The Sardine-2 Terrace forms an ~400 m-wide strip on the north side of Ship Creek and dips gently to the south. The marine sands are described as ilmenite-rich, typical of beach and near-shore sand found along the modern coast to the north, and are considered a direct RSL indicator (Figure 4A WID 50); however, elevation constraint is poor and the uncertainty on the actual elevation of these sediments is large, 27.4 ± 7.87 m amsl.

Due to the rate of vegetative growth, the sections described by Nathan and Moar (1975) were quickly overgrown, but reanalysis of the Knights Point Terrace was made possible by road-widening early in the new century (Cooper and Kostro, 2006). Cooper and Kostro (2006) provided a detailed stratigraphic description and interpretation with IRSL age constraint. Seven lithofacies indicate a transgressive system within a high-energy coastal environment, possibly inclusive of storm event deposits. In section drawings, the basal platform varies between 104.2 m and 105.5 m amsl. The surface elevation of the uppermost facies, Facies 7, reaches a maximum height of 113 m amsl. Facies 7 is interpreted to have been deposited within the breaker or swash zone of a coastal beach and is considered a direct RSL indicator (Figure 4A WID 52). The lower elevation of Facies 7 is not provided and is defined within WALIS as existing between 113 and 110 m amsl because the unit is drawn as exceeding 3 m thickness in the publication stratigraphic sections (Figure 3 of Cooper and Kostro, 2006). Two OSL samples (KP-01-TL and KP-03-TL) were collected, both from Facies 7. The resulting age from KP-01-TL of 123 ± 7 ka was accepted. KP-03-TL had an older age (146 ± 8.4 ka) but the sample was reported to be in radioactive disequilibrium and was considered likely to be an overestimate of age. However, based on the SAR measurement procedure used at the time, it is possible that these ages suffer from 'anomalous fading' (Wintle, 1973) and should be viewed as minimum ages only. Although, correlation to other marine terraces in New Zealand is discussed, the ramifications of the MIS 5e age of Knights Point Terrace for the ages of the Sardine-1 and Sardine-2 Terraces is not (Cooper and Kostro, 2006). Given the uncertainty of the Knights Point ages, the original interpretation by Nathan and Moar (1975) with MIS 5 at ~ 24 m to 32 m amsl, retains validity.

Bull and Cooper (1986) claimed to have identified the remnants of numerous marine terraces in the Southern Alps on either side of the Alpine Fault, inland of the West Coast locations described immediately above. Terrace age was determined by correlation to the Huon Peninsula terrace sequence in Papua New Guinea (Chappell, 1974). Terraces correlated with MIS 5 were identified by morphology: notched spur ridges associated with sea cliffs and "shore platform" remnants overlain by well-rounded quartz pebbles and cobbles interpreted to have been formed within a beach environment. The average altitude of these terraces is between 686 m and 899 m amsl. This work was refuted by Ward (1988b) on the basis of terrace morphology, the origin of the quartz pebbles, and the terrace altitudes. Ward (1988b) argued terrace morphology is not preserved as described,

the pebbles could be moa gizzard stones, and cites concerns regarding correlation to the distant Papua New Guinea sequence and the implications of the inferred uplift rates, suggesting that any semi-regular sequence of terrace altitudes could be correlated with the Huon Peninsula – points refuted by Bull and Cooper (1988). Pillans (1990a) also considered the evidence for a marine origin of the terraces ambiguous and demonstrated, not only that the concerns of Ward (1988b) regarding the altitude correlation with Huon Peninsula were justified, but also that the terraces are as likely to be ridge crest notches controlled by drainage density, slope, and uplift rate. It is the opinion of these authors that the argument remains unresolved and further research would be necessary to conclude whether the features are marine terraces.

Fiordland, forming the southwest corner of Southland and the South Island, is extremely rugged and largely inaccessible. Although numerous marine terraces have been identified, they have only been marginally described and studied (Wellman and Wilson, 1964; Bishop, 1985; Ward, 1988a, Kim and Sutherland, 2004). Wellman and Wilson (1964) identified a marine bench at 500 ft [152 m] cut across a penultimate glaciation moraine within Hollyford Valley and continuing to the coast where it forms a notch in the headland on the south side of Martins Bay. Bishop (1985) identified eleven levels of marine "surfaces", inclusive of the present-day intertidal reef to the highest at ~1000 m amsl, between Preservation Inlet and Knife and Steel Harbour from which he inferred uplift rates. The terrace sediments are poorly exposed and not described. The age of the Pleistocene terraces was determined by comparing the terrace sequence morphology to that of the terrace sequence at Taranaki on the North Island (Pillans, 1983). The terrace surface correlated with 120 ka, designated $h_6$, is extensive with an average altitude of 370 m amsl at the strandline. Terraces $h_5$ (300 m) and $h_4$ (210 m) are correlated with MIS 5c and 5a respectively. There are localized outcrops of rounded pebbles and cobbles and fossil islands and stacks rise above the terrace surface south of Preservation Inlet. However, no precise localities or descriptions are provided in the publication and a suitable RSL indicator was not identified.

Ward (1988a), working between the outlet of Big River and Te Waewae Bay, slightly overlapping the study area of Bishop, described a sequence of at least 13 marine terraces reaching up to 1000 m altitude. Finding a lack of suitable material for any geochronological method, terraces were matched to late Quaternary oxygen isotope stratigraphy of deep-sea cores using a "simple uplift model". The terraces are described in general terms: the lower terraces, inclusive of the last interglacial, consist of loess overlying several meters (up to 20 m locally) of marine gravel and sand on bedrock. Terraces 2, 3, and 4 are correlated to MIS 5a, 5c, and 5e respectively. The height of the inner margin of each terrace above mean sea level was estimated from a combination of 1:63,360 topographic maps (NZMS1 S173 & 174), oblique air and ground photographs, and limited ground truthing with an altimeter: Terrace 2 at 60 m, Terrace 3 at 90 m, Terrace 4 at 140 m. These heights are distinctly different from the height of the terraces correlated to MIS 5 (*sensu lato*) by Bishop (1985) for the adjacent coastline to the west (Table 1). Ward (1988b) correlates the 370 m terrace in his study to MIS 9. Although the regions described by Bishop (1985) and Ward (1988a) are bisected by the Hauroko Fault, both authors recognize that there is no observable displacement of the terrace surfaces due to the fault and the difference in terrace heights is due to different interpretations of terrace age within each study.

Kim and Sutherland (2004) applied two methods of cosmogenic nuclide surface exposure dating ($^{10}$Be and $^{26}$Al) to date shore platform erosion surfaces in bedrock located to the west of the Bishop (1985) and Ward (1988a) studies, between Newton River and Dusky Sound. Three direct RSL indicators from the rocky shore platform surface were derived from this study: 65 ± 13 m amsl (WID 53), 55 ± 11 m amsl (WID 54), and 72 ± 14.4 m amsl (WID 55; Figure 4B). They are not considered high quality for a number of reasons. Sample locations are only briefly described as "uplifted beach region" and "uplifted sea stacks and reefs", providing little context. Strandline elevations are generalized, determined from air photographs, a map of undefined scale, and field interpretations. The terrace strandline correlated with MIS 5e is described as 65 m amsl even though sample elevations are reported to have been collected from 51 m to 72 m amsl. The terrace ages cluster between 102.1 ± 4.6 ka to 118 ± 5.9 ka. The ages derived from $^{10}$Be and $^{26}$Al are consistent with each other, but of the three samples documented in WALIS (because the sample location and elevation could be constrained), an $^{26}$Al age was derived from only one (WC02_15). The

designation by the authors of the 65 m terrace to MIS 5e provides support to the terrace ages to the east derived by Ward (1988a); although, in the east the terrace described at 60 m is assigned to MIS 5a. The cluster of ages reported by Kim and Sutherland (2004) are better correlated to MIS 5c with a peak sea level at ~109 ka, which would result in better agreement between the studies. However; later work (Putnam et al., 2010; Kaplan et al., 2011) to calibrate *in situ* cosmogenic [10]Be production rates indicate lower production in the Southern Hemisphere and cosmogenic exposure ages published prior to 2010 are likely to be too young, probably by at least 12% (Williams et al., 2015). Due to these discrepancies, the terrace is assigned to MIS 5 (*sensu lato*) within WALIS. Kim and Sutherland dated an additional surface between 92 m and 136 m amsl, which is at similar elevation to Terraces 3 (90 m) and 4 (140 m) correlated by Ward (1988a) with MIS 5c and 5e respectively, but all results were considered anomalously low and no age determination for that surface was made.

Beavan and Litchfield (2012) report the coastline of eastern Southland, into the Otago Region, includes regions of stability, referencing Turnbull and Allibone (2003). However, this is a reference to a geological map without any precise locations, elevations, or geochronological constraint. No other publications detailing potential RSL indicators in the region were identified in our review. Nevertheless, this indicates eastern Southland is a potential source of valuable RSL indicators little affected by tectonic deformation and worthy of further investigation.

### 4.2.2    Otago Region

The Otago Region has historically been considered tectonically stable or to have had little tectonic movement (Gage, 1953; Gibb, 1986; Pillans, 1990a, Beavan and Litchfield, 2012); however, a number of studies on coastal faults (e.g. Litchfield and Lian, 2004; Taylor-Silva et al., 2020; Craw et al., 2020) suggest that this is at best a generalization that is not true for the entire Otago coast. Relatively few studies of shorelines have been completed there since the early review of Gage (1953). Ongley (1939) mapped an intermittently preserved 50 ft to 60 ft [15.2-18.3 m] terrace surface and underlying basal platform at 20 ft [6.1 m] northwards from the Clutha [then Molyneux] River mouth to Brighton. Cotton (1957), although referencing Ongley (1939), refers to this terrace as the 40 ft [12.2 m] terrace and discusses the possibility that it and the underlying basal platform are of Monastirian age; i.e. Last Interglacial. No specific locations are provided for the terrace, but it is described to extend from the Clutha River north to Brighton and north from Shag River into Canterbury.

Later work at Taieri Beach, in the region south of Brighton, was completed by Rees-Jones et al. (2000) and Litchfield and Lian (2004). The coastline south of Taieri Mouth (Waipori River) is on the upthrown (east) side of the reverse Akatore Fault. Rees-Jones et al. (2000) revisited a raised beach deposit identified by Bishop (1994) as 'h$_2$' and sampled sediments for IRSL determination. The terrace surface is described as occurring at 8-11 m above "high sea level" with cover beds consisting of two sand units upon a wave-cut platform capped by loess. The lower sand unit, ~1 m thick, is a coarse, well-sorted, reddish-orange sand and gravel, and is considered a direct RSL indicator of a beach deposit at 5.4 ± 2.2 m above "high sea level" (Figure 4C WID 47). The upper sand, ~1.5 m thick, is fine and lightly weathered yellow sands of aeolian origin. The overlying loess varies between 1 to 4 m in thickness. Samples were taken from all three units and were dated using IRSL. The IRSL signal measured from K-feldspar inclusions inside of quartz sand grains coming from the beach sand (sample NZ10) returned an age of 71 ± 14 ka. The deposition of this unit, based upon this age estimate, was correlated with MIS 5a. Litchfield and Lian (2004), citing concerns for the luminescence technique used, resampled the beach sand (5.5 ± 2.5 m amsl Figure 4C WID 48) and analyzed it using both thermoluminescence (TL) of quartz (Sample W2857) and IRSL of polymineral silt grains (Sample TBE1). The resulting ages of 117 ± 13 ka (TL) and 117 ± 12 ka (IRSL) prompted reassignment of the beach sand to MIS 5e. However, the uncertainties provided are at 1-sigma deviation and the ages are consistent with any time over a span of ~50 ka. Furthermore, anomalous fading was observed in the IRSL sample (TBE1) indicating a potential underestimated age, but because age was consistent with the TL sample (W2857), the fading was considered insignificant and no correction was

made. There are valid concerns regarding the luminescence methods at the time both studies were completed (Section 5.3.4) and the ages must be considered with skepticism.

The movement along the Akatore Fault, which runs roughly sub-parallel to the coast for ~22.5 km before moving offshore near the mouth of the Tokamairiro River to the south, had been recognized in the earlier work of Cotton (1957). He cited the mapping of the 40 ft surface (Ongley; 1939) to argue that the movement appeared to be localized to the coastal regions immediately surrounding the fault (not fully recognized at the time) as they are the only locations where the terrace shows any offset in height. Trenching across the fault revealed at least three reverse fault ruptures since c. 13,300 yr BP, with single event displacements of 1.6 m to 2.5 m (Taylor-Silva et al., 2020). Ground penetrating radar profiles show that the displacement of the basal platform is ~3 m and the marine terrace has likely only been displaced by the relatively recent events, following a minimum 110 ka period of quiescence on the fault. However, this does imply that the RSL indicator at Taieri Beach, has been displaced by potentially up to 3 m. The distinction in age of the beach sand is important for calculating uplift rates and implications for stability along the Otago coastline (Litchfield and Lian, 2004). If the beach sand (5.5 ± 2.5 m amsl) is correlated to MIS 5e, then relatively little uplift along the coastline since the Last Interglacial until reactivation of the Akatore Fault is inferred; however, if deposited during MIS 5a, then the elevation of the beach sand would indicate significant uplift has occurred.

Litchfield and Lian (2004) also analyzed a marine terrace location at Warrington, within Blueskin Bay north of Dunedin. The terrace surface reaches 4-8 m amsl and is slightly deformed by a tilt to the northeast and a displacement of 1 m by a small, unnamed northwest-striking fault, providing additional evidence of instability within the Otago Region. Although the terrace is located adjacent to the Waitati Fault, that fault is not considered responsible for any deformation as it displays no evidence of Quaternary activity. The marine terrace sediments, exposed in a modern marine cliff, consist of Quaternary volcanic boulder beach deposits overlain by quartzofeldspathic beach sand of undefined thicknesses. The beach sands are overlain by loess and the sequence variably overlies an older loess or basal platform cut into Miocene volcanics. The contact between the beach boulders and beach sands at 1.0 ± 1.5 m amsl is considered a direct RSL indicator of beach deposition (Figure 4C WID 49). One meter of uncertainty was added to the measurement because the elevation is derived from a stratigraphic drawing, which is referenced to a 0 m sea-level mark (Figure 2 in Litchfield and Lian, 2004). Samples for IRSL dating of K-feldspar were taken from the beach sand (WBE2) and the underlying loess (WBE1), providing ages of 97 ± 11 ka and 96 ± 5 ka, respectively. Although inverse, the ages are practically indistinguishable and both samples were considered to have suffered from anomalous fading and to be minimum ages. Litchfield and Lian (2004) argue for MIS 5e deposition of the beach sand due to the likelihood that the underlying loess was deposited during MIS 6.

Kennedy et al. (2007) surveyed the coast north of Warrington to Oamaru and described the height of the last interglacial shoreline as varying between 5 and 8 m above low water level. It is unclear how accurate the measured height reflects paleo sea level as the surface is described as a mixture of deposits consisting of eroded platforms, gravel beaches, and estuarine mud and beach sand and a detailed stratigraphic description is only provided for a section at Shag Point. On the southeastern end of Shag Point, imbricated boulders and marine sands are deposited upon a basal platform surface cut into carbonaceous mudstone. The boulders (≤2.5 m long $a$-axis, 0.2 m to 1.2 m thick $c$-axis) are only found on the seaward edge of the terrace. Directly landward of the boulders is the well-sorted and laminated marine sand, which continues landward and is visible on both sides of the point. The boulders and sand are overlain by two distinct loess units of ~2 m total thickness. The distance from the inner margin is not stated and an accurate thickness of the marine sand is not provided; therefore, the basal platform at 7.4 ± 0.25 m above mean low water level is considered marine limiting (Figure 4C WID 51). The sand directly behind the boulders was dated by IRSL analysis of the Na-feldspar component (WLL181), providing an age of 81.9 ± 11.7 ka. The overlying loess deposits were also analyzed and indicate two younger phases of deposition, 78.6 ± 4.2 ka and 28.9 ± 4.4 ka. The apparent MIS 5a age of the marine sands is used to constrain the age of the boulders and argue for deposition by a tsunami

wave not exceeding 3 m. However, given the likely location of sea level during MIS 5a (-10.5 ± 5.5 m; Creveling et al., 2017) and concerns for the optical dating of Na-feldspar (Section 5.2.3), this interpretation must be regarded with skepticism.

Sections of the MIS 5 marine Hillsgrove Formation have been described on the southern coastline of Cape Wanbrow, located immediately south of Oamaru (Grant-Mackie and Scarlett, 1973). The two sections consist of nearshore marine sediments, likely deposited within a few meters depth of water, as indicated by the marine and terrestrial mollusc and avian fossil assemblages. The section at the north end of the South Oamaru Beach is described for height above the base of the cliff backing the modern beach; constraint above modern sea level is not provided and a relative sea level indicator cannot be derived. The

section to the east is constrained to height above a modern shore platform with the inner margin designated '0 m'. The provided elevations are considered as heights above mean higher high water (MHHW) because the inner margins of modern shore platforms are typically identified at that height in relation to mean sea level (Rovere et al., 2016). The sediments are deposited upon a basal platform at 4 m above MHHW, formed on Upper Eocene volcanics and around an erosion-resistant stack of Upper Eocene limestone. A basal gravel (0.3 m to 1.0 m thick) grades to a consolidated weathered yellow sand and clay with

some pebbles (1.5 m to 2.0 m thick) containing avifauna fossils of large moa, which grades upwards through two gravel bands and into overlying loess and paleosols. The good preservation of the fossil fauna, the height of the deposits above sea level, and a minimum radiocarbon age (undescribed) are used to correlate the marine sediments to the Last Interglacial (*sensu lato*; Grant-Mackie and Scarlett, 1973). Only the basal gravel is considered marine following the later description of the overlying sands as dunes (Worthy and Grant-Mackie, 2003) and is designated a direct RSL indicator (Figure 4C WID 768) at 4.5 ± 1.1

m above MHHW.

### 4.2.3    Canterbury and Marlborough Regions

The Canterbury Plains is a coastal plain stretching 160 km from Timaru to the Waipara River. The plain is formed of a series of coalescing alluvial fans, emerging from the Southern Alps and exceeding 50 km in width, that has been subsiding under the weight of glacial advances and outwash. The last interglacial RSL record of the southern Canterbury Region is dominated by

stratigraphic correlation of the Bromley Formation beneath the Canterbury Plains. Brown et al. (1988) correlated the subsurface strata described in water well cores on the basis of gravel aquifers and intervening fine-sediment strata. The type section for the last interglacial (*sensu lato*) Bromley Formation is derived from two wells within the Bromley suburb (M35/1875 and M35/1926; Figure 4Di WIDs 74 and 75). The lithology consists of beach, lagoonal, dune, and coastal swamp sediments associated with rising, high, and declining sea level. The broad definition of the Bromley Formation has limited its use to a

terrestrial-limiting point. Twenty-one indicators, with upper and lower elevations, were identified (Figure 4D WIDs 65 and 66, Figure 4Di WIDs 67 to 85). The maximum depth of the Bromley Formation is 70 ± 10.3 m below mean sea level (bmsl) and the minimum depth is 8 ± 2.2 m bmsl. Heights were also constrained for WALIS entry by use of the Canterbury Regional Council well database (www.ecan.govt.nz/data/well-search; Environment Canterbury, 2020).

The well cores investigated by Brown et al. (1988) are predominantly clustered within Christchurch. Outside of the city, the

Bromley Formation was identified at 33 m to 56 m bmsl within a well (M36/1251, Figure 4D WID 72) on the shore of Lake Ellesmere, south of Banks Peninsula (Brown et al., 1988). An additional core from the nearby Gebbies Valley was analyzed by Shulmeister et al (1999). This 75 m long core retains a 200-ka record of marine transgressions and regressions. Sediments attributed to MIS 5e are not identified; however, aqueous deposits within Units G through K are correlated to MIS 5a and MIS 5c (Figure 4D WIDs 103 to 105). Units G and H record a transgressive succession and marine embayment, of which Lake

Ellesmere is considered a modern equivalent. Unit H (34.10 m to 24.67 m bmsl; WID 103) is marine-limiting. Unit I is correlated to a relative sea-level fall, probably during MIS 5b, before the next sea-level transgression represented by Units J and K. Unit J is interpreted as a beach or storm deposit, at or close to (within 5 m) of sea level, and is considered a direct indicator of RSL (22.74 m to 22.15 m bmsl; WID 104). Diatoms and phytoliths within Unit K (22.15 m to 14.12 m bmsl; WID

105) indicate a transition from a marine embayment to a lake or lagoon environment in probably interglacial conditions. The unit is considered a terrestrial-limiting point of sea level. The nine TL results for samples from 12.52 m to 43.75 m bmsl concentrate around 90 ka to 130 ka. Age estimates from throughout the core were further refined with radiocarbon ages and paleoenvironmental data to calibrate the sedimentary units to the oxygen isotope record (Martinson et al., 1987). Lagoon sediments (Unit C) were identified at greater depth (61.50 to 48.25 m bmsl) and correlated to MIS 7. Although in relatively close proximity (<5.5 km) to the Lake Ellesmere core reported by Brown et al. (1988), comparison of the core interpretations is difficult without age constraint or detailed stratigraphic description from the latter.

North of the Canterbury Plains and into the Marlborough Region, last interglacial marine terraces are uplifted by the North Canterbury fold-and-thrust belt, with terraces most prominent on the limbs of actively growing anticlines (Oakley et al., 2017). This coast was subject to extensive description by both Jobberns (1928) and Suggate (1965). One distinct terrace and multiple smaller, higher and more dissected terraces, occur at varying elevations intermittently along the coast. The marine terrace surfaces are in many locations obscured by later, glacial-period gravel fans. Additional studies (Powers, 1962; Fleming and Suggate, 1964) indicate locations of possible LIG marine terraces within the northern Canterbury Region. Studies providing the necessary stratigraphic descriptions, age and elevation constraint to identify RSL indicators have been concentrated in five coastal segments: the Waipara River to Motunau Beach, Haumuri Bluffs, Kaikoura Peninsula, Clarence River to Woodbank Stream, and Long Point to Boo Boo Stream. The latter coastal section is within the Marlborough Region.

North of the Waipara River, terrace remnants are continuous along a vertically-steep coast for a distance of ~6 km. The terraces are then absent for ~10 km until the coastline broadens at Motunau Beach (Jobberns, 1928; Jobberns and King, 1933; Suggate, 1965). Yousif (1989), using remote sensing, mapped the terrace surfaces following the nomenclature and age correlation of Jobberns (1928) and Carr (1970); although, these designations were often inclusive of multiple terrace surfaces at different altitudes within one interglacial. The most extensive terrace immediately north of the Waipara River is the Tiromoana Terrace (Carr, 1970); at Glenafric the surface tilts upwards to the northeast from ~45 m to ~80 m amsl (Oakley et al., 2017). At Motunau Beach the most extensive terrace has been alternatively identified as the Motunau Coastal Plain or Motunau Terrace and extends offshore to include Motunau Island (Jobberns, 1928; Jobberns and King, 1933; Suggate, 1965; Carr, 1970; Oakley et al., 2017). Jobberns and King (1933) provided early detailed lithology for the marine sediments of the Motunau Beach Terrace, including identification of marine and estuarine fossil mollusc found *in situ*. Representative stratigraphy drawn by Oakley et al. (2017) show both Tiromoana and Motunau Terraces to consist of a basal platform overlain in upwards succession by fossiliferous marine sand and gravel, alluvium, and loess. Oakley et al. (2017) sampled both terraces in multiple locations for IRSL of loess and marine sediments and AAR analyses of fossil mollusc shell. Elevations for the terrace inner margins are not provided in Oakley et al. (2017), but a later publication (Oakley et al., 2018) provides multiple elevations for the Tiromoana Terrace inner margin estimated from basal platform elevations measured in transect; however, exact locations for the inner margin elevations are not provided. The marine-limiting points for Tiromoana (61.5 ± 2 m amsl; Figure 4D WID 108) and Motunau Beach (70 ± 4 m amsl; Figure 4D WID 172) marine sediments were derived from AAR sample (GA5 and MB5, respectively) locations (Oakley et al. 2017).

The Tiromoana Terrace is correlated with MIS 5c indicating that higher terrace remnants in close proximity are of MIS 5e and MIS 7 age. Oakley et al. (2018) correlate the Bob's Flat Terrace, a less extensive terrace located to the southwest and northeast of the Tiromoana Terrace, with MIS 5e. However, this age interpretation appears to be reliant on the higher elevation of the terrace as no AAR or IRSL samples or data for this terrace are provided in Oakley et al. (2017). The IRSL and AAR results suggest at least two phases of deposition at Motunau Beach, a possibility first proposed by Suggate (1965). The distribution in ages indicates partial reoccupation, or incision with deposition, of the seaward edge of an earlier terrace during the MIS 3 highstand. The earlier, more extensive portion of the terrace, is correlated to MIS 5a, with the likelihood that the more minor terrace remnants at higher elevations are of MIS 5e and MIS 7 age. Oakley et al. (2017) argue that wave erosion explains the

minimal record older terraces in the area as well as the lack of a MIS 5c terrace above the Motunau Beach Terrace and MIS 5a terrace below the Tiromoana Terrace. The apparent higher elevation of the younger MIS 5a Motunau Beach Terrace to the older MIS 5c Tiromoana Terrace is not discussed by Oakley et al. (2018), but is likely due to the variable uplift rates along the coastline.

Further north the Haumuri Bluffs (alternatively spelled "Amuri") have been subject to many studies (Jobberns, 1928; Fleming and Suggate, 1964; Suggate, 1965; Ota et al. 1984; 1996; Oakley et al. 2017; 2018). Ota et al. (1984) has provided the most extensive description of the area. Four marine terraces are identified in the sequence; from highest elevation to lowest they are: Tarapuhi Terrace, Kemps Hill Upper and Lower Terraces, and Amuri Bluff Terrace. The terraces each consist of a basal platform overlain by marine sands, localized alluvial fan deposits, and loess. The cover beds of the Amuri Bluff Terrace include

extensive fluvial deposits. The Kemps Hill Terraces are not present at Haumuri Bluffs but appear in the sequence approximately 2 km to the south. The dip of the terrace surfaces indicates northward tilting.

The Tarapuhi Terrace retains a diverse molluscan fossil assemblage and collections were described by both Fleming and Suggate (1964) and Ota et al. (1996). The fossil molluscs were used by Ota et al. (1996) for AAR analyses in the first attempt to provide numerical age to the sequence and returned a broad age of 135 ± 35 ka. The cold-water environment indicated by

the fossil assemblage was used to constrain deposition of the Tarapuhi Terrace to MIS 5c. The age of the Kemps Hill Upper, Kemps Hill Lower, and Amuri Bluff Terraces were determined by best fit to the Huon Peninsula sea-level curve (Chappell and Shackleton, 1986) constraining formation within MIS 5 and MIS 4. Additional AAR analysis of the fossil mollusc assemblages within the Tarapuhi and Amuri Bluff Terraces, as well as IRSL analysis of the surrounding marine sediments, strengthened correlation of the Tarapuhi Terrace to MIS 5c and indicated formation of the Amuri Bluff Terrace within MIS 5a

(Oakley et al., 2017). Formation of the two Kemps Hill terraces is attributed to a double peak in sea level at the beginning of MIS 5a (90.6 ± 2.0 ka and 84.0 ± 1.6 ka) based upon the sea level curves derived from coral terraces (Lambeck and Chappell, 2001) and a $\delta^{18}O$ curve (Siddall et al., 2007; Oakley et al., 2018). RSL indicators are derived from the Tarapuhi Terrace inner margin (173.1 ± 2 m amsl; Figure 4E WID 107; Oakley et al., 2017; 2018) and an overlying fossiliferous marine unit (163 ± 2.2 m amsl; Figure 4E WID 102; Ota et al., 1996) and from the elevation of the Amuri Bluff Terrace inner margin (40.7 ± 5 m

amsl; Figure 4E WID 106; Oakley et al., 2017; 2018) and the contact between the basal platform and overlying marine gravels (32 ± 3 m amsl and 50 ± 5 m amsl; Figure 4E WIDs 63 and 64; Ota et al., 1984). The Kemps Hill terraces have not been subject to direct geochronological analysis and indicators for these terraces are not identified. Ota et al. (1984) includes over a dozen stratigraphic sections for the terraces; however, specific measurements above sea level are not provided and description in text can differ from the drawn section (e.g. thickness of the deposit and height above sea level) introducing uncertainty.

Furthermore, the surface that underlies the terraces undulates by tens of meters. Only two marine-limiting points (WIDs 63 and 64) for the Amuri Bluff Terrace are included from this publication.

Approximately 20 km to the northeast of Haumuri Bluff, the Kaikoura Peninsula forms a prominent headland. Suggate (1965) identified four distinct surfaces, of which the third highest and most extensive surface (170 ft to 200 ft [~52 m to 61 m] altitude), could be traced on the mainland to the west. Ota et al. (1996) identified five marine terraces, in addition to Holocene

surfaces, deformed by numerous folds and faults, which in general cut perpendicular across the peninsula, resulting in a down-tilt to the northwest, parallel to the long axis of the peninsula. The terraces are labelled in decreasing elevation and age as I, II, III, IV, and V. Terrace III is the extensive terrace identified by Suggate (1965). Representative sections of each terrace were drawn from auger cores and AAR analyses of fossil mollusc from Terrace I provided geochronological constraint. A total of seven direct RSL indicators are derived from Terrace I, II, and III, where rounded pebbles and gravel are interpreted as beach

deposits, providing direct indicators except for one marine-limiting point from Terrace III at 58.8 ± 2 m amsl (Figure 4F WID 101). The beach deposits for each terrace were measured: for Terrace I at 89.5 ± 2 m amsl (Figure 4F WID 100); for Terrace II at 61.3 ± 2.0 m amsl, 71.5 ± 2.1 m asml, and 73 ± 2.1 m amsl (Figure 4F WIDs 97, 98, 99); and for Terrace III at 40 ± 2.1

m amsl and 51 ± 2.1 m amsl (Figure 4F WIDs 95 and 96). Ota et al. (1996) analyzed the same mollusc species for AAR from Terrace I as from the Tarapuhi Terrace at Haumuri Bluff, *Tawera spissa*, allowing for direct comparison of data. The age derived from Terrace I of 110 ± 20 ka and similar dominance of cool-water species within the fossil assemblage, prompted the correlation of Terrace 1 with MIS 5c and the Tarapuhi Terrace (Table 1). Terrace II was correlated with sea-level fall into MIS 5b and Terrace III, the most expansive terrace, with MIS 5a. The remaining Terraces IV and V were considered younger than MIS 5. The age estimate for the remaining terraces were determined by assuming constant rates of uplift and correlation with the Haumuri Bluff Terraces; however, the complexity of the tectonics in the intervening distance recommends against comparing the two locations (Duffy, 2020).

The two remaining marine terrace sequences on the northeast South Island coast are located between Clarence River and Woodbank Stream in north Canterbury, and Long Point and Boo Boo Stream in the Marlborough Region (Ota et al., 1996). The main marine terrace within these coastal sections is designated "MM". Below MM, at lower elevation is a later Holocene terrace. Minor remnants of an upper terrace, "UM", above MM, are present at both locations, but marine sediments were not identified within the Marlborough section and a marine origin for the UM Terrace there cannot be confirmed. Between the Clarence River and Woodbank Stream, the MM Terrace is covered by thick (often >10 m) slope-wash deposits. The elevation of the underlying basal platform decreases from 143 m amsl in the south to 105 m amsl in the north along a distance of 5 km, indicating significant down-tilting to the north. Detailed stratigraphic descriptions were derived from five exposures of the terrace sediments (Ota et al., 1996), allowing for the identification of multiple direct sea level indicators at elevations, from south to north, of 144 ± 2.2 m amsl, 117 ± 3.6 m amsl, 112 ± 2.2 m amsl, 105 ± 2 m amsl, 107 ± 2.8 m amsl (Figure 4E WIDs 94 to 90). Beach deposits are identified in each exposure, inclusive of bored boulders abutting a bedrock cliff at Location 10, indicating exact shoreline position. Four of the five exposures show the beach deposits directly overlying the basal platform, but at Location 7 (WID 91), the beach deposits are found overlying estuarine silt, indicating that the marine terrace sediments record a transgressive rise in sea level.

Between Long Point and Boo Boo Stream, the surface of the MM terrace is well defined with the best preservation to the north and increasing dissection by streams to the south (Ota et al., 1996). The elevation of the inner margin is measured between 55 and 80 m amsl, with the highest elevations at either end of the terrace. The marine sediments and terrestrial cover beds of the MM Terrace are exposed along the former sea cliff between it and the younger Holocene terrace and four direct RSL indicators were identified at elevations, from south to north, of 72.5 ± 2.5 m amsl, 45.5 ± 2.5 m amsl, 42.5 ± 3.2 m amsl, and 51.5 ± 3.2 m amsl (Figure 4H WIDs 89 to 86). The well-stratified beach deposits, composed of well-rounded pebbles within a coarse, sandy matrix, are typically about 3 m thick overlying the basal platform. The surface elevation of the beach sediments varies between 45 m to 74 m amsl. The non-marine cover beds vary in thickness between 4 m and 11 m and are inclusive of multiple paleosols within loess. At the most representative section, Location 3 (WID 88), 11 m of non-marine cover beds includes at least 4 loess units with three paleosols.

Ota et al. (1996) correlated the MM Terrace from both regions with MIS 5e based upon the size of the terrace, the well-defined beach sediments, and position above earlier Holocene terraces. To the south, between the Clarence River and Woodbank Stream, a fluvial terrace merges with the MM Terrace. A TL age from loess overlying the fluvial terrace provides minimum age constraint in support of the MIS 5e designation. Ota et al. (1996) also argue that the number of loess layers (typically 3 to 4) overlying the MM terrace is consistent with loess stratigraphy overlying other last interglacial deposits in the northern South Island and southern North Island. The UM Terrace is correlated to MIS 7 due to its higher elevation.

### 4.3    Summary of New Zealand RSL indicators

This work identified and, using the WALIS database, standardized 77 unique RSL indicators (direct, marine- or terrestrial-limiting points) along the coastline of the North and South Islands of New Zealand (http://doi.org/10.5281/zenodo.4590188

Ryan et al., 2020a). The slim majority (39) are direct indicators of sea level, with the remaining terrestrial-limiting (25) and marine-limiting (13) points. Direct indicators of sea level were identified from elevation measurements of beach deposits or marine terrace shoreline angles. All but one of the terrestrial-limiting points are derived from the below surface Bromley Formation record described by Brown et al. (1988) from well cores within the Canterbury Region. Marine-limiting points are derived from basal platform elevations or poorly described marine sediments from which proximity to the coastline could not be determined. The majority (39) of RSL indicators are correlated broadly with MIS 5 (*sensu lato*). The remainder are correlated with the interglacial peak MIS 5e (18), warm interstadials MIS 5c (7) and MIS 5a (10), and indicators from one marine terrace on the Kaikoura Peninsula were correlated with sea-level fall from MIS 5c into MIS 5b (3). The most common methods for age determination are terrace correlation, luminescence methods, and amino acid racemization.

The North Island contains eighteen RSL indicators, of which the majority are direct indicators of sea level (11) correlating broadly with MIS 5 (9) (Figure 3E). Five RSL indicators (3 direct, 2 marine-limiting) have been correlated with MIS 5e. The only indicator not likely to have been displaced by VLM is the One Tree Point direct sea-level indicator in the Northland Region with most other indicators identified in regions of well-recognized tectonic deformation associated with the Hikurangi Margin and Whanganui Basin (Figure 2). The Otamaroa and Te Papa Terraces on the Raukumara Peninsula are most likely MIS 5e and MIS 5a in age, in contrast to the luminescence results within Wilson et al (2007). We agree with the reasoning of Wilson et al. (2007) that although the peninsula is uplifting, it is unlikely that the MIS 5a and MIS 3 record would be preserved and the MIS 5e would not. The IRSL luminescence method at the time was not fully developed and the results most likely represent minimum ages (Section 5.3.4). On the west coast, the Shelly Beach Formation, younger undifferentiated Hawera sediments, and correlated Waiau A and B Formations and northern Taranaki marine terraces need reassessment. These formations and terraces would likely be sources of multiple RSL indicators, but have not been described in detail since Chappell (1975) and have not been analyzed by any absolute geochronological method. Reassessment of the age of these features is important because the use of terrace correlation by Chappell (1975) to determine age introduces uncertainty in the chronology of the region.

The South Island contains fifty-nine RSL indicators, of which the majority are direct indicators of sea level (39) correlating broadly with MIS 5 (30) (Figure 4I). Only twelve of the fifty-nine South Island RSL indicators are located outside of the Canterbury-Marlborough Regions. Thirteen RSL indicators (all direct) have been correlated with MIS 5e; three of which are located within the Otago Region. Except for those located within the Otago Region, all indicators have been identified in locations subject to significant coastal deformation. Of the four Otago locations, at least two (Taieri Beach and Blueskin Bay) appear to have been subjected to tectonic displacement since deposition. Either the section of coastline within Otago is not as stable as previously presumed, or care must be taken to ensure future field locations are not located adjacent to minor faults. The indicators from the West Coast (4) and Southland (3) regions are broadly correlated with MIS 5; although most of the West Coast and Southland indicators have been analyzed by either luminescence or stable isotope methods, there are concerns for the validity of the results (Section 5.3.4). Similarly, other than the recent reassessment by Oakley et al. (2017), age constraint for sediments and marine terraces in the Canterbury and Marlborough Regions is poor, relying predominantly upon stratigraphic succession or terrace correlations. The minimal preservation of the MIS 5e terrace and variable preservation of MIS 5c and MIS 5a in the coastal sections of the northern Canterbury Region is attributed to wave erosion and fluvial dissection by Oakley et al. (2017).

The varying elevation of the RSL indicators (Figure 3E and 4I) illustrates the role of tectonics in shaping the New Zealand coastline. For example, the transition from the subsiding landscape of the Canterbury Plains to an uplifting one in the north is distinctly marked by the elevation of RSL indicators. The substantial imprint of tectonics makes difficult the development of a sea-level record that can be resolved to modern sea level (Section 5.2) and underlines the historical tendency to use these records for determining VLM rates but not a record of sea level.

In summary, in agreement with Gage (1953) and Pillans (1990a), the New Zealand record of last interglacial sea level lacks quality description, measurement, and age constraint even amongst those records produced since 1990. The following section will provide in greater detail the sources of uncertainty. However, the literature indicates robust relative sea-level indicators are present, not only for MIS 5e, but also the following MIS 5c and 5a interstadials. Better constraint on these records would assist in the understanding of sea-level fluctuation throughout MIS 5 (*sensu lato*). Furthermore, there is potential for the development of a regionally-specific New Zealand sea-level record for MIS 5e from the Northland Regions and SE South Island; although care must be taken within the latter region to ensure no influence from minor tectonic movement. Such a record would not only improve estimates of VLM around New Zealand by providing greater accuracy of peak highstand elevation during MIS 5e, but also benefit global studies of sea-level and ice-volume change.

## 5    Sources of uncertainty

This section provides a summary of the issues contributing the greatest amount of uncertainty to the New Zealand marine terrace records: inconsistent terrace terminology and nomenclature, coastal deformation, and methods of geochronological constraint.

### 5.1    Terrace terminology and nomenclature

The disparate application of terminology and nomenclature used when discussing marine terraces in New Zealand contributed to uncertainty in interpreting the meaning of potential sea-level indicators within our database. The example used by Pillans (1990a, pg. 221) in his argument for the use of standardized terminology for marine terraces was the prevalence for authors to refer to a terrace by altitude, e.g. the "100 metre terrace". The questions being: which feature of the terrace is at 100 metre and at what position within the terrace? The work of numerous authors over the decades has also contributed to an inconsistency in terrace nomenclature. For example, in the Horowhenua lowlands where the Tokomaru Marine Terrace designation refers to the basal platform, the inner margin fossil marine cliff, and the terrace surface but the cover beds are identified as the Otaki Formation (Section 4.1.6).

Uncertainty is also introduced through age correlation (in absence of a numerical age) to either distant marine terrace sequences or local stage names. The early 20th Century practice of correlating marine terrace sequences to Mediterranean stages (Monastirian, Tyrrhenian, Milazzian, Sicilian), which at the time also lacked numerical age and have since been redefined, precludes the correlation of any potential RSL indicator to MIS 5 with certainty. The youngest formally accepted Pleistocene stage within New Zealand is the Haweran Stage, which encompasses the past 0.340 Ma (Raine et al., 2015). The stage boundary is identified at the base of the Rangitawa Tephra - the oldest bed of the Hamilton Ash Formation (Section 5.3.2). Local stage names for interglacial/glacial periods within the Pleistocene have been proposed (e.g. Suggate, 1965; 1985), but these were often developed from local sediment sequences without numerical age constraint and have not been formally adopted.

Within our database we have endeavored to adapt any inconsistent marine terrace terminology to that recommended by Pillans (1990a; Figure 1) and provide clarity for terrace nomenclature. To avoid confusion and to assist in understanding the global context of the New Zealand sea-level record, we have chosen not to use local New Zealand stage names for the Pleistocene and have given preference for marine oxygen-isotope stages (MIS) designations, an internationally recognized scale.

### 5.2    Coastal deformation and GIA

Excessive and prevalent coastal deformation will preclude the development a sea-level reconstruction that can be registered to present day, regardless of the quality of sea-level indicators. The position of the New Zealand archipelago straddling the active boundary of the Australian and Pacific plates has produced a coastline subject to variable rates of vertical land movement (VLM) due to complex tectonics and displacement associated with earthquakes. As has been shown above, the New Zealand

paleo-shorelines and marine terraces (and below-surface marine deposits) have been essential for determining estimates of long-term (beyond Holocene) rates of uplift or subsidence (e.g. Chappell, 1975; Pillans, 1983; Bishop, 1985; Suggate, 1992; Berryman, 1993; Begg et al., 2004; Wilson et al., 2007; Beavan and Litchfield, 2012; Oakley et al., 2018). However, because the research focus has been on determining long-term VLM rates, and due to a lack of adequate geochronological methods for many studies, potential sea-level indicators have been described in less detail than desired for such use. Although any sea-level reconstruction derived from these indicators may not be useful for a relative sea-level curve with relation to present sea level, more precise descriptions and age constraint can improve estimates of VLM rates, especially where there is uncertainty in correlation to the appropriate MIS 5 highstand. Furthermore, where high rates of uplift have produced marine terrace sequences recording multiple substages of MIS 5, there is opportunity to better constrain not only regional sea-level fluctuations within MIS 5 but paleoenvironmental change as well.

GIA encompasses all solid Earth deformations, as well as gravitational, and rotational-induced changes of the mean sea surface in response to the buildup and retreat of ice sheets, with residual and variable affect along coastal sections depending upon their proximity to former glaciers, ice caps and sheets (Arctic and Antarctic, Simms et al., 2016). In other words, the magnitude of the solid Earth response to ice-and water-load history depends on the geographical location and varies with time (in relation to glacial maxima) because of the viscous Mantle flow, thus producing a time-dependent regional gradient in relative sea level that is modulated by mantle rheology. Neglecting GIA on active coastlines when determining rates of VLM has been shown to lead to overestimated uplift rates at an overage of 40%, but also up to 72% (Simms et al., 2016; Stocchi et al., 2018).

New Zealand sits on a 'sweet spot' with respect to Antarctica, such that when the northern hemisphere ice sheets are neglected, the local RSL response to either growth or retreat of the Antarctic ice Sheet (AIS) is nearly eustatic. However, deviations from eustatic may increase dependent upon ice mass fluctuations within specific sectors of the AIS. In particular, melting from the east AIS, which is closer to New Zealand, would shift the eustatic band crossing the North and South Islands northward, above the North Island and cause a lower-than-eustatic local sea-level rise. Various scenarios of Antarctic ice geometries indicate New Zealand RSL approximates eustatic sea level with GIA having little effect (~2-3 m deviations from eustatic) on New Zealand (Grant et al., 2019), thus making it useful for constraining global ice-volumes during MIS 5e. The predicted deviations of RSL from the eustatic during MIS 5e are partly due to ocean syphoning and continental levering. The former causes local (New Zealand) sea-level drop in response to water flow towards the subsiding peripheral forebulges that surround the glaciated areas. The latter causes relative sea-level drop in response to local crustal uplift as a consequence of water-loading-induced crustal tilt. Hence, both processes result in a New Zealand highstand 1-3 kyr earlier that eustatic, which is then followed by a RSL drop (Figure 5). The combined effect of ocean syphoning and continental levering may explain the variability of Holocene sea-level change around New Zealand. For example, the Holocene highstand peaked in the North Island at ~2.65 m apsl between 8.1 to 7.2 cal ka BP, whereas in the South Island, the highstand peaked later, between 7.0 and 6.4 cal ka BP, at no more than ~2 m apsl (Clement et al., 2016).

**Figure 5: Local RSL curves for MIS 5e sea level at the northernmost tip of North Island (Te Hapua; latitude -34.39, longitude 173.02), Auckland (latitude -36.85, longitude 174.76), and the southernmost tip of South Island (Coal Island; latitude -46.21, longitude 166.66) generated from ANICE-SELEN and ICE-6G models. Similar to the Holocene, sea level peaks earlier and higher in the North Island. The Northland region (Te Hapua) RSL curve is nearest to eustatic. Deviations of the RSL curves from eustatic within the models is driven by ocean syphoning, suggesting it serves as a primary driver of variability in the timing and height of peak sea level across New Zealand. Note the different scale to x- and y- axes between model outputs.**

The significance of New Zealand glacier ice-volume change on coastal deformation and sea-level reconstructions has not been quantified for New Zealand (King et al., 2020). Resolving any flexure within New Zealand beyond the Holocene, due to regional glacier ice-volume change or any other regional drivers, is unlikely due to the extensive coastal deformation. The gravitational effect of local glaciers would be hard to detect, and similarly with solid Earth deformations. Given the short

wavelength of glaciers, their deformations would be most likely elastic and would therefore be compensated by space-limited upper lithosphere flexure/deformation.

The only last interglacial site identified so far that is most likely to have been unaffected by deformation is One Tree Point in Northland (Section 4.1.1), highlighting the importance of this region for additional study. The apparent stability of portions of southeastern coastline of the South Island (Sections 4.2.1 and 4.2.2) also warrants additional investigation.

## 5.3    Geochronological constraint

Marine terrace ages have been constrained in New Zealand using multiple different approaches. Correlated-age determination of a terrace at a known elevation, either with distant terrace sequences or a marine oxygen-isotope sea-level curve, is the most consistently applied method to determine age. Three additional age constraint methods are core to New Zealand marine terrace chronology: tephrochronology for terraces proximal to volcanic centers in the North Island; amino acid racemization (AAR); and luminescence techniques. The limitations of these methods and the consequent impacts on age constraint and terrace correlation are briefly discussed here.

### 5.3.1    Marine terrace correlation

Muhs (2000) summarized three common global correlation methods that have been used in New Zealand to determine marine terrace age (and subsequently uplift rates): 1) assumption of a constant uplift rate on a shore-normal terrace sequence; 2) relation diagrams for shore-parallel terrace sequences; and 3) unique altitudinal spacing of terraces assuming a constant uplift rate. Methods 2 and 3 were developed within New Zealand, the former was used by Pillans (1983) to assist with age determination of the Whanganui Basin sequence, and the latter by Bull (1985) to determine the age of marine terraces along the Alpine Fault. Major assumptions of each of these methods are that the uplift rate has been constant over time and/or that there has been no gradient of uplift normal to the coastline; although shore-parallel variation in uplift rate is expected in method 2 (Muhs, 2000). All of these methods also suffer from the circular problem of assigning a glacio-eustatic sea level to a terrace to estimate its uplift rate, which is then used to determine age of additional terraces.

Global correlation methods largely rely upon a chosen distant terrace sequence (e.g. Huon Peninsula) for paleo sea level and/or a marine oxygen-isotope curve to assist in determining age (Table 3). However, these records are unlikely to accurately reflect the timing and height of sea-level highstands that formed New Zealand marine terraces. Distant terrace sequences are expected to have a local peak sea level differing in both height and timing due to their own GIA signal and other localized processes (e.g. steric effects, Creveling et al., 2015). Marine oxygen-isotope records represent a convoluted signal of ocean mass and sea temperature variation resulting in a paleo global mean sea-level estimate with considerable uncertainty (Rovere et al., 2016) and can also include their own tectonic uplift correction (Simms et al., 2016).

The importance of chosen sea level record was recently stressed by Duffy (2020) when determining slip rates on the Kaikoura Peninsula of New Zealand. Concerned with the use of a LIG sea-level highstand estimate from a region without the same GIA characteristics, Duffy (2020) expressed preference for an MIS 5e highstand estimate of +2.1 ± 0.5 m that was derived from RSL indicators identified on the tectonically stable, far-field Gawler Craton in South Australia (Murray-Wallace, 2002; Murray-Wallace et al., 2016). This record was preferred because southern Australia and the South Island lie within 5° of latitude, a comparable distance from the Antarctic ice cap peripheral bulge, and because Australia and New Zealand display similar records of the mid-Holocene highstand and subsequent sea-level fall (Sloss et al., 2007; Clement et al., 2016; Duffy, 2020). However, the Australian record can be expected to differ from that of New Zealand because the North Island extends into much lower latitudes and due to South Island ice volume changes (e.g. Golledge et al., 2012; James et al., 2019; Carrivick et al., 2020); the role of Holocene glacial mass on GIA requires further investigation (King et al., 2020). The difference in the timing and amplitude of the Holocene highstand between South and North Island (Clements et al., 2016) makes it is reasonable

to expect a similar difference during the Last Interglacial; although, given the current resolution of available geochronological methods, it is unlikely that difference can be discerned within the LIG or deeper time.

An additional concern is the correlation of paleo-coastlines to the correct MIS. As pointed out by Litchfield and Lian (2004), the correlation of the Taieri Beach to either MIS 5e or MIS 5a has significant implications for derived uplift rates. Furthermore, estimates of the height and timing for each sea-level highstand within MIS 5 (*sensu lato*) has changed, particularly for the interstadials, significantly through time (Table 3).

### 5.3.2 Tephrochronology

On the North Island, two tephras have been used consistently as stratigraphic markers to constrain and identify MIS 5 terraces and associated sediments: the Rotoehu Ash and the Hamilton Ash (Pullar and Grant-Mackie, 1972; Chappell, 1975; Pain, 1976; Iso et al., 1982; Ota et al., 1989b; Berryman, 1993; Wilson et al., 2007; Claessens et al., 2009). The Rotoehu Ash, the basal member of the Rotoiti Tephra Formation, erupted from the Taupo Volcanic Zone and serves as an important regional stratigraphic marker in the North Island. Numerous age estimates, ranging from c. 61 ka to 45 ka, have been published (see comprehensive summary by Flude and Storey, 2016), but the most recently published age estimates for the Rotoehu Ash (derived from a combination of $^{14}$C-acclerator mass spectrometry, (U-Th)/He, and $^{40}$Ar/$^{39}$Ar geochronological methods) place deposition at 47.5 ± 2.1 ka (Danišík et al., 2012; Flude and Storey, 2016). The age of the Rotoehu Ash, although well-constrained, does only provide minimum age constraint to Pleistocene marine terraces.

The Hamilton Ash Formation has been described as a sequence of time transgressive ash beds (H1 to H7) (Ward, 1967; Vucetich and Pullar, 1969; Iso et al., 1982; Lowe et al., 2001). Recently (Lowe, 2019), it has been proposed that the sequence is more accurately described as a, "composite set of clayey, welded paleosols very probably developed by upbuilding pedogenesis from MIS 10 to 5" (p. 23 of 24). Alternatively, Briggs et al. (2006) found the designation as a Formation inappropriate given that some of the tephra beds could be found preserved as individual layers in the Maketu area – near to the locus of Mid-Pleistocene TVZ activity, the likely source of the tephras. Where both are present, the Hamilton Ash sequence is overlain by the Rotoehu Ash (Ward, 1967; Vucetich and Pullar, 1969; Iso et al., 1982; Lowe et al., 2001). The Hamilton Ash has been used to constrain marine terrace development or associated dune sand deposition with the understanding that the youngest tephra bed dated to ~MIS 6/5 (Pullar and Grant-Mackie, 1972; Chappell, 1975; Pain, 1976; Iso et al., 1982; Ota et al., 1989b). Only the basal ash bed, the Rangitawa Tephra (H1), has been directly dated using radiometric methods to c. 400 ka to 340 ka (Kohn et al., 1992; Pillans et al., 1996; Lowe et al., 2001). Age constraint for the overlying ashes is based upon the physical characteristics, clay mineral assemblages, and climatostratigraphic associations of the deposits and their paleosols (Lowe, 2019; Lowe et al., 2001). The uppermost Hamilton Ash, the Tikotiko Ash (H6/H7), was likely deposited within MIS 6 or MIS 5e and subject to pedogenic alteration throughout MIS 5. A minimum age for the surface has been estimated at c. 74 ka; however, the only other chronostratigraphic constraint for the H2 through H7 ash beds is that they are older than the Rotoehu Ash (Lowe et al., 2001; Lowe, 2019). The possible 74 ka age for the H6/H7 beds does have implication for marine terrace chronology, where the beds have been used to argue a minimum MIS 5e age. Until the age of the H6/H7 bed is better constrained, it is not recommended for use as a chronostratigraphic marker to discern MIS 5e age.

It should be noted that loess cover bed stratigraphy has also been used to assign relative age to marine terrace sequences in New Zealand, e.g. the Mahia Peninsula (Berryman, 1993). However, loess stratigraphy, as described in the publications reviewed, is generally localized and lacking in regional (or greater) spatial correlations. Furthermore, without numerical age constraint to the sedimentation history of the location, the resolution of such an approach is questionable (Muhs, 2000).

### 5.3.3 Amino Acid Racemization (AAR)

Amino acid racemization has been applied to both wood and marine mollusc shell to produce numerical ages for marine terraces in New Zealand (http://doi.org/10.5281/zenodo.4590188 Ryan et al., 2020a). The first numerical age constraint was derived from alloisoleucine-isoleucine D/L values of fossil wood fragments overlying marine terraces of the Whanganui Basin terrace sequence analyzed on a modified Technicon amino acid auto-analyaer. The fossil wood D/L values were calibrated with the Rangitawa Tephra, which overlies numerous mid-Pleistocene marine terraces, the youngest of which is the Ararata Terrace (Pillans, 1983; Pillans, 1990b; Pillans and Kohn, 1981; Pillans et al., 1996). The age of the tephra, at c. 370 ka, was determined by fission-track dating of both zircon and glass components, and used to constrain the age of the Ararata Terrace to c. 400 ka, MIS 11. Calibration of the fossil wood retrieved from the Ararata Terrace lignite cover bed, closely underlying the Rangitawa Tephra, allowed numerical ages to be derived from the D/L values of fossil wood samples from the younger Rapanui Terrace using the integrated rate equation.

The minimum age derived for the Rapanui Terrace was 110 ka and the terrace was correlated with the MIS 5e transgression cycle, culminating at c. 120 ka with peak sea level between 5 and 8 m apsl (Chappell and Veeh, 1978; Pillans, 1983). An uplift model, developed using the Ararata and Rapanui Terraces as anchors, was used to calculate the ages for the other marine terraces, which were correlated with every interglacial (odd MIS) from MIS 17 to MIS 3. The only marine isotope stage to have multiple sea-level peaks represented is MIS 5, the Rapanui Terrace (MIS 5e, 120 ka), the Inaha Terrace (MIS 5c, 100 ka), and the Hauriri Terrace (MIS 5a, 80 ka) (Section 4.1.2). This geochronological framework has not only served as a basis for determining regional uplift rates of the marginal Whanganui Basin in the mid-late Quaternary, but also underpins many of the marine terrace age correlations within New Zealand since (Bishop, 1985; Ward, 1988a; Bussell, 1990; Pillans, 1990a; Ota et al., 1996). Ota et al. (1996) used unpublished AAR alloisoleucine-isoleucine D/L values of *Tawera* shell from the Hauriri Terrace (80 ka) to calculate numerical ages from alloisoleucine-isoleucine D/L values (determined using an automated amino acid analyzer) of *Tawera spissa* (*T. spissa*) shell collected on the Kaikoura Peninsula and Haumuri Bluffs (Section 4.2.3). The numerical ages were calculated using the integrated rate equation and the Arrhenius equation to allow for the difference in long-term temperature history between the regions.

The aminostratigraphy of the Whanganui sequence was redefined by Bowen et al. (1998) using shell of the marine bivalves *T. spissa* and *Austrovenus stutchburyi*, analyzed by ion-exchange high-performance liquid chromatography (HPLC) to provide alloisoleucine-isoleucine D/L values. The D/L values were constrained by the previous fission-track ages and also biostratigraphy, magnetostratigraphy, and correlated with marine $\delta^{18}O$ isotope stages (Beu and Edwards, 1984; Beu et al., 1987; Shackleton et al., 1990; Bassinot et al., 1994; Pillans et al., 1994). The reassessment by Bowen et al. (1998), although not resulting in any change in MIS correlation, provided more direct constraint for the marine terraces as the previous analysis by Pillans (1983) was of wood fragments of unknown genera in lignite beds overlying marine sediments. Furthermore, there are methodological concerns regarding the application of AAR to wood that have never been resolved; namely the kinetics of racemization within wood and the presence of internal sugars (e.g. arabinose), which interact with amino acids to form melanoidin polymers (Zumberge et al., 1980; Blunt et al., 1987; Rutter and Vlahos, 1988). Additional causes for concern are wood degradation and the seemingly increased sensitivity of wood to internal and external environmental factors.

The aminostratigraphy developed by Bowen et al (1998) does not appear to have as broad application to determining marine terrace age as the former framework developed by Pillans (1983). However, recently, Oakley et al. (2017) collected additional shell samples from sites used by Bowen et al. (1998), as well as other sites of known age (MISs 1, 5a, 5e, 7, 11, 17), to provide multiple calibration points for AAR analyses of *T. spissa* shell from marine terraces in the North Canterbury region. The different pretreatment methods and machine types used in each study (ion-exchange HPLC by Bowen et al (1998) vs. reverse-phase HPLC by Oakley et al. (2017)) to determine D/L values, and preference for different amino acids in developing

chronologies, meant that results from the studies were not directly comparable. Oakley et al. (2017) also utilized a new approach of Bayesian statistical methods (Allen et al., 2013) to determine the best fit AAR age equation and calculate uncertainties. The best fitting function for relating sample age to D/L value was determined to be the simple power-law kinetics where an exponent is applied to either the D/L value or *t*, time (Goodfriend et al., 1995), with aspartic acid D/L values used for calibration (Oakley et al., 2017). These results were used in combination with infrared stimulated luminescence to develop a new chronology for North Canterbury marine terrace development, resulting in significantly different chronologies in some locations than earlier studies where elevation or degree of fluvial dissection of the marine terrace had been used to assess age (e.g. Carr, 1970; Yousif, 1987).

AAR has also been applied to fossil shell of the warm-water estuarine bivalve *Anadara trapezia* found in multiple locations around North Island associated with interglacials ranging in age from MIS 11 through MIS 5e (Beu and Maxwell, 1990; Murray-Wallace et al., 2000). Currently extinct in New Zealand, almost all fossil *A. trapezia* correlated with MIS 5e and MIS 7 were restricted to the northeast, with exceptions at Gisborne and Mahia Peninsula dating to MIS 5e, and southern Hawke's Bay to MIS 7. The only known samples correlated with MIS 11 (on the basis of stratigraphic evidence) are found within the Whanganui Basin and are the only *A. trapezia* identified in the southwest. The geographic distribution of fossil *A. trapezia*, increasingly restricted to the northeast with successive interglacials, probably reflects the poor fossil record of estuarine fauna, and decreasing extent of warm-waters during interglacials as a consequence of geographic changes to the New Zealand landmass from progressive uplift (Murray-Wallace et al., 2000).

In summary, although early age constraint for the Whanganui Basin marine terraces was derived from AAR analysis of wood fragments (Pillans, 1983), an unreliable method, AAR of mollusc shell has proven useful in building an aminostratigraphic framework in New Zealand and for discerning shell of different interglacial age; e.g. MIS 5 vs MIS 7 (Ota et al., 1996; Bowen et al., 1998; Murray-Wallace et al., 2000; Oakley et al., 2017). It has been shown to be complimentary to luminescence methods, providing more certainty where results converge, with the ability to assist in resolving discrepant results in the latter method (Oakley et al., 2017). The relatively new (Allen et al., 2013) Bayesian method to determine numerical age from D/L values has been proven successful when applied to a New Zealand Pleistocene dataset (Oakley et al., 2017). The applicability of AAR to mollusc and foraminifer tests for resolving relative age at Pleistocene timescales, identifying reworked contributions to deposits, and developing useful aminostratigraphic frameworks, as well as its complimentary nature to luminescence techniques, has been proven in numerous locations globally (Hearty et al., 1992; 2004; Murray-Wallace and Belperio, 1994; Murray-Wallace et al., 2010; Wehmiller et al., 1995; Wehmiller, 2013; Kaufman et al., 2013; Ryan et al., 2020b). These capabilities of AAR, and its relatively low cost (in comparison to other geochronological methods), recommend it for continued and future use resolving geochronology in New Zealand.

### 5.3.4 Luminescence

Both thermoluminescence (TL) and infrared stimulated luminescence (IRSL) have been used in the dating of New Zealand last interglacial sediments; predominantly of multi-grain aliquots of fine polymineral silts (4-11 μm), and rarely, coarser sand-size grains of quartz and feldspars (http://doi.org/10.5281/zenodo.4590188 Ryan et al., 2020a). The application of luminescence dating to New Zealand quartz grains has been found to be unsuitable because the quartz grains generally have a very dim optically stimulated luminescence (OSL) signal intensity, are adversely affected by thermal transfer of charge between single-aliquot regenerative-dose (SAR) measurement cycles, and display unpredictable sensitivity changes across these cycles (Preusser et al., 2005; 2006). These characteristics are, at least in part, attributable to the young sedimentary history of the quartz grains, but may also be a characteristic of the geological provenance of these grains influencing their intrinsic brightness (Preusser et al., 2006). Even if these hurdles could be overcome, the relatively high dose rates in New Zealand sediments (e.g. in comparison to Australia) would more than likely lead to the saturation of the quartz OSL signal

well within 100 ka (Roberts et al., 2015), meaning OSL analysis of quartz sediments dating to the last interglacial would more than likely result in minimum ages only. These numerous issues effectively remove quartz OSL from the chronological toolkit for dating Last Interglacial sediments in New Zealand.

Greater success in dating LIG sediments may be gained using K-feldspar grains; although these too have impediments. The primary concern is 'anomalous fading' (Wintle, 1973), which is the leak of electrons from unstable traps in the grains over the period of burial resulting in underestimated ages. Significant developments in the procedures used for measuring K-feldspar grains allow for this fading to be either reduced to negligible levels or accounted for (Thiel et al., 2011; Buylaert et al., 2012; Rui et al., 2019). However, the stability of the non-fading component of the IRSL signal results in a much slower IRSL decay
rate (Buylaert et al., 2012; Li et al., 2013; Smedley et al., 2015). Thus, a longer period of sunlight exposure is required to fully reset the signal and/or a residual dose measurement must be made to account for this before age determination. Alternatively, the measurement of single-grains of K-feldspar (rather than multi-grain polyminerals) would enable the adequacy of the previous resetting to be tested, which the use of multi-grain aliquots cannot as reliably disentangle (Jacobs and Roberts, 2007). Finally, unlike quartz, the onset of saturation of K-feldspars occurs at much higher doses enabling the accurate dating of LIG
and older sediments (Aitken, 1998; Huntley and Lamonthe, 2001; Li et al., 2014).

The lack of critical information regarding how the luminescence signal/s was measured in the studies reviewed here does not instill confidence in the reported ages. Given the methodological advances in luminescence measurement procedures since many of the studies reviewed here were completed, with the exception of the recent work by Oakley et al. (2017), the ages presented above represent, at best, a starting point. This uncertainty in absolute age of the associated marine terrace sequences
and correlation with the appropriate sea-level peak within MIS 5 has significant implications for estimates of uplift rates in some locations (e.g. Taieri Beach, Section 4.2.2). Advances in method, particularly in using the elevated temperature post-infrared IRSL (pIRIR), have been shown to be effective in overcoming the fading issues in very young New Zealand coastal sediments (Madsen et al., 2011) and indicate the ability to produce ages with a higher degree of accuracy. The use of single feldspar grains, instead of polymineral aliquots, has the potential to assist in the identification and removal of grains that are
'poorly behaved' prior to age estimation, identify whether or not the deposits contain a reworked component, and the extent to which the previous resetting of electron traps was completed (Jacobs and Roberts, 2007). The applicability of the single grain method depends upon the grains having a sufficiently bright signal for accurate equivalent dose measurement – New Zealand quartz is often reported as rather dim, K-feldspars as much brighter. We therefore recommend a complete and systematic reassessment of all previously reported ages. Furthermore, in future all luminescence data (e.g. individual dose rate
components, fading factors, equivalent dose distribution patterns and associated statistics, and age model selection) needs to be reported completely. The WALIS database interface allows for the reporting and archiving of all critical luminescence data and full recommendation list is provided in the Supplementary Materials. Such transparency will enable future reviewers to make a much more informed judgement about the quality of the data.

## 6     Further details: Pleistocene and Holocene sea-level fluctuations and climatic change

The Whanganui Basin, in addition to the Rapanui (MIS 5e), Inaha (MIS 5c), and Hauriri (MIS 5a) terraces (Section 4.1.2), retains a sequence of shallow marine transgressive, highstand, and regressive sediments correlated with each high sea-level marine oxygen isotope stage of the past 2.6 Ma, reflecting cyclic, orbitally-paced eustatic sea-level fluctuation (Pillans, 1991; Beu et al., 2004; Pillans et al., 2005; 2017; Naish et al., 1998; Grant et al., 2019). This record is extremely well-preserved and has been subject to an extensive variety of geochronological and stratigraphical methods. Not only does it offer a detailed
paleoenvironmental record from an isolated part of the South Pacific, it serves as a paleo-proxy for the amplitude of interglacial-glacial relative sea-level change and constrains polar ice-volume variability within the Pliocene (3.30 to 2.50 Ma) when atmospheric carbon dioxide concentration was last ~400 parts per million – a climatic condition recently met. It retains

a long record of tephra (to c. 2.17 Ma) and loess (to c. 0.50 Ma) deposition, which provides a framework for regional stratigraphic correlation in the North Island. The stratotype sections and points of the four stages representing Quaternary New Zealand are defined by the fossiliferous marine sediments within the Whanganui Basin. A paleovegetation and paleoclimatic record spanning much of the Haweran Stage (0.340 Ma to present) has been developed from the marine and terrestrial sequence. Unfortunately, the ongoing and complex tectonics of the North Island preclude any sea-level reconstruction registered to present day – indeed it is the relatively recent uplift of the basin margins which has allowed preservation of marine terraces formed over the past 0.7 Ma, including the Rapanui, Inaha, and Hauriri terraces. Although the Whanganui Basin sequence is surely the longest and most complete record reflecting global sea-level fluctuations and climatic change within New Zealand, it is not the only one to extend beyond MIS 5. The vertical land movement along several sections of the New Zealand coastline has allowed for the preservation of paleo records of marine, coastal and terrestrial environments extending not only to the present but also farther into the Pleistocene; e.g. Mahia Peninsula, North Island (Berryman, 1993) and the Fiordland, South Island (Bishop, 1985; Ward, 1988a). The sediments preserved along the New Zealand coastline have proven valuable sources of proxy data (commonly in the form of fossil marine mollusc assemblages or fossil pollen) useful for biostratigraphy and paleoclimatic reconstructions (e.g. Grant-Mackie and Scarlett, 1973; Dickson et al., 1974; Moar, 1975; McGlone et al., 1984; Mildenhall, 1985; 1995; Moar and Mildenhall, 1988; Bussell, 1990; 1992; Berryman, 1992; 1993; Ota et al., 1996; Shulmeister et al., 1999; Murray-Wallace et al., 2000). However, lacking the scrutiny of the Whanganui Basin, any relative sea-level record (or associated climatic record) derived from older Pleistocene marine terraces and sediments is likely to suffer the similar problems of insufficient description as the MIS 5 record, leading to large uncertainty in age and interpretation.

A recent study (Clement et al., 2016) found spatial and temporal variation in New Zealand's Holocene relative sea-level change may be influenced by a number of different mechanisms. A north-south gradient in RSL may be a result of the position of the archipelago within the intermediate field around Antarctica across broad degrees of latitude. Continental levering could have a significant effect on the timing and magnitude of sea-level change at a regional to local scale as driven by glacial meltwater loading and width of the adjacent continental shelf. Sea-surface height relative to the ellipsoid is also variable around New Zealand (King et al., 2020). Potentially significant drivers of relative sea-level change are regional and local effects of tectonic regime, wave climate, and sediment regime – all of which require further research to characterize (Clement et al., 2016).

## 7    Future research directions

Producing an accurate and precise reconstruction of last interglacial sea level in New Zealand is made difficult by challenges stemming from extensive tectonism, incomplete information (e.g., elevation data, descriptions of indicative meaning), and age constraint. A New Zealand regional sea-level curve, derived from Northland and the southeastern South Island, would provide a valuable record within the remote South Pacific, which could assist in understanding the eustatic sea-level response to ice mass change in Antarctica and would allow for better assessment of coastal deformation and improved estimates of long-term vertical land movement around New Zealand. In recognition that most RSL indicators are poorly described, not related to a defined sea-level datum, lack numerical age constraint, and that most existing luminescence-derived numerical ages were derived from outdated methods, greater accuracy would also require reassessment of nearly all RSL indicators identified to present. The apparent extensive preservation of MIS 5e, 5c, and 5a marine and coastal sediments and the terrestrial sediments of intervening MIS 5d and MIS 5b substages provides opportunity for study of sea-level fluctuations and climatic and paleoenvironmental changes throughout MIS 5 (*sensu lato*) as well as changes to long-term rates of vertical land movement.

## 8    Data Availability

The database is available open access and kept updated as necessary at the following link: http://doi.org/10.5281/zenodo.4590188 (Ryan et al., 2020a). The files at this link were exported from the WALIS database

interface on 23 September 2020. Description of each data field in the database is contained at this link: https://doi.org/10.5281/zenodo.3961543 (Rovere et al., 2020), that is readily accessible and searchable here: https://walis-help.readthedocs.io/en/latest/. More information on the World Atlas of Last Interglacial Shorelines can be found here: https://warmcoasts.eu/world-atlas.html. Users of our database are encouraged to cite the original sources in addition to our database and this article.

## 9    Author Contributions

D.D.R. was primary author, responsible for all entries into WALIS, and the conceptualization, development and writing of manuscript, as well as providing expert review of amino acid racemization data. A.C. contributed to the structure and writing of the manuscript, and assisted D.D.R. in developing figures. N.R.J. provided expert review of luminescence data, contributing significantly to that section of the manuscript as well as more minor contributions elsewhere. P.S. contributed significantly to the discussion of coastal deformation and GIA in New Zealand.

## 10    Competing Interests

The authors declare that they have no conflict of interest.

## 11    Acknowledgements

The data presented in this publication were compiled in WALIS, a sea-level database interface, developed with funding from the ERC Starting Grant "WARMCOASTS" (ERC-StG-802414), in collaboration with PALSEA (PAGES/INQUA) Working Group. The database structure was designed by A. Rovere, D. Ryan, T. Lorscheid, A. Dutton, P. Chutcharavan, D. Brill, N. Jankowski, D. Mueller, M. Bartz. E. Gowan and K. Cohen.

D.D.R. wishes to thank Prof. David Lowe for clarifying aspects of New Zealand tephrochronology and providing useful publications towards that purpose and also Prof. Alessio Rovere for discussions during the development of the manuscript. The authors also thank the reviewers, Prof. Tim Naish and Dr. Nicola Litchfield, whose constructive comments and suggestions served to improve this publication.

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

# Table 1: Changes in nomenclature and age correlation of interglacial sequences in North and South Islands of New Zealand

| Region | North Island | MIS 7 | MIS 5e | MIS 5c | MIS 5a | Post MIS 5 |
|---|---|---|---|---|---|---|
| **Auckland - Northland** | Brothers, 1954 | Shelly Beach Formation (33.5 - 39.6 m) | Waioneke Formation (13.7 - 22.9 m) | Waioneke Formation (4.5- 7.6 m) | | |
| | Chappell, 1975 | | Shelly Beach Formation (40 m) | Waioneke Formation | | |
| | Richardson, 1985 | South Head Formation | Shelly Beach Formation/Terrace (40 m) | Undifferentiated Hawera sediments (24 to 8 m) | | |
| | **Nichol, 2002 | | One Tree Point (prograding beach face 5 +/- 4 m) | | | |
| **Whanganui - Taranaki - Waikato** | **Chappell, 1970 | Waiau A Formation (marine sediments, max. height 39.6 m) | Waiau B Formation (isolated littoral beds, max. height of 12.9 m) | | | |
| | Chappell, 1975 | | Waiau A Formation (marine sediments, max. height 39.6 m) | Waiau B Formation (isolated littoral beds, max. height of 12.9 m) | | |
| | Chappell, 1975 | | NT3 (shore platform 40-42 m) | NT2 (littoral and estuarine beds to 20 m) | | |
| | Pain, 1976 | | | Te Akeake Sands Member (Waiau B Formation) | | |
| | Alloway et al, 2005 | NT3 | NT2 | | NT1 | |
| | Fleming, 1953 | Brunswick Terrace | | Rapanui Terrace | | |
| | Dickson et al, 1974 | Brunswick Formation/Terrace | Ngarino Formation/Terrace | Rapanui Formation / Terrace | | |
| | Chappell, 1975 | | Ngarino Terrace | Rapanui Terrace | | |
| | **Pillans, 1983 | Ngarino Terrace (76-128 m) | Rapanui Terrace (<30-70 m) | Inaha Terrace (24-40 m) | Hauriri Terrace (16 m) | |
| **Gisborne - Bay of Plenty** | Chapman-Smith & Grant-Mackie, 1971 | Otamaroa Terrace; *or younger* (63 m) | Te Papa Terrace; *or younger* (33 m) | | | |
| | Pullar & Grant-Mackie, 1972 | | | Otamaroa Terrace | Te Papa Terrace | |
| | Chappell 1975 | BOP3 (shore platform >50 m) | BOP2 (shore platform >20 m) | | | |
| | Yoshikawa et al, 1980 | Matakaoa Terrace (20-130 m) | Otamaroa Terrace (10-300m) | | Te Papa Terrace (10-215 m) | |
| | Ota et al, 1989 | | Otamaroa Terrace (terrace surface 68-287 m) | Raukokore Terrace (terrace surface minus 4 m for tephra, ≥214 m) | Te Papa Terrace (terrace surface minus 4 m for tephra, 28-92 m) | |
| | **Wilson et al, 2007 | | Otamaroa Terrace (62-279 m); Te Papa Terrace (62-279) | | | |
| **Hawke's Bay** | Cashman and Kelsey, 1990 | | Kidnappers Terrace (100-200 m) | | | |
| | **Berryman, 1993 | Moemoto Terrace (88-230 m) | Mahia III (42-150 m) | Mahia II (51-150 m) | Mahia I (83-116 m) | Auroa 2 Terrace (63-64 m) |
| **Wellington - Wairarapa** | **Mildenhall, 1995 | | Biozone P6 of the Hutt Formation (depth below surface: 82.0-105.4 m decreasing landwards to 48-65 m below surface) | | | |
| | Begg et al, 2004 | | Willford Shellbed | | | |
| | Heine, 1982 | | MIS 5 terrace levels (40-42 m and 18-20 m) | | | |
| | Leamy, 1958 | Tyrrhenian (30.7 - 31.6 m) | Main Monastirian (15.5 - 16.5 m) | Late Monastirian (5.1 m amsl) | | |
| | Cowie, 1961 | | Tokomaru Marine Terrace | | | |
| | **Fleming, 1972 | | Otaki Formation | | | |
| | **Palmer, 1988 | | Tokomaru Marine Terrace & Otaki Formation. Beach sand (~30 m amsl) | | | |
| | Sewell, 1991 | | Tokomaru Marine Terrace & Otaki Formation | Post Tokomaru Marine Terrace 1 | Post Tokomaru Marine Terrace 2 | |

**indicates source of RSL indicator

**Table 1 continued: Changes in nomenclature and age correlation of interglacial sequences in North and South Islands of New Zealand**

| Region | South Island | MIS 7 | MIS 5e | MIS 5c | MIS 5a | Post MIS 5 |
|---|---|---|---|---|---|---|
| West Coast - Southland | Suggate, 1965 | Karoro Formation | Awatuna Formation | | | |
| | Suggate, 1985 | Karoro Formation | Rutherglen Formation | Awaturna Formation | | |
| | **Nathan & Moar, 1975 | | Sardine-2 Terrace (24.4-32 m) | | | |
| | **Cooper & Kostro, 2006 | | Knights Point Terrace (113-110 m) | | | ?Sardine-2 Terrace |
| | Bull & Cooper, 1986 | | MIS 5 Terrace(s) (686-899 m) | | | |
| | Bishop, 1985 | $h_7$ strandline (ave. 500 m) | $h_6$ strandline (ave. 370 m) | $h_5$ strandline (ave. 300 m) | $h_4$ strandline (ave. 210 m) | |
| | Ward, 1988b | Terrace 5 (230 m) | Terrace 4 (140 m) | Terrace 3 (90 m) | Terrace 2 (60 m) | |
| | Kim & Sutherland, 2004 | | MIS 5e strandline (ave. 65 m) | | | |
| Otago | Cotton, 1957 | | 40 ft terrace (marine sediments, max height 12.2 m) | | | |
| | **Rees-Jones et al. 2000 | | | | $h_2$ (beach sand, ~9 m high sea level) | |
| | **Litchfield & Lian, 2004 | | $h_2$ (beach sand, ~9 m high sea level) | | | |
| | **Litchfield & Lian, 2004 | | Blueskin Bay Terrace (4-8 m) | | | |
| | **Kennedy et al, 2007 | | | | Shag Point shore platform (~7.4 m mean low water) | |
| Marlborough - Canterbury — Canterbury Plains | **Brown, 1988 | | Bromley Formation (79-5 m below msl) | | | |
| | **Shulmeister et al, 1999 | Unit C (lagoon sediments, 61.5-48.2 m bmsl) | | Unit J (beach deposit, 22.7-22.1 m bmsl) | Unit H (marine embayment, 34.1-24.6 m bmsl) | |
| Waipara R. to Motunau B. | Suggate, 1965 | | Hurunui-Blythe Rivers Terrace (54.9 m) | | | |
| | Suggate, 1965 | Bob's Flat (106.7-97.3 m) | Motunau Plain Lower Terrace (54.9-45.7 m) | | | |
| | Carr, 1970 | | Tiromoana Terrace (80-45 m) | | | |
| | **Oakley et al., 2017; 2018 | | Bob's Flat Terrace (128 m) | Tiromoana Terrace (63 m) | Motunau Terrace (marine fossil 70 m) | |
| Haumuri Bluff | Suggate, 1965 | Upper Terrace (?121.9-106.7 m) | Amuri Bluff Lower Terrace (48.8-45.7 m) | | | |
| | **Ota et al., 1984 | Kemps Hill Terraces. Upper (240-220 m) Lower (170-150 m) | Amuri Bluff Terrace (100-40 m) | | | |
| | **Ota et al., 1996 | | | Tarapuhi Terrace (165 m) | Kemps Upper (110 m) | Kemps Lower (90 m) |
| | **Oakley et al., 2017; 2018 | | | Tarapuhi Terrace (173.1 m) | Kemps Hill Upper (98 m), Keps Hill Lower (90 m), Amuri Bluff (40.7 m) | |
| Kaikoura Peninsula | Suggate, 1965 | 200-170 ft terrace (60.9-51.8) | | | | |
| | **Ota et al., 1996 | | | Terrace 1 (105 m) | Terrace II (80 m) | Terrace III (60 m) |
| Long Clarence River Point | Suggate, 1965 | Parikawa Formation (152 m) | | | | |
| | **Ota et al., 1996 | UM Terrace (222-190 m) | MM Terrace (shore platform 143-105 m) = Parikawa Formation | | | |
| | **Ota et al., 1996 | | MM Terrace (inner margin 80-55 m) = Parikawa Formation | | | |

**indicates source of RSL indicator

**Table 2: Indicative meaning of relative sea-level indicators**

| Name | Indicative Range | | Reference Water Level (in relation to tidal level or geodetic datum) |
|---|---|---|---|
| | upper limit | lower limit | |
| Beach deposit | ordinary berm | breaking depth | (Ordinary berm + breaking depth) / 2 |
| Beach swash | spring tidal range / 2 **OR** mean higher high water | mean sea level | (spring tidal range / 2 **OR** mean higher high water + mean sea level) / 2 |
| Marine Terrace | storm wave swash height | breaking depth | (storm wave swash height + breaking depth) / 2 |
| Shore platform | mean higher high water | midway between mean lower low water and breaking depth | (mean higher high water + (breaking depth - mean lower low water) / 2) / 2 |
| Lagoon deposit | mean lower low water | depth of lagoon bottom | (mean lower low water + modern lagoon depth) / 2 |

**Table 3: Numerical estimates of the height and timing of MIS 5 sea-level highstands used in age correlations of RSL indicators and/or to generate uplift rates**

| RSL Indicator Primary Reference | Last Interglacial | | |
|---|---|---|---|
| | Height (m) | Age (ka) | Reference(s) |
| **MIS 5** | | | |
| Te Punga 1962; Fleming, 1972 | - | 120-80 | - |
| Mildenhall, 1995 | - | 125-70 | Pillans, 1991 |
| **MIS 5e** | | | |
| Chappell, 1970; 1975 | 5 ± 3 | 120 | Chappell, 1974; Bloom et al., 1974 |
| Pillans, 1983; 1990b | 5 | 120 | Chappell & Veeh, 1978 |
| Palmer, 1988 | - | c. 120 | Pillans, 1983 |
| Berryman, 1993 | 6 ± 5 | 124 ± 5 | Chappell & Shackleton, 1986 |
| Kim and Sutherland, 2004 | 3 ± 2 | 120 | Lambeck & Chappell, 2001 |
| Litchfield and Lian, 2004 | - | 128-113 | Chappell et al., 1996 |
| Cooper and Kostro, 2006 | 5 ± 2 | c. 125 | Veeh & Chappell, 1970; Harmon et al., 1983; Chappell et al., 1996; Stirling et al., 1996 |
| Wilson et al 2007 | 0 ± 5 | 125 ± 5 | Pillans et al., 1998 |
| Oakley et al, 2017; 2018 | 5 ± 2 | 124.5 ± 5.5 | Lambeck & Chappell, 2001; Siddall et al., 2007 |
| **MIS 5c** | | | |
| Chappell, 1970; 1975 | -14 | 104 | Chappell, 1974; Bloom et al., 1974 |
| Berryman, 1993 | -19 ± 5 | 106 ± 5 | Chappell & Shackleton, 1986 |
| Ota et al., 1996 | -9 ± 3 | 100 ± 5 | Chappell & Shackleton, 1996 |
| Wilson et al 2007 | -28 ± 5 | 105 ± 5 | Pillans et al., 1998 |
| Oakley et al, 2017; 2018 | -23 ± 11 | 106.9 ± 3.0 | Lambeck & Chappell, 2001; Siddall et al., 2007 |
| **MIS 5b** | | | |
| Ota et al., 1996 | -24 ± 5 | 96 ± 5 | Chappell & Shackleton, 1986 |
| **MIS 5a** | | | |
| Chappell, 1970; 1975 | -15 | 83 | Chappell, 1974; Bloom et al., 1974 |
| Berryman, 1993 | -19 ± 5 | 81 ± 5 | Chappell & Shackleton, 1986 |
| Ota et al., 1996 | -19 ± 5 | 81 ± 5 | Chappell & Shackleton, 1986 |
| Rees-Jones et al 2000 | -20 to 0 | c. 80 | Pillans, 1983; Chappell & Shackleton, 1986; Ludwig et al., 1996 |
| Wilson et al 2007 | -24 ± 5 | 80 ± 5 | Pillans et al., 1998 |
| Oakley et al, 2017; 2018 | -50 ± 10 | 71.3 ± 2.0 | Lambeck & Chappell, 2001; Siddall et al., 2007 |


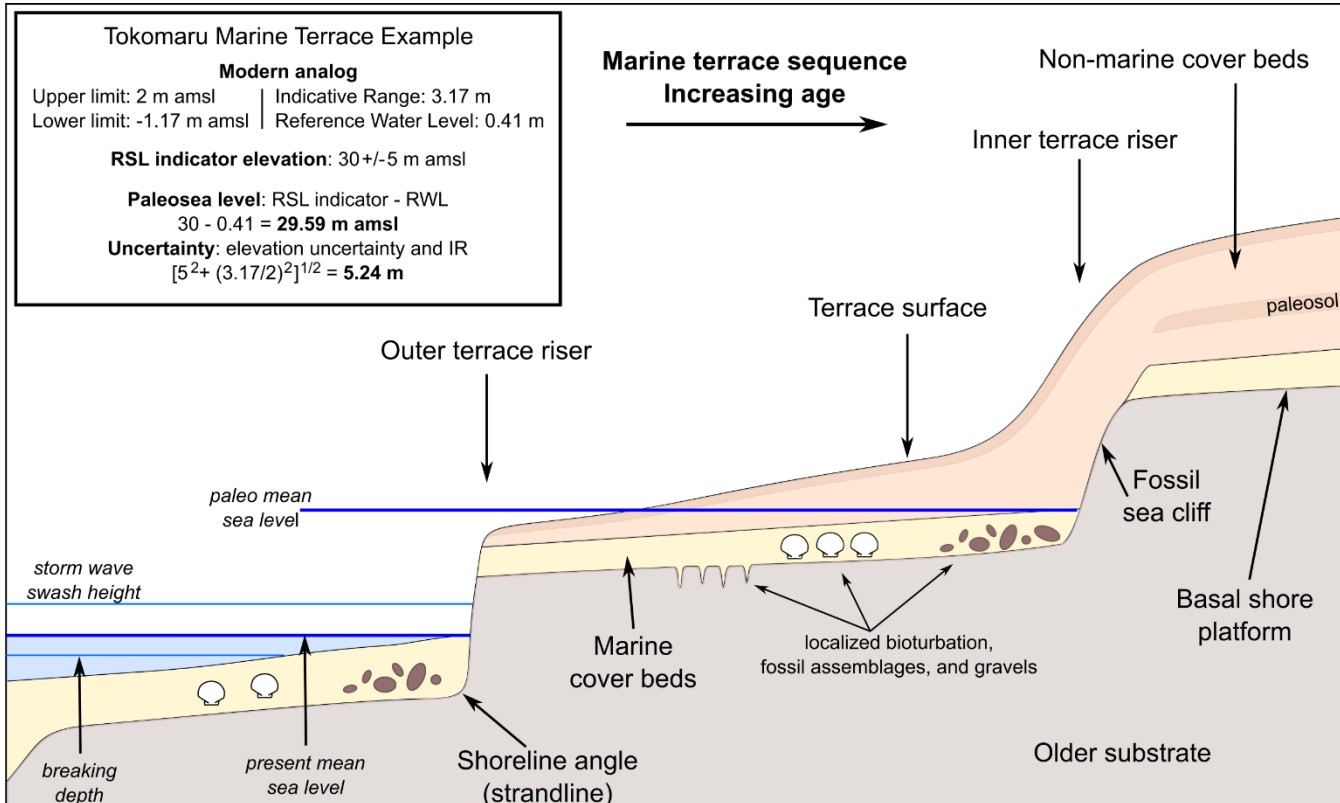

**Figure 1: Illustration of marine terrace terminology (adapted from Pillans, 1990b). The indicative range of a marine terrace landform is marked by the storm wave swash height (upper limit) and breaking depth (lower limit; Rovere et al., 2016). (See Table 2 for indicative range of other RSL indicators.) Marine terrace cover beds can be terrestrial-limiting (terrestrial sediments) or either marine-limiting or a direct indicator of sea level depending upon the availability of stratigraphic or sedimentologic information with which to relate to sea level. In this analysis, basal shore platforms underlying marine sediments were considered marine-limiting when elevations were taken from the outer terrace riser or an undefined distance from the inner margin and relationship with paleo-sea level could not be constrained. The inset provides a worked example of the paleo sea-level calculation for the MIS 5 fossil sea cliff (30 ± 5 m amsl) of the Tokomaru Terrace near Otaki (Palmer et al., 1988; Section 4.1.6).**

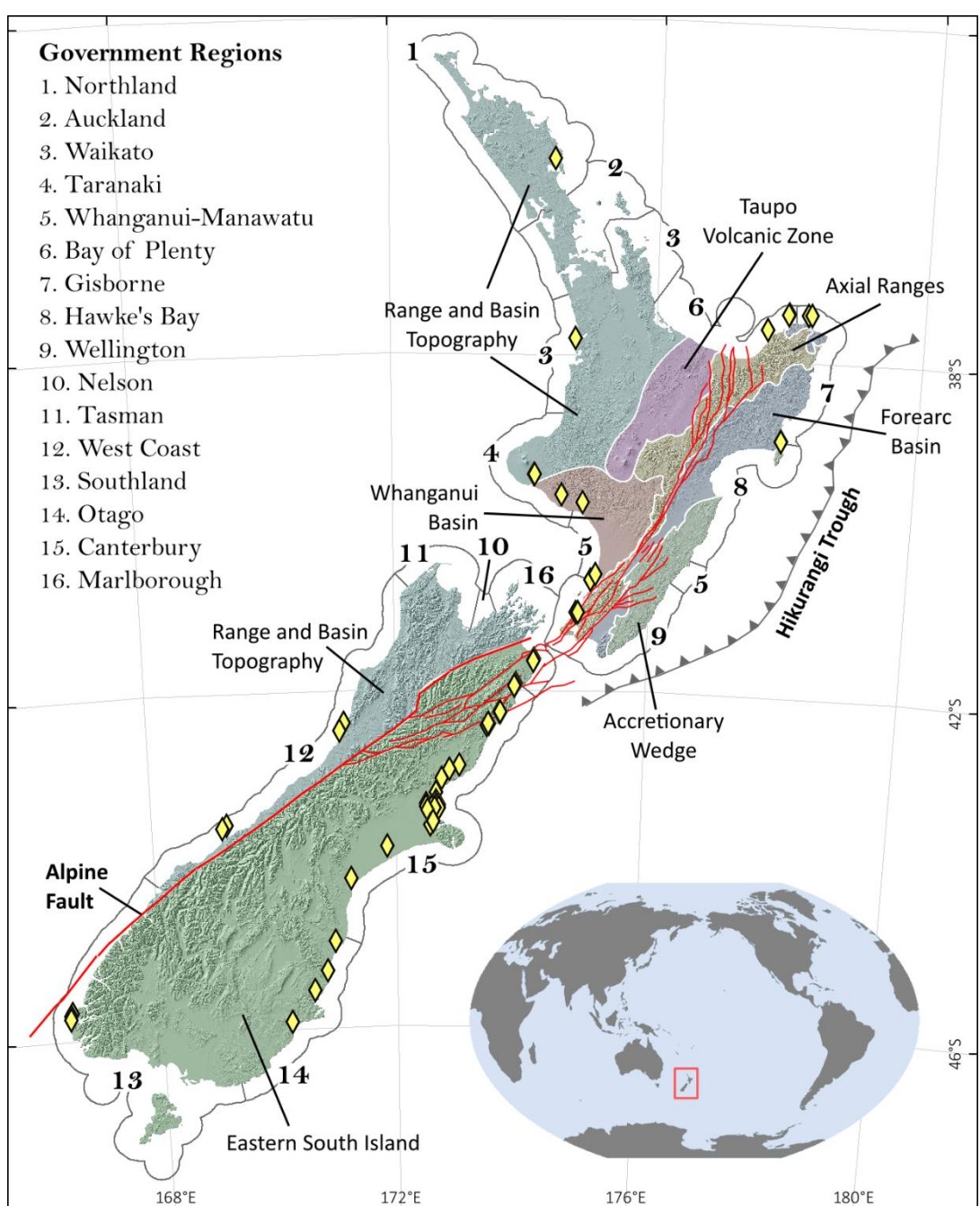

Figure 2: **The New Zealand North and South Islands illustrated with government regions and dominant tectonic regimes (discerned by color-shading) drawn after Ballance (2017) and Villamor et al. (2017). Also shown are the primary faults of the North Island Dextral Fault Belt, the Marlborough Fault System, and the Alpine Fault (Langridge et al., 2016). Diamonds indicate locations of RSL indicators within the WALIS database and described here. Hill shade data sourced from the LINZ Data Service and licensed for reuse under CC BY 4.0 (LINZ, 2020b).**

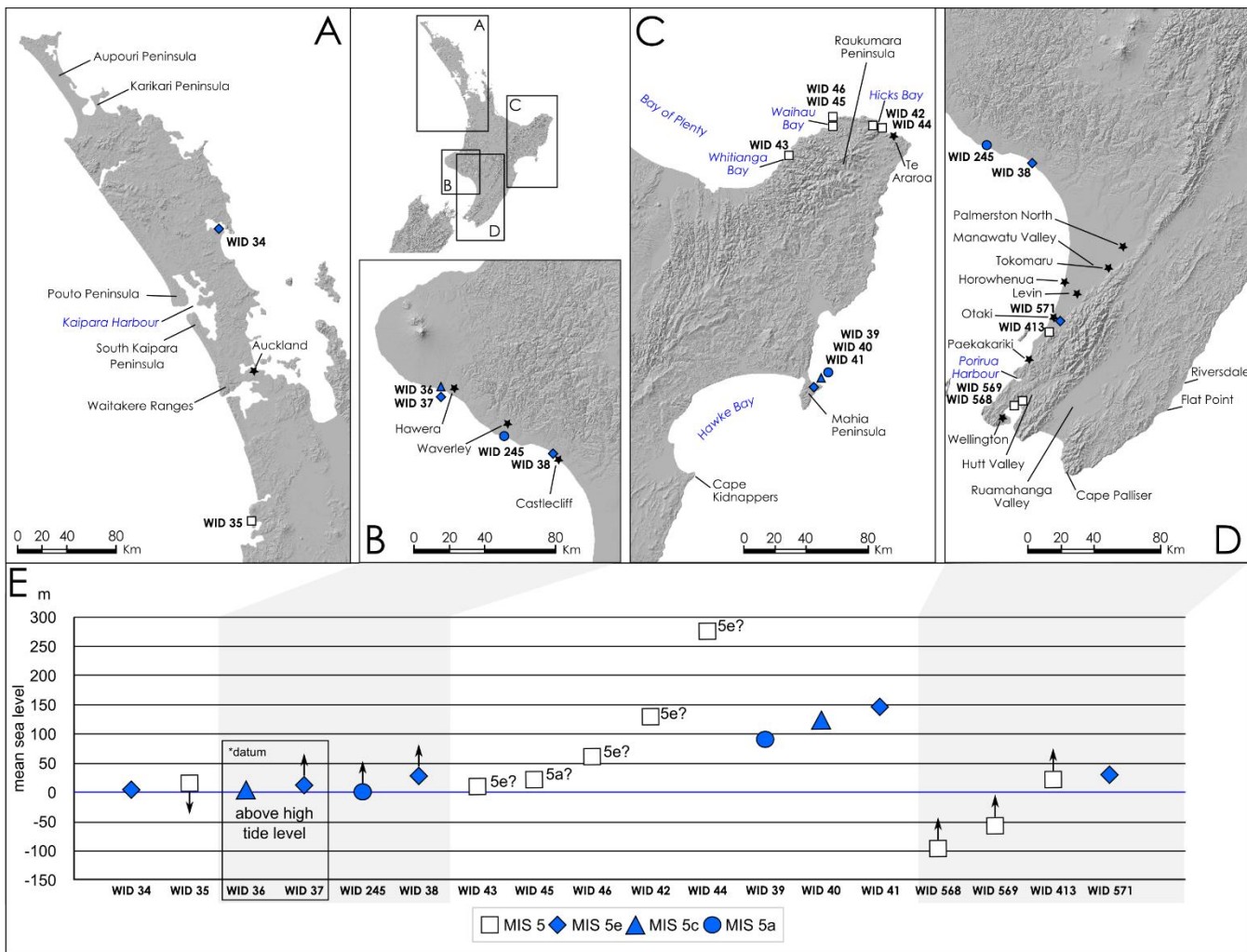

**Figure 3: Position, elevation, and age correlation of the North Island RSL indicators within the WALIS database and described here. Elevations are provided above mean sea level (Section 3) unless indicated otherwise; e.g. WID 36 and 37. Each RSL indicator is identified with their unique WALIS database identifiers (e.g. WID 34) and is correlated in text with their original publication. Symbols of indicators that would stack, extend offshore for clarity. Marine- and terrestrial- limiting points are indicated with up- and down- arrows, respectively. Locations and landscape features mentioned in text are also shown with built-up areas (e.g. towns and cities) indicated by stars. Hill shade data sourced from the LINZ Data Service and licensed for reuse under CC BY 4.0 (LINZ, 2020b).**

1880

1885

1890

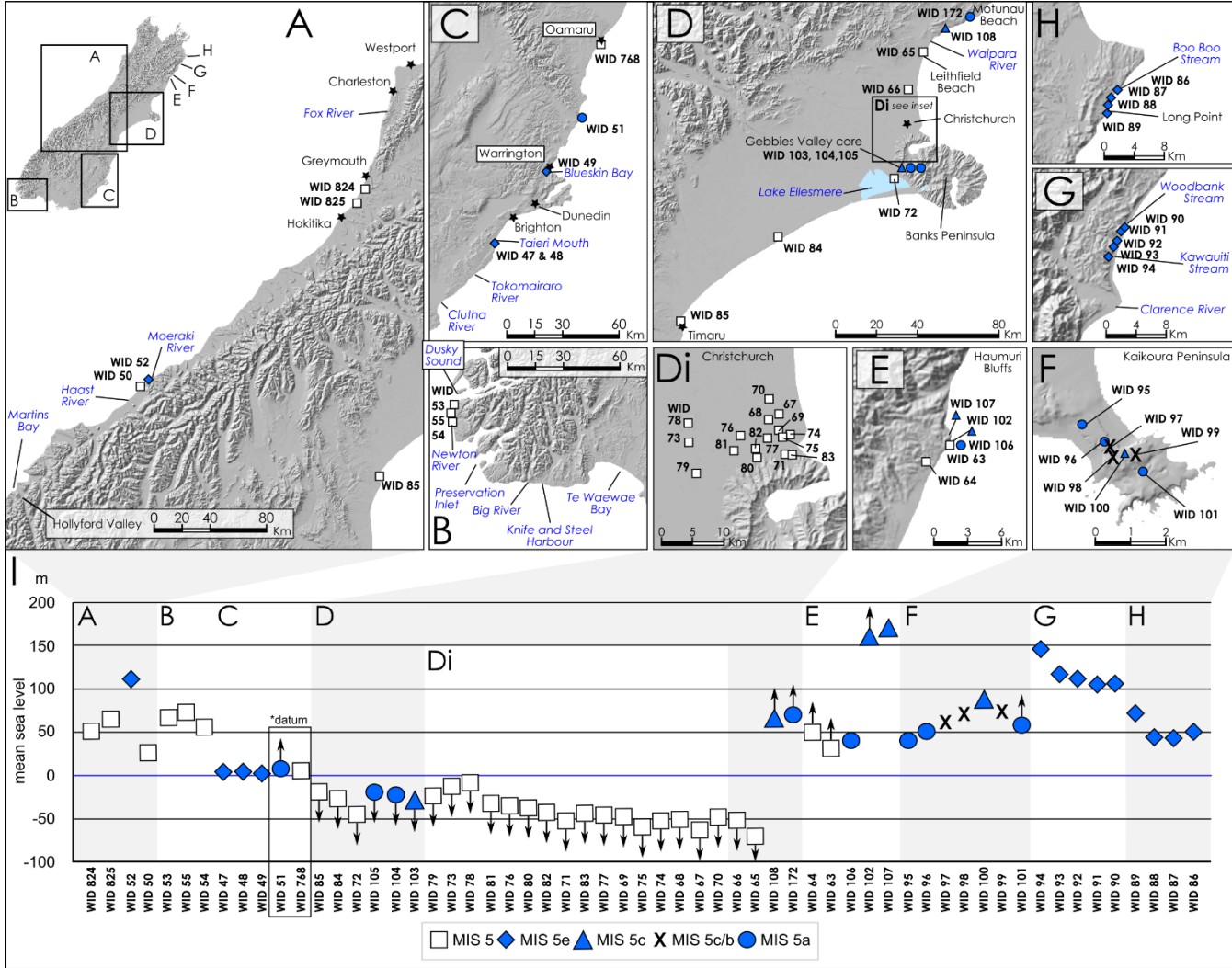

**Figure 4: Position, elevation, and age correlation of the South Island RSL indicators within the WALIS database and described here.** Elevations are provided above mean sea level (Section 3) except for WID 51 and WID 768 (Panel C), for which the sea-level datums are mean low tide and mean higher high water, respectively. Each RSL indicator is identified with their unique WALIS database identifiers (e.g. WID 34) and is correlated in text with their original publication. Symbols of indicators that would stack extend offshore for clarity. Marine- and terrestrial- limiting points are indicated with up- and down- arrows, respectively. . Locations and landscape features mentioned in text are also shown with built-up areas (e.g. towns and cities) indicated by stars. Hill shade data sourced from the LINZ Data Service and licensed for reuse under CC BY 4.0 (LINZ, 2020b).

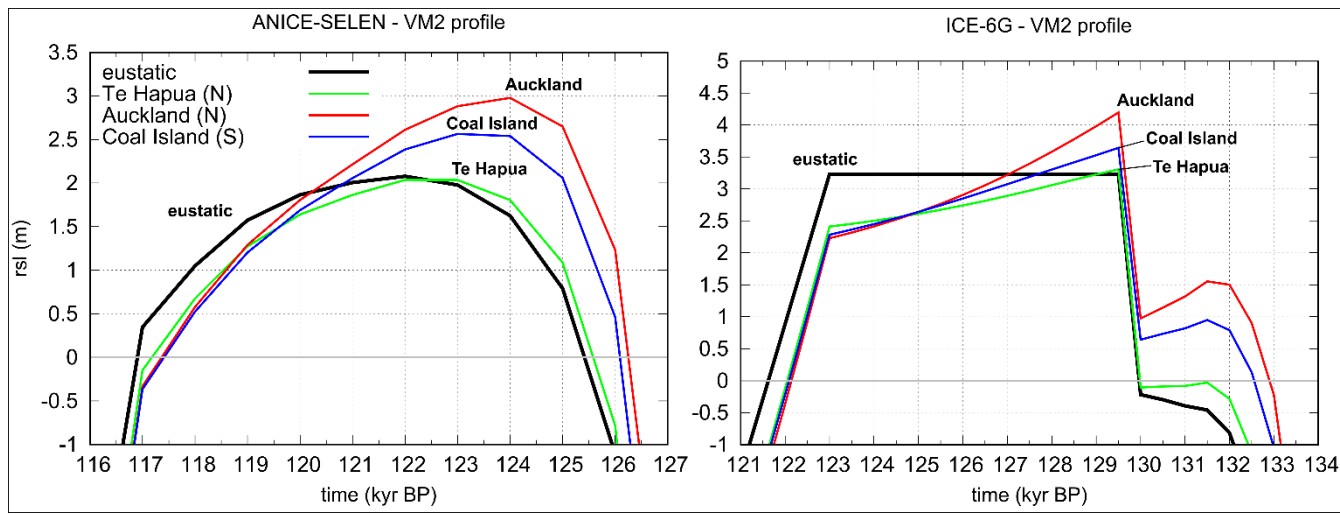

**Figure 5: Local RSL curves for MIS 5e sea level at the northernmost tip of North Island (Te Hapua; latitude -34.39, longitude 173.02), Auckland (latitude -36.85, longitude 174.76), and the southernmost tip of South Island (Coal Island; latitude -46.21, longitude 166.66) generated from ANICE-SELEN and ICE-6G models. Similar to the Holocene, sea level peaks earlier and higher in the North Island. The Northland region (Te Hapua) RSL curve is nearest to eustatic. Deviations of the RSL curves from eustatic within the models is driven by ocean syphoning, suggesting it serves as a primary driver of variability in the timing and height of peak sea level across New Zealand. Note the different scale to x- and y- axes between model outputs.**

1915