# Peer review of "The last interglacial sea-level record of Aotearoa New Zealand"

_Earth System Science Data, 2020_

## Short Comment (SC1) · 8 Nov 2020

This is a nice review of the MIS 5 sea level indicators in New Zealand. I have a minor comment, however. I noticed that you say of the Oakley et al. (2017) samples that "Precise sample locations are not provided, but are shown on a publication map" (lines 749-750), so I wanted to point out that a spreadsheet of the precise locations can be found in the supplementary materials with that paper.

---

## Author Comment (AC1) · 9 Nov 2020

Dear Dr. Oakley. Thank you for bringing to our attention that precise locations for samples from Oakley et al. (2017) are available within the supplementary materials. We apologize for the oversight and will correct the manuscript and WALIS database.

---

## Referee Comment (RC1) · Tim Naish (Referee) · 10 Nov 2020

Overview

The paper by Ryan et al on "The Last Interglacial Sea-Level Record of New Zealand" is a very comprehensive and robust review/evaluation of the previous studies and the literature. It will be an important review for anyone considering further work and it makes some clear recommendations for future work. I commend the authors on this paper, but I do have some concerns about the way in which it assesses some of the previous work. I declare up front that I may be a little defensive having spent most of my career working on sea-level records in New Zealand, and I have worked closely with many of the researchers cited and assessed in this paper.

[Figure]

I think its very important that the authors show they understand the history and context of sea-level studies in New Zealand. Within the Quaternary community New Zealand has held a special place because of its wide range of terrestrial and marine deposits – owing to its convergent plate boundary setting. As I outline below, many of these studies were pioneering and well ahead of the thinking in the Northern Hemisphere that up until the 1980s was influenced by a "Pleistocene" containing 4 glacials and intervening interglacials. While I realise that this paper primarily focusses on the LIG sea-level records and deposits, the introduction and the discussion do refer to the earlier work on older deposits and the LIG is discussed in this context, but I believe a bit inappropriately in the context of the broader work.

Its all very well to say that the LIG deposits are poorly dated, measured and described for sea-level indicators, compared to modern standards, but New Zealand geologists have known that they have always been of limited value for global sea-level reconstructions, because of extensive and widespread tectonic deformation. As the authors quite rightly point out, the dated sea-level indicators where used for understanding long term rates of VLM, not global sea-level change. The paper shows quite clearly that after a robust re-analysis that there is only one site in Northland that has the potential to contribute to global sea-level studies. There is considerable uncertainty about the tectonic stability of Otago coastline based on recent reconstructions using salt marsh cores, GPS and INSAR and geodynamical studies. Also, the latest geodetic work actually shows that Auckland is not stable and is subsiding, slowly.

I initially thought this paper, rather than being a review, was going to contribute some new site elevation data or information on shoreline indicators or geochronology, but realised that that is not really the case. It concludes that New Zealand still has the potential to be an important South Pacific site, based on the one Northland site. I was slightly surprised by the GIA discussion that suggested there could be significant deviations from eustatic during interglacials, when the paper cited, for which Stocchi did the GIA modelling, showed this not be case, especially for the plausible range of

LIG ice sheet configurations and meltwater contributions.

My overall recommendation is that this is an important and useful review, but that the authors should relook at the context in which they introduce and discuss the review and adjust the text accordingly.

Specific comment follow

• Introduction – I was surprised to see the Introduction treat previous work a little dismissively. There are many challenges to identifying, mapping, dating and correlating paleoshorelines, inferred strandlines and wave cut marine terraces in such a tectonically active setting. Given that much of this work occurred in the pre-1980s, with some outstanding geomorphic and geological descriptions and interpretations ahead of their time (e.g. Fleming, Chappell, Gage, Suggate and Pillans), a more generous approach could be taken recognising this early pioneering work. The work of Brad Pillans on the South Taranaki-Whanganui marine terrace sequences remains one of the classic examples in the world today of a suite of wave cut platforms and associated strandlines recording every interglacial from MIS 3-13. The authors should also appreciate more generously, the difficulty in dating these deposits, which also hampers many recent studies (e.g. Hearty et al. 2020 South Africa), and has long been acknowledged by people such as Brad Pillans and Kelvin Berryman as a challenge in NZ. Its only really in the last decade that new approaches for identifying shoreline indicators and the influence of post-depositional processes such as GIA and dynamic topography have been appreciated. However, the authors of this paper have done a nice job of recognising the need for modern approaches of analysing shoreline indicators in their reassessment of the NZ record.

• Quaternary oxygen isotope records were published in the late 1970s and early 1980s, which provided a revolutionary tool to understand and date the NZ marine terraces mapped by earlier workers and interpreted within the existing paradigm of 4 NH glaciations. As the authors highlight, absolute age determination of strandlines was

limited to the intermittent tephra, biostratigraphic inferences, thermoluminescence and amino acid racemisation dating which have many assumptions and problems. Notwithstanding these challenges, it became clear that some strand lines and terraces could be correlated with specific interglacials identified in the benthic oxygen isotope record, and that based on this, particularly in coastal Taranaki and Whanganui, each interglacial shoreline could be matched to successively older orbitally-paced interglacials. This allowed two things to be established: 1) That there was far field physical evidence that confirmed interpretations of the newly developed benthic isotope sea-level proxy records, and that they really were measuring global ice volume and sea-level (older than the Huon Peninsula coral terraces) [I had this conversation several times with the late Sir Nicholas Shackleton, who always recognised this significance], and 2) based on this relationship, an orbital chronology could be established that allowed uplift rates to be estimated for western North Island.

• It should be noted that pre-GPS surface elevations were largely achieved by handheld altimetry. Moreover, the focus was never really on reconstructing the absolute magnitude of RSL because of the tectonic complication. Although, paleo shorelines were identified, dated and correlated to the benthic oxygen isotope stack the main outcome was constraining vertical land movements thought to be primarily driven by the active plate boundary setting.

• Line 90 – Written negatively. This was not "an unfortunate outcome" that there were many different names for terraces around NZ, it was the correct stratigraphic approach at the time. Back when they were first described and named the relationship between marine terraces and their deposits in different parts of NZ could not be established, and in some cases are still difficult to establish. The recognition, through improved age control, the development and application of the benthic isotopic proxy of global sea-level then allowed different terrace sequences from different parts of New Zealand to be compared within a unifying chronostratigraphic framework. Pillans 1990 made a big step forward in this regard. Likewise the New Zealand Neogene sedimentary stratigraphy went through the same evolution, of initially being subdivided at regional scale and then national scale on the basis of endemic biostratigraphic criteria, and then meaningless correlations were established the type sections in the Mediterranean. With the advent of magnetostratigraphy calibrated by radiometric dating of tephra, it became possible to correlate orbital sequences with benthic oxygen isotope curve (global climate) and astronomically calibrate age models.

• Table -1 For completeness you should cite Fleming 1953. For example he was the first to identify the Brunswick terrace, which Pillans subsequently dated as MIS 9 and adopted Fleming's nomenclature. Previously it had been incorrectly associated with the MIS 7 Ngarino Terrace.

• Line 199 – I understand that this is a generalistaion but eastern North Island vertical land movements (VLMs) both short and long-term are complex, partly because of the TVZ and mostly because of the Hikurangi margin, with subsidence in the Hawkes Bay Basin and near Mahia Peninsula and in the Bay of Plenty near Whakatane.

• Figure 3. Why did you not use the NT1 wave cut surface of Alloway et al. 2005 as a site?

• Also there is a lot of work on the Southern Wellington-Wairarapa coastline in Dee Ninnis' PhD thesis, that the authors had access to and the potential to include and collaborate with. Some figures below that were made available to the authors. Was there a reason for excluding this data?

Line 933 – GIA. It's a bit unclear what the message from this section really is. A number of studies (Pliocene and Recent) have shown that New Zealand is largely unaffected by GIA and sits close to the GMSL in most reconstructions where polar meltwater contributions and viscosity models are varied. While a local signal due to water loading of the shelf has been implied for the Holocene, resolving the fingerprint and any flexural wave along NZ is extremely unlikely for the last interglacial given there is really only one site in Northland that is likely to be unaffected by tectonics, and considering the

uncertainties the authors discuss. It has been implied the effect of GIA could be 5% deviation from North to South. One of the co-authors did a range of GIA experiments for the Pliocene and showed that under a reasonable range of ice histories, NZ lay on the eustatic. For NZ to have a significant GIA deviation during the LIG most of the meltwater would need to come from NH (Greenland) which seems precluded by most geological and ice core reconstructions.

Line – 1134 - It wouldn't hurt the authors to cite Naish et al 1998 (QSR) or Pillans et al (2005) here, if they are going refer to these older sequences.

Line – 1150 – I would suggest the authors also read Grant et al 2019 (Nature), and some of the other Whanganui-based Pleistocene literature. This statement is misleading and shows a lack of understanding as to how the older sea-level reconstructions are established using the shallow-marine record. They are not absolute sea-level indicators like wave cut platforms and strandlines. They cannot be compared to MIS 5 sea-level indicators around NZ. They do however fundamentally record many transgressions and regressions of the shoreline in response to orbitally-paced global sea-level and associated water depth changes. But reconstructed water depth changes are also affected by local tectonic subsidence, compaction, and loading that all have to be accounted for when deriving the amplitude glacial-interglacial global sea-level change. The sea-level cycles reconstructed from the Whanganui sequences cannot be registered directly to present day sea-level because of late Quaternary uplift of western North island, which is the reason the marine terraces are preserved post MIS 15.

---

## Author Comment (AC2) · 26 Jan 2021

**Response to Comments by Prof. Tim Naish (Referee) to "The last interglacial sea-level record of New Zealand (Aotearoa)"**

Deirdre D. Ryan[1*], Alastair J.H. Clement[2], Nathan R. Jankowski[3,4], Paolo Stocchi[5]

[1]MARUM – Center for Marine Environmental Sciences, University of Bremen, Bremen, Germany
[2]School of Agriculture and Environment, Massey University, Palmerston North, New Zealand
[3] Centre for Archeological Science, School of Earth, Atmospheric and Life Sciences, University of Wollongong, Wollongong, Australia
[4]Australian Research Council (ARC) Centre of Excellence for Australian Biodiversity and Heritage, University of Wollongong, Wollongong, Australia
[5]NIOZ, Royal Netherlands Institute for Sea Research, Coastal Systems Department, and Utrecht University, PO Box 59 1790 AB Den Burg (Texel), The Netherlands

26 January 2021

**1. Summary**

We thank Prof. Naish for his review of our manuscript. Within this response we address each comment individually and detail any changes made to the original manuscript. Comments by Prof. Naish are provided in italic, with our response in standard font, and changes to manuscript are in blue.

**2. Overview Comment**

*The paper by Ryan et al on "The Last Interglacial Sea-Level Record of New Zealand" is a very comprehensive and robust review/evaluation of the previous studies and the literature. It will be an important review for anyone considering further work and it makes some clear recommendations for future work. I commend the authors on this paper, but I do have some concerns about the way in which it assesses some of the previous work. I declare up front that I may be a little defensive having spent most of my career working on sea-level records in New Zealand, and I have worked closely with many of the researchers cited and assessed in this paper.*

*I think its very important that the authors show they understand the history and context of sea-level studies in New Zealand. Within the Quaternary community New Zealand has held a special place because of its wide range of terrestrial and marine deposits – owing to its convergent plate boundary setting. As I outline below, many of these studies were pioneering and well ahead of the thinking in the Northern Hemisphere that up until the 1980s was influenced by a "Pleistocene" containing 4 glacials and intervening interglacials. While I realise that this paper primarily focusses on the LIG sea-level records and deposits, the introduction and the discussion do refer to the earlier work on older deposits and the LIG is discussed in this context, but I believe a bit inappropriately in the context of the broader work.*

*Its all very well to say that the LIG deposits are poorly dated, measured and described for sea-level indicators, compared to modern standards, but New Zealand geologists have known that they have always been of limited value for global sea-level reconstructions, because of extensive and widespread tectonic deformation. As the authors quite rightly point out, the dated sea-level indicators where used for understanding long term rates of VLM, not global sea-level change. The paper shows quite clearly that after a robust re-analysis that there is only one site in Northland that has the potential to contribute to global sea-level studies. There is considerable uncertainty about the tectonic stability of Otago coastline based on recent reconstructions using salt marsh cores, GPS and INSAR and geodynamical studies. Also, the latest geodetic work actually shows that Auckland is not stable and is subsiding, slowly.*

*I initially thought this paper, rather than being a review, was going to contribute some new site elevation data or information on shoreline indicators or geochronology, but realised that that is not really the case. It concludes that New Zealand still has the potential to be an important South Pacific site, based on the one Northland site. I was slightly surprised by the GIA discussion that suggested there could be significant deviations from eustatic during interglacials, when the paper cited, for which Stocchi did the GIA modelling, showed this not be case, especially for the plausible range of LIG ice sheet configurations and meltwater contributions.*

*My overall recommendation is that this is an important and useful review, but that the authors should relook at the context in which they introduce and discuss the review and adjust the text accordingly.*

Responses to the below specific comments address the issues summarized within this overview comment. Although Prof. Naish seems to feel that we are judging the New Zealand record harshly, the intention of this review was to provide readers an understanding of the current state of knowledge of last interglacial sea level within New Zealand. The conclusion that the record is lacking in some areas necessary for sea-level interpretations is not due to a lack of quality research in New Zealand, but rather that it has not been the focus of most research. This conclusion is hardly new as it is an echo of the conclusions made by Gage (1953) and Pillans (1990a) in earlier reviews of shoreline studies within New Zealand as noted within the Introduction.

**3. Specific Comments**

**Comment 1**: *Introduction – I was surprised to see the Introduction treat previous work a little dismissively. There are many challenges to identifying, mapping, dating and correlating paleoshorelines, inferred strandlines and wave cut marine terraces in such a tectonically active setting. Given that much of this work occurred in the pre-1980s, with some outstanding geomorphic and geological descriptions and interpretations ahead of their time (e.g. Fleming, Chappell, Gage, Suggate and Pillans), a more generous approach could be taken recognising this early pioneering work. The work of Brad Pillans on the South Taranaki-Whanganui marine terrace sequences remains one of the classic examples in the world today of a suite of wave cut platforms and associated strandlines recording every interglacial from MIS 3-13. The authors should also appreciate more generously, the difficulty in dating these deposits, which also hampers many recent studies (e.g. Hearty et al. 2020 South Africa), and has long been acknowledged by people such as Brad Pillans and Kelvin Berryman as a challenge in NZ. Its only really in the last decade that new approaches for identifying shoreline indicators and the influence of post-depositional processes such as GIA and dynamic topography have been appreciated. However, the authors of this paper have done a nice job of recognising the need for modern approaches of analysing shoreline indicators in their reassessment of the NZ record.*

It is unfortunate that Prof. Naish feels we have treated previous work in New Zealand a little dismissively as this was certainly not our intention. We have added language throughout the Abstract and Introduction (Section 1) to 'soften' our approach and provide greater context to the amount and quality of previous research and to highlight the difficulties confronting sea-level studies in New Zealand. We recognize that NZ has a rich record of paleo-shoreline studies and it is for this reason that we chose to write a succinct overview of previous work, directly related to the topic of the manuscript – sea-level records of the last interglacial. Section 4 (the bulk of the manuscript) has a very detailed review of previous studies concerning MIS 5 records. This section is quite long, reflecting the development and wealth of research completed in New Zealand. Section 5 provides a detailed discussion of difficulties in establishing geochronology, and Section 6 touches upon (albeit briefly) on the length of the record beyond the LIG. We are mindful that given the length of the manuscript already and the extensive work that has been done a more detailed review would expand the manuscript beyond its scope. We hope that our softening of the abstract and introduction is sufficient. Rewritten or added sentences are:

Abstract

> Extensive coastal deformation around New Zealand has prompted research focused on active tectonics, which requires less precision than sea-level reconstruction. The range of paleo-shoreline elevations are significant on both the North Island (276.8 to -94.2 m msl) and South Island (173.1 to -70.0 m msl) and have been used to estimate rates of vertical land movement; however, in many instances lack adequate description and age constraint for high-quality RSL indicators.

Introduction

> This latter characteristic has facilitated the preservation of marine terrace sequences spanning the Pleistocene and into the Holocene that have been the target of much research and discussion of New Zealand geomorphology and geology since the late 19[th] Century. Importantly, the terraces also serve as the primary source of last interglacial paleo-sea level records.

> Resolving a sea-level record from New Zealand marine terraces (and other sea-level indicators) has historically been complicated by the long-term and ongoing coastal deformation (Section 5.2) and difficulties in geochronology (Section 5.3).

This review of the LIG sea-level indicators in New Zealand, like the preceding reviews by Gage (1953) and Pillans (1990a), found considerable lack of necessary detail in the published literature for the identification of robust sea-level indicators defined by the WALIS framework; including that published post-1990.

It must be stressed that the lack of detail for last interglacial sea-level studies in New Zealand reflects the active landscape and that the focus of many studies has not been sea-level at all, but rather understanding the active tectonics and vertical land movements of the archipelago, which require less precision than sea-level reconstruction. Furthermore, until late into the 20th Century studies had to contend with difficulties in correlating distant sequences due to limited geochronological methods and still must grapple with the tectonically active nature of the islands and difficult topography.

**Comment 2**: *Quaternary oxygen isotope records were published in the late 1970s and early 1980s, which provided a revolutionary tool to understand and date the NZ marine terraces mapped by earlier workers and interpreted within the existing paradigm of 4 NH glaciations. As the authors highlight, absolute age determination of strandlines was limited to the intermittent tephra, biostratigraphic inferences, thermoluminescence and amino acid racemisation dating which have many assumptions and problems. Notwithstanding these challenges, it became clear that some strand lines and terraces could be correlated with specific interglacials identified in the benthic oxygen isotope record, and that based on this, particularly in coastal Taranaki and Whanganui, each interglacial shoreline could be matched to successively older orbitally-paced interglacials. This allowed two things to be established: 1) That there was far field physical evidence that confirmed interpretations of the newly developed benthic isotope sea-level proxy records, and that they really were measuring global ice volume and sea-level (older than the Huon Peninsula coral terraces) [I had this conversation several times with the late Sir Nicholas Shackleton, who always recognised this significance], and 2) based on this relationship, an orbital chronology could be established that allowed uplift rates to be estimated for western North Island.*

Sentences were added to highlight the implementation of marine oxygen-isotope records in the interpretation of New Zealand sea-level and geomorphology. Specifically,

Following the publication of marine oxygen-isotope records, age determination moved beyond the constraints of the Mediterranean and Huon Peninsula sequences and New Zealand marine terraces were correlated with specific sea-level intervals reflecting global sea-level and ice-volume (e.g. Beu and Edwards, 1984; Ward, 1988a; Pillans, 1980a; 1994; Suggate, 1992; Berryman, 1993; Ota et al., 1996). These correlations not only provided greater certainty in the interpretation of New Zealand geomorphology, but also lent strength to the newly developed marine oxygen-isotope sea-level proxy records.

**Comment 3**: *It should be noted that pre-GPS surface elevations were largely achieved by handheld altimetry. Moreover, the focus was never really on reconstructing the absolute magnitude of RSL because of the tectonic complication. Although, paleo shorelines were identified, dated and correlated to the benthic oxygen isotope stack the main outcome was constraining vertical land movements thought to be primarily driven by the active plate boundary setting.*

The prevalent use of altimetry is noted in Section 3 "Location and elevation measurements" (Line 178), which the reader is referred to in Lines 54-55, "how geographic location, elevation, and associated uncertainty were assigned to each indicator (Section 3)". We are also not sure at which point of the introduction this comment was written, but at line 79, we expressly note that due to regional tectonics the research focus was the determining VLM and not sea level.

**Comment 4**: *Line 90 – Written negatively. This was not "an unfortunate outcome" that there were many different names for terraces around NZ, it was the correct stratigraphic approach at the time. Back when they were first described and named the relationship between marine terraces and their deposits in different parts of NZ could not be established, and in some cases are still difficult to establish. The recognition, through improved age control, the development and application of the benthic isotopic proxy of global sea-level then allowed different terrace sequences from different parts of New Zealand to be compared within a unifying chronostratigraphic framework. Pillans 1990 made a big step forward in this regard. Likewise the New Zealand Neogene sedimentary stratigraphy went through the same evolution, of initially being subdivided at regional scale and then national scale on the basis of endemic biostratigraphic criteria, and then meaningless correlations were established the type sections in the Mediterranean. With the advent of magnetostratigraphy calibrated by radiometric dating of tephra, it became possible to correlate orbital sequences with benthic oxygen isotope curve (global climate) and astronomically calibrate age models.*

The interpretation of our phrasing was not as intended and is adjusted as follows:

> Due to the difficulty in relating spatially distant marine terraces, in part due to difficulties in determining a numerical age, nomenclature for the terraces and their associated sediments is regionally specific and has evolved through time with better age constraint (Table 1; Section 4).

**Comment 5**: *Table -1 For completeness you should cite Fleming 1953. For example he was the first to identify the Brunswick terrace, which Pillans subsequently dated as MIS 9 and adopted Fleming's nomenclature. Previously it had been incorrectly associated with the MIS 7 Ngarino Terrace.*

Since submission of the manuscript we have obtained a copy of Fleming 1953 and agree it should be cited it in this work. Fleming 1953 has been added to Table 1 and in addition to further citations, the following was added to Section 4.1.2:

> Fleming (1953) provided early, detailed descriptions of the Brunswick and Rapanui Terraces, recognizing both as having formed over multiple cycles of sea-level transgression and regression (Table 1). The Rapanui Terrace was later divided into the older Ngarino Terrace and younger Rapanui Terrace (Dickson, 1974).

**Comment 6**: *Line 199 – I understand that this is a generalistaion but eastern North Island vertical land movements (VLMs) both short and long-term are complex, partly because of the TVZ and mostly because of the Hikurangi margin, with subsidence in the Hawkes Bay Basin and near Mahia Peninsula and in the Bay of Plenty near Whakatane.*

We agree that eastern North Island is tectonically complex and that our brief overview could be expanded. As suggested, we have added remarks to explain the subsidence in the east. This paragraph has been expanded to include the following:

> The entire eastern North Island constitutes the Hikurangi Margin forearc that is being compressed by the collision of the Pacific and Australian Plates (Nicol et al., 2017). The forearc structure from east to west is composed of an accretionary wedge (outer forearc, mostly located offshore), a forearc basin (inner forearc), and the Axial Ranges (frontal ridge) bisected by the North Island Dextral Fault Belt (NIDFB) (Berryman, 1988). The majority of the eastern North Island is being uplifted as the forearc is compressed with the exception of parts of Hawke's Bay: the Heretaunga Plains, a subsiding tectonic depression on the margin of the convergent plate boundary (Lee et al., 2011), and the coast near Wairoa, which experienced late Quaternary co-seismic subsidence (Ota et al., 1989a; Litchfield, 2008). The oblique subduction of the Pacific Plate is rotating the eastern North Island clockwise producing a backarc rift system and volcanic arc: the Taupo Volcanic Zone (TVZ). The Whakatane Graben, at the northern end of the TVZ, is a subsiding tectonic depression bounded by normal faults and infilled with up to 2 km of late Quaternary sediments (Wright, 1990; Beanland and Berryman, 1992).

**Comment 7**: *Figure 3. Why did you not use the NT1 wave cut surface of Alloway et al. 2005 as a site?*

Alloway et al. (2005) refers to NT1 and NT2 (wave cut surfaces correlated with MIS 5a and 5e, respectively), describing their stratigraphic relationship to debris-avalanche deposits. However, stratigraphic sections are drawn cm below surface and no heights above a sea-level datum are provided, precluding entry into WALIS.

**Comment 8**: *Also there is a lot of work on the Southern Wellington-Wairarapa coastline in Dee Ninnis' PhD thesis, that the authors had access to and the potential to include and collaborate with. Some figures below that were made available to the authors. Was there a reason for excluding this data?*

There are two reasons for this. First, the intention was only to include published peer-reviewed data, which precludes Masters and PhD theses. We did include some theses (e.g. Goldie, 1975; Hicks, 1975, etc.) that had been referenced by multiple later authors and that seemed likely would not be published independently given the time elapsed since completion. DDR and AC did discuss the inclusion of Dr. Ninnis's recently (2018) completed PhD work. However, we are aware that Dr. Ninnis is working towards publication of that work and did not want to 'scoop' what will be a valuable contribution to New Zealand geology and geomorphology and deserving of initial recognition under Dr. Ninnis's authorship. We would be very happy to have Dr. Ninnis archive her work in WALIS – indeed it is the purpose of WALIS and we encourage her to do so.

**Comment 9**: *GIA. It's a bit unclear what the message from this section really is. A number of studies (Pliocene and Recent) have shown that New Zealand is largely unaffected by GIA and sits close to the GMSL in most reconstructions where polar meltwater contributions and viscosity models are varied. While a local signal due to water loading of the shelf has been implied for the Holocene, resolving the fingerprint and any flexural wave along NZ is extremely unlikely for the last interglacial given there is really only one site in Northland that is likely to be unaffected by tectonics, and considering the uncertainties the*

*authors discuss. It has been implied the effect of GIA could be 5% deviation from North to South. One of the co-authors did a range of GIA experiments for the Pliocene and showed that under a reasonable range of ice histories, NZ lay on the eustatic. For NZ to have a significant GIA deviation during the LIG most of the meltwater would need to come from NH (Greenland) which seems precluded by most geological and ice core reconstructions.*

We agree, this can be written more clearly. For clarity, the entire section content is shown below with significantly altered or new sentences shown in blue font. We have also added a figure to help clarify the GIA discussion.

Excessive and prevalent coastal deformation will preclude the development of a sea-level reconstruction that can be registered to present day, regardless of the quality of sea-level indicators. The position of the New Zealand archipelago straddling the active boundary of the Australian and Pacific plates has produced a coastline subject to variable rates of vertical land movement (VLM) due to complex tectonics and displacement associated with earthquakes. Although displacement by an earthquake can have dramatic effect, interseismic deformation (deformation between earthquake events) is more likely to influence long-term trends in VLM (Beavan and Litchfield, 2012). As has been shown above, the New Zealand paleo-shorelines and marine terraces (and below-surface marine deposits) have been essential for determining estimates of long-term (beyond Holocene) rates of uplift or subsidence (e.g. Chappell, 1975; Pillans, 1983; Bishop, 1985; Suggate, 1992; Berryman, 1993; Begg et al., 2004; Wilson et al., 2007; Beavan and Litchfield, 2012; Oakley et al., 2018). However, because the research focus has been on determining long-term VLM rates, and due to a lack of adequate geochronological methods for many studies, potential sea-level indicators have been described in less detail than desired for such use. Although any sea-level reconstruction derived from these indicators may not be useful for a relative sea-level curve with relation to present sea level, more precise descriptions and age constraint can improve estimates of VLM rates, especially where there is uncertainty in correlation to the appropriate MIS 5 highstand.

GIA encompasses all deformational, as well as gravitational, and rotational-induced changes to relative sea level in response to the buildup and retreat of ice sheets with residual and variable affect along coastal sections depending upon their proximity to former glaciers, ice caps and sheets (Arctic and Antarctic, Simms et al., 2016). In other words, the magnitude and wavelength of the solid Earth response to ice-and water-load history varies with time (in relation to glacial maxima) and geographical location producing a gradient in relative sea level that is modulated by mantle rheology. Neglecting GIA on active coastlines when determining rates of VLM has been shown to lead to overestimated uplift rates at an overage of 40%, but also up to 72% (Simms et al., 2016; Stocchi et al., 2018).

New Zealand sits on a 'sweet spot' with respect to Antarctica, such that when the northern hemisphere ice sheets are neglected, the local RSL response to either growth or retreat of the Antarctic ice Sheet (AIS) is nearly eustatic. However, deviations from eustatic may increase dependent upon ice mass fluctuations within specific sectors of the AIS. In particular, melting from the east AIS, which is closer to New Zealand, would shift the eustatic band crossing the North and South Islands northward, above the North Island and cause a lower-than-eustatic local sea-level rise. Various scenarios of Antarctic ice geometries indicate New Zealand RSL approximates eustatic sea level with GIA having little effect (~2-3 m deviations from eustatic) on New Zealand (Grant et al., 2019), thus making it useful for constraining global ice-volumes during MIS 5e. The predicted deviations of RSL from the eustatic during MIS 5e are partly due to ocean syphoning and continental levering. The former causes local (New Zealand) sea-level drop in response to water flow towards the subsiding peripheral forebulges that surround the glaciated areas. The latter causes relative sea-level drop in response to local crustal uplift as a consequence of water-loading-induced crustal tilt. Hence, both processes result in a New Zealand highstand 1-3 kyr earlier that eustatic, which is then followed by a RSL drop (Figure 5). The combined effect of ocean syphoning and continental levering may explain the variability of Holocene sea-level change around New Zealand. For example, the Holocene highstand peaked in the North Island at ~2.65 m apsl between 8.1 to 7.2 cal ka BP, whereas in the South Island, the highstand peaked later, between 7.0 and 6.4 cal ka BP, at no more than ~2 m apsl (Clement et al., 2016).

The significance of New Zealand glacier ice-volume change on coastal deformation and sea-level reconstructions has not been quantified for New Zealand (King et al., 2020). Resolving any flexure within New Zealand beyond the Holocene, due to regional glacier ice-volume change or any other regional drivers, is unlikely due to the extensive coastal deformation. The gravitational effect of local glaciers would be hard to detect, and similarly with solid Earth deformations. Given the short wavelength of glaciers, their deformations would be most likely elastic and would therefore be compensated by space-limited upper lithosphere flexure/deformation.

The only last interglacial site identified so far that is most likely to have been unaffected by deformation is One Tree Point in Northland (Section 4.1.1), highlighting the importance of this region for additional study.

[Figure]

Figure 5: Local RSL curves for MIS 5e sea level at the northernmost tip of North Island (Te Hapua; latitude -34.39, longitude 173.02), Auckland (latitude -36.85, longitude 174.76), and the southernmost tip of South Island (Coal Island; latitude -46.21, longitude 166.66) generated from ANICE-SELEN and ICE-6G models. Similar to the Holocene, sea level peaks earliest and higher in the North Island. The Northland region (Te Hapua) RSL curve is near to eustatic. Deviations of the RSL curves from eustatic is driven by ocean syphoning, suggesting it serves as a primary driver of variability in the timing and height of peak sea level across New Zealand. Note the different scale to x- and y- axes between model outputs.

**Comment 10**: *Line – 1134 - It wouldn't hurt the authors to cite Naish et al 1998 (QSR) or Pillans et al (2005) here, if they are going refer to these older sequences.*

We thank Prof. Naish for bringing these publications to our attention. They are added as references.

**Comment 11**: *Line – 1150 – I would suggest the authors also read Grant et al 2019 (Nature), and some of the other Whanganui-based Pleistocene literature. This statement is misleading and shows a lack of understanding as to how the older sea-level reconstructions are established using the shallow-marine record. They are not absolute sea-level indicators like wave cut platforms and strandlines. They cannot be compared to MIS 5 sea-level indicators around NZ. They do however fundamentally record many transgressions and regressions of the shoreline in response to orbitally-paced global sea-level and associated water depth changes. But reconstructed water depth changes are also affected by local tectonic subsidence, compaction, and loading that all have to be accounted for when deriving the amplitude glacial-interglacial global sea-level change. The sea-level cycles reconstructed from the Whanganui sequences cannot be registered directly to present day sea-level because of late Quaternary uplift of western North island, which is the reason the marine terraces are preserved post MIS 15.*

We regret that our writing here was not clear, and we have rewritten material accordingly. We did not say, nor intend for it to be interpreted, that the Whanganui Basin sequence provides absolute sea-level indicators. Nor were we trying to compare them to other MIS 5 sea-level indicators around NZ. Rather, our reference to the record of sea-level fluctuations is to the transgressive and regressive sequences preserved within the basin. With this in mind, the first paragraph of this section (Section 6) was rewritten to better reflect the character of the Whanganui Basin sequence.

The Whanganui Basin, in addition to the Rapanui (MIS 5e), Inaha (MIS 5c), and Hauriri (MIS 5a) terraces (Section 4.1.2), retains a sequence of shallow marine transgressive, highstand, and regressive sediments correlated with each high sea-level marine oxygen isotope stage of the past 2.6 Ma, reflecting cyclic, orbitally-paced eustatic sea-level fluctuation (Pillans, 1991; Beu et al., 2004; Pillans et al., 2005; 2017; Naish et al., 1998; Grant et al., 2019). This record is extremely well-preserved and has been subject to an extensive variety of geochronological and stratigraphical methods. Not only does it offer a detailed paleoenvironmental record from an isolated part of the South Pacific, it serves as a paleo-proxy for the amplitude of interglacial-glacial relative sea-level change and constrains polar ice-volume variability within the Pliocene (3.30 to 2.50 Ma) when atmospheric carbon dioxide concentration was last ~400 parts per million – a climatic condition recently met. It retains a long record of tephra (to c. 2.17 Ma) and loess (to c. 0.50 Ma) deposition, which provides a framework for regional stratigraphic correlation in the North Island. The stratotype sections and points of the four stages representing Quaternary New Zealand are defined by the

fossiliferous marine sediments within the Whanganui Basin. A paleovegetation and paleoclimatic record spanning much of the Haweran Stage (0.340 Ma to present) has been developed from the marine and terrestrial sequence. Unfortunately, the ongoing and complex tectonics of the North Island preclude any sea-level reconstruction registered to present day – indeed it is the relatively recent uplift of the basin margins which has allowed preservation of marine terraces formed over the past 0.7 Ma, including the Rapanui, Inaha, and Hauriri terraces.

---

## Referee Comment (RC2) · Nicola Litchfield (Referee) · 5 Feb 2021

This paper presents a very useful new dataset of last interglacial relative sea level indicators for New Zealand (NZ) Aotearoa. The dataset appears to have been rigorously compiled to the WALIS global standard and it is useful, albeit humbling (in terms of the relatively few datapoints able to be compiled), to see these data in a global context. The paper is very well written and appropriately illustrated, bar the minor points I discuss below and on the annotated pdf. I have not checked the calculations, in part because I don't fully understand the methods (discussed below), but I suspect there are no major errors as the calculations are not complex.

The largest issue I have with the paper in its current form is the lack of detail on the

methods (section 2 is notably brief). I appreciate that this paper is part of a special issue and the methods and parameter descriptions are described elsewhere, but nevertheless I do think there should be some be some brief additions to this paper to make it more stand-alone. This will be particularly useful to NZ readers. I have noted these on the annotated pdf, but the key methods are: 1) What criteria did you use to include datapoints? Can you provide more info (e.g., a list) on the publications reviewed but not included? 2) Explain the RSL indicators (marine-limiting, direct sea level, terrestrial-limiting). It would be particularly helpful to see these labelled on Figure 1, similar to Figure 2 of Rovere et al. (2016). 3) Explain the indicative meaning calculation. In particular the calculations in the IMCalc tool, and the calculation of uncertainties. For each of these I'm thinking of one or a two sentences of relatively high explanation and then referring to the other papers and links for more details. It would also be helpful (again, especially for a NZ audience) to work through an example for 2 and 3.

To expand upon the criteria method a bit further, having done similar compilations in the past (e.g., Beavan and Litchfield, 2012), I am aware of a number of other data sources such as geological maps, student theses and groundwater reports that have some LIG SL information in them. I suspect many of them were examined and were unable to be used because they didn't contain specific locations, elevations, or ages, or perhaps there was a preference for peer-reviewed papers or publicly available reports? For example I was also going to raise the Ninis (2018) thesis raised by reviewer Tim Naish, and I see from the response that the authors were aware of it but chose not to include it so as not to scoop the publication (thanks for that and yes, it is in progress and can be provided later). Some that I thought would have provided useful information are the Ota et al. (1981) Late Quaternary tectonic map of NZ, the Dravid and Brown (1987) Heretaunga Plains Groundwater report (references are contained in Beavan and Litchfield 2012) and the paper by Schermer et al. (2009) Tectonics, 28: TC6008; doi: 10.1029/2008TC002426.

Another surprising omission is any discussion of the relative sea levels that have been

compiled. While I appreciate this is primarily a data paper, I was surprised to not see any discussion of the graphs in the lower parts of Figures 3 and 4. While I am biased in that I am a tectonic geomorphologist, and the patterns that are obtained are almost entirely likely the result of tectonics, I still find it odd not to discuss the results. The total range of RSLs are also mentioned in the abstract, but not elsewhere. Again, I'm only suggesting a few sentences should be added, probably to section 4.3.

In several places there are references to the Otago coastline previously being considered to be tectonically stable. While this was certainly a key assumption for construction of the NZ Holocene sea level curve (Gibb, 1986) I think the deformation of the coast south of Dunedin has been long known. Personally (and I acknowledge I'm biased and sensitive to this since it was my PhD area) I've always considered the stable part of the South Island to be farther south, in southeast Southland (including Rakiura/Stewart Island). So I'd recommend changing the references to Otago being stable to be "parts of Otago/SE Southland", or if this is too much of a mouthful, just say "the SE South Island" rather than singling out Otago. You may even like to reiterate the concerns about the impact on the Holocene NZ sea level curve from the deformation of the LIG terrace at Blueskin Bay, as I don't think this point has really been made strongly enough previously.

I applaud the use and updating of the Pillans (1990) figure in Figure 1, but I do have an issue with the depiction of the modern beach and platform. There is a significant body of research on processes of formation of rock shore platforms in NZ (none of which I note is referenced) and my understanding of it and my own personal observations is that where there is a well developed shore platform, it generally forms between high tide and low tide. I appreciate that discussion of much of this is beyond the scope of this paper, but I would like to see this part of Figure 1 updated to reflect this and at least some reference to the shore platform literature. A good starting place would be review paper by Dickson and Stephenson (2014) – The rock coast of NZ, Chapter 13, in: Kennedy, D. M., Stephenson, W. J. & Naylor, L. A. (eds) 2014. Rock Coast
Geomorphology: A Global Synthesis. Geological Society, London, Memoirs, 40, 225–234. http://dx.doi.org/10.1144/M40.13.

There are quite a few references missing from the reference list, which I've listed below. Some may be typos that I wasn't confident enough to correct on the pdf, but in particular most of the references in Table 2 are missing. Bowen et al. (1988) Bowen et al. (1998) Brown (1988) Bull (1985) Chappell et al. (1996) Cowie (1961) Cowie (1981) Goodfriend et al. (1995) Hammon et al. (1983) Lambeck and Chappell (2001) Lowe (2019) Ludwig et al. (1996) McGlone et al. (2004) Muhs (2000) Palmer (1988) Pillans (1985) Pillans (1986) Pillans (1991) Pillans et al. (1994) Pillans et al. (1998) Siddall et al. (2007) Veeh and Chappell (1970) Vucetich and Pullar (1982) Ward (1967) Worth and Grant-Mackie (2003)

Please also note the supplement to this comment:
https://essd.copernicus.org/preprints/essd-2020-288/essd-2020-288-RC2-supplement.pdf

---

## Author Comment (AC3) · 8 Mar 2021

**Response to Comments by Dr. Nicola Litchfield (Referee) to "The last interglacial sea-level record of New Zealand (Aotearoa)"**

Deirdre D. Ryan[1*], Alastair J.H. Clement[2], Nathan R. Jankowski[3,4], Paolo Stocchi[5]

[1]MARUM – Center for Marine Environmental Sciences, University of Bremen, Bremen, Germany
[2]School of Agriculture and Environment, Massey University, Palmerston North, New Zealand
[3] Centre for Archeological Science, School of Earth, Atmospheric and Life Sciences, University of Wollongong, Wollongong, Australia
[4]Australian Research Council (ARC) Centre of Excellence for Australian Biodiversity and Heritage, University of Wollongong, Wollongong, Australia
[5]NIOZ, Royal Netherlands Institute for Sea Research, Coastal Systems Department, and Utrecht University, PO Box 59 1790 AB Den Burg (Texel), The Netherlands

**8 March 2021**

**1. Summary**

We thank Dr. Litchfield for her review of our manuscript. Within this response we address each comment individually and detail any changes made to the original manuscript. Comments by Dr. Litchfield are provided in italic, with our response in standard font, and changes to manuscript are in blue.

**2. Overview Comment**

*This paper presents a very useful new dataset of last interglacial relative sea level indicators for New Zealand (NZ) Aotearoa. The dataset appears to have been rigorously compiled to the WALIS global standard and it is useful, albeit humbling (in terms of the relatively few datapoints able to be compiled), to see these data in a global context. The paper is very well written and appropriately illustrated, bar the minor points I discuss below and on the annotated pdf. I have not checked the calculations, in part because I don't fully understand the methods (discussed below), but I suspect there are no major errors as the calculations are not complex.*

We are pleased that Dr. Litchfield found our compilation useful and we thank her for her constructive comments, which will serve to improve this work. Responses to specific comments are provided below.

**3. Specific Comments**

**Comment 1.** *The largest issue I have with the paper in its current form is the lack of detail on the methods (section 2 is notably brief). I appreciate that this paper is part of a special issue and the methods and parameter descriptions are described elsewhere, but nevertheless I do think there should be some be some brief additions to this paper to make it more stand-alone. This will be particularly useful to NZ readers. I have noted these on the annotated pdf, but the key methods are: 1) What criteria did you use to include datapoints? Can you provide more info (e.g., a list) on the publications reviewed but not included? 2) Explain the RSL indicators (marine-limiting, direct sea level, terrestrial limiting). It would be particularly helpful to see these labelled on Figure 1, similar to Figure 2 of Rovere et al. (2016). 3) Explain the indicative meaning calculation. In particular the calculations in the IMCalc tool, and the calculation of uncertainties. For each of these I'm thinking of one or a two sentences of relatively high explanation and then referring to the other papers and links for more details. It would also be helpful (again, especially for a NZ audience) to work through an example for 2 and 3.*

In response to the numbered criteria in the comment:

1) A sentence was added to clarify the criteria required for a data point to be considered an RSL indicator:

   In order to be considered for entry into WALIS as an RSL indicator, a data point must have three characteristics (Rovere et al., 2016): 1) elevation referred to a defined sea-level datum, and position (latitude and longitude) referred to a known geographic system; 2) its offset (relative or absolute) from a former sea-level needs to be known; and 3) it must have an established age (relative or absolute).

2 and 3) An example of a marine-limiting and direct RSL indicator is already provided in Section 2. However, we expanded the explanation to provide further clarity. Greater description of indicative meaning and the IMCalc tool was also added. This includes a new Table 1 which details the indicative meaning of the RSL indicators identified in this review. Figure 1 and its caption were altered to provide better illustration and description of indicative meaning and the calculation of paleo sea level. Changes to text are as follows:

[revised manuscript text omitted]

**Comment 2.** *To expand upon the criteria method a bit further, having done similar compilations in the past (e.g., Beavan and Litchfield, 2012), I am aware of a number of other data sources such as geological maps, student theses and groundwater reports that have some LIG SL information in them. I suspect many of them were examined and were unable to be used because they didn't contain specific locations, elevations, or ages, or perhaps there was a preference for peer-reviewed papers or publicly available reports? For example I was also going to raise the Ninis (2018) thesis raised by reviewer Tim Naish, and I see from the response that the authors were aware of it but chose not to include it so as not to scoop the publication (thanks for that and yes, it is in progress and can be provided later). Some that I thought would have provided useful information are the Ota et al. (1981) Late Quaternary tectonic map of NZ, the Dravid and Brown (1987) Heretaunga Plains Groundwater report (references are contained in Beavan and Litchfield 2012) and the paper by Schermer et al. (2009) Tectonics, 28: TC6008; doi: 10.1029/2008TC002426.*

To clarify what publications were examined for potential inclusion into WALIS, the following sentences were added to Section 1:

> For our review, preference was given to peer-reviewed publications; although student theses heavily referenced in later work were included (e.g. Goldie 1975; Hicks, 1975) if made available. Geological maps, while useful in indicating the extent of marine deposits and general location of a paleo-shoreline, seldom provide precise descriptions, geographical locations, or elevations above sea level of a singular data point; e.g. Nathan, 1975; Begg and Johnston, 2000. However, a publication not having sufficient information for the identification of an RSL indicator, did not preclude its usefulness or inclusion in this review. Many publications contained research worthy of discussion in Section 4. Other publications, some geological maps and government reports were behind paywalls or were not found in the course of this review.

In response to the specifically listed references by Dr. Litchfield: Ota et al. (1981) we were aware of but it is behind a paywall. We could only locate an executive summary of the Dravid and Brown (1987) report, which did not provide sufficient information to identify RSL indicators. Schermer et al. (2009) we were not aware of. It is worth noting, that at some point the publication chase must stop and the writing must begin. While we tried to be as comprehensive as possible, it is unrealistic to expect availability or awareness of every single publication that may contain data points qualifying for entry into WALIS or even for consideration in the paleosea-level discussion. One of the great qualities of WALIS is that it is a 'living' database – meaning that anyone can add additional data points and future versions of reviews can build off of this first special issue.

Dr. Litchfield, in her pdf comments (and above in **Comment 1**), also requests a list of publications reviewed. Most publications are included in this review and are therefore, listed in the references. Some publications that were reviewed but found not meeting the standards necessary for inclusion in WALIS were not downloaded and we did not keep a record keep a record of this. While we appreciate that it would be somewhat helpful to provide a list of these publications, identifying them would require re-reviewing the literature, and for completeness, including reasoning for why they were rejected. This would add a considerable amount of work at this stage and also falls outside the scope of WALIS. As stated in the in sentences added to Section 1, most publications reviewed, even if not providing a RSL indicator, were included if they contributed to the discussion within Section 4.

**Comment 3.** *Another surprising omission is any discussion of the relative sea levels that have been compiled. While I appreciate this is primarily a data paper, I was surprised to not see any discussion of the graphs in the lower parts of Figures 3 and 4. While I am biased in that I am a tectonic geomorphologist, and the patterns that are obtained are almost entirely likely the result of tectonics, I still find it odd not to discuss the results. The total range of RSLs are also mentioned in the abstract, but not elsewhere. Again, I'm only suggesting a few sentences should be added, probably to section 4.3.*

The role of coastal deformation in resolving a useful RSL record is discussed in Section 5.2, which has been significantly altered from the version Dr. Litchfield has read in response to Reviewer 1 comments. (See also response below to **Pdf Comments Section 5.2**). However, Dr. Litchfield is right in that we should mention the role of tectonics in this summary. The following was added:

> The varying elevation of the RSL indicators (Figure 3E and 4I) illustrates the role of tectonics in shaping the New Zealand coastline. For example, the transition from the subsiding landscape of the Canterbury Plains to an uplifting one in the north is distinctly marked by the elevation of RSL indicators. The substantial imprint of tectonics makes difficult the development of a sea-level record that can be resolved to modern sea level (Section 5.2) and underlines the historical tendency to use these records for determining VLM rates but not a record of sea level.

**Comment 4.** *In several places there are references to the Otago coastline previously being considered to be tectonically stable. While this was certainly a key assumption for construction of the NZ Holocene sea level curve (Gibb, 1986) I think the deformation of the coast south of Dunedin has been long known. Personally (and I acknowledge I'm biased and sensitive to this since it was my PhD area) I've always considered the stable part of the South Island to be farther south, in southeast Southland (including Rakiura/Stewart Island). So I'd recommend changing the references to Otago being stable to be "parts of Otago/SE Southland", or if this is too much of a mouthful, just say "the SE South Island" rather than singling out Otago. You may even like to reiterate the concerns about the impact on the Holocene NZ sea level curve from the deformation of the LIG terrace at Blueskin Bay, as I don't think this point has really been made strongly enough previously.*

It is difficult to include the southeast as stable when there are no publications providing record of a sea-level indicator in support. We noted that Beavan and Litchfield (2012) referred to portions of this coast as stable citing Turnbull and Allibone (2003). This is a geologic map and only provides very generalized information regarding a terrace 'inferred to date from OI stage 5 (70 – 130 ka)' (p. 48). Furthermore, the terrace is described as increasing in height from 5-7 m above sea level near the Mataura River to 25-60 m above sea level west of the Waiau River – hardly indicating stability. However, in appreciation for Dr. Litchfield's familiarity with the section of the coastline, we have added the following to the end of Section 4.2.1 West Coast and Southland:

> Beavan and Litchfield (2012) report the coastline of eastern Southland, into the Otago Region, includes regions of stability, referencing Turnbull and Allibone (2003). However, this is a reference to a geological map without any precise locations, elevations, or geochronological constraint. No other publications detailing potential RSL indicators in the region were identified in our review. Nevertheless, this indicates eastern Southland is a potential source of valuable RSL indicators little affected by tectonic deformation and worthy of further investigation.

Additionally, reference to eastern Southland stability was added throughout the manuscript.

We also agree with Dr. Litchfield that the deformation of the LIG terrace at Blueskin Bay has serious implications for the Holocene NZ sea level curve but have opted to not include a comment within the manuscript. However, we added emphasis that the fault movement provides evidence of instability within Otago.

**Comment 5.** *I applaud the use and updating of the Pillans (1990) figure in Figure 1, but I do have an issue with the depiction of the modern beach and platform. There is a significant body of research on processes of formation of rock shore platforms in NZ (none of which I note is referenced) and my understanding of it and my own personal observations is that where there is a well developed shore platform, it generally forms between high tide and low tide. I appreciate that discussion of much of this is beyond the scope of this paper, but I would like to see this part of Figure 1 updated to reflect this and at least some reference to the shore platform literature. A good starting place would be review paper by Dickson and Stephenson (2014) – The rock coast of NZ, Chapter 13, in: Kennedy, D. M., Stephenson, W. J. & Naylor, L. A. (eds) 2014. Rock Coast Geomorphology: A Global Synthesis. Geological Society, London, Memoirs, 40, 225–234. http://dx.doi.org/10.1144/M40.13.*

It is important here to clarify the landform features being discussed: a rocky shore platform with no or minimal overlying sediments versus the sub-horizontal platform upon which marine terrace sediments have been deposited. These landforms typically differ in scale – the former are often only a few tens of meters wide (outer to inner margin), whereas marine terraces can be hundreds to thousands of meters in width. Furthermore, as Dr. Litchfield notes, a rocky shore platform forms at a different position relative to mean sea level than a marine terrace and therefore, also differs in indicative range. We would argue the more relevant issue to be recognized from Dr. Litchfield's comment is with regards to terminology and the importance of adequately describing landform features. As noted in the manuscript, Pillans (1990a; b) recognized significant confusion in nomenclature of marine terrace features inspiring him to establish terms and definitions that continue to be used, including 'wave-cut shore platform' used within our manuscript. However, with respect for the current discussion surrounding the formation of shore platforms, we have changed instances of 'wave-cut shore platform' to 'basal platform' when discussing the sub-horizontal feature beneath marine terrace sediments unless specifically referred to as wave-cut shore platform by original author. We have added text to Section 2 to clarify our distinction between a marine terrace (with a basal shore platform) and a rocky shore platform. We have also added the differing indicative meanings with the inclusion of Table 2 (See response to **Comment 1**).

> Marine terraces have a similar morphology to rocky shore platforms, resulting in shared terminology. However, shore platforms are distinguished from marine terraces by their exposed bedrock surface and relative lack of overlying sediment reflecting active erosional and weathering processes (Griggs and Trenhaile, 1994). The relative roles of marine and subaerial processes in shaping rocky shore platforms remains equivocal (e.g. Stephenson, 2000; Stephenson and Kirk, 2000; Trenhaile, 2008), and shore platforms do not form at uniform elevations because of sea-level alone (e.g. Kennedy and Dickson, 2007; Kennedy et al., 2011; Stephenson and Naylor, 2011). A number of publications from the West Coast and Southland Regions (e.g. Wellman and Wilson, 1964; Bishop, 1985; Bull and Cooper, 1986; Section 4.2.1) allude to landforms which may possibly be features of rocky shore platforms; however, sufficient morphological description to identify a RSL indicator from a rocky shore platform was only provided by Kim and Sutherland (2004; Section 4.2.1). To avoid confusion within Section 4 between rocky shore platforms and the basal shore platforms underlying marine terraces, the latter is referred to as basal platform within Section 4 where it is apparent that the platform is overlain by marine sediments.

**Comment 6.** *There are quite a few references missing from the reference list, which I've listed below. Some may be typos that I wasn't confident enough to correct on the pdf, but in particular most of the references in Table 2 are missing. Bowen et al. (1988) Bowen et al. (1998) Brown (1988) Bull (1985) Chappell et al. (1996) Cowie (1961) Cowie (1981) Goodfriend et al. (1995) Hammon et al. (1983) Lambeck and Chappell (2001) Lowe (2019) Ludwig et al. (1996) McGlone et al. (2004) Muhs (2000) Palmer (1988) Pillans (1985) Pillans (1986) Pillans (1991) Pillans et al. (1994) Pillans et al. (1998) Siddall et al. (2007) Veeh and Chappell (1970) Vucetich and Pullar (1982) Ward (1967) Worth and Grant-Mackie (2003).*

This is a rather embarrassing lack of oversight and we are extremely grateful to Dr. Litchfield for having pointed out these omissions. We have corrected typos, added the missing references, and completed a more thorough check of the manuscript to make sure no other references were omitted from the reference list.

Typos: Bowen et al., 1988; Brown, 1988; McGlone et al., 2004; Palmer et al., 1988; Pillans, 1983

Added to references: Bowen et al., 1998; Bull, 1985; Chappell et al., 1996; Cowie, 1961; Harmon et al., 1983; Lambeck and Chappell, 2001; Lowe, 2019; Ludwig et al., 1996; Muhs, 2000; Pillans, 1991; Pillans et al., 1994; 1998; Siddall et al., 2007; Stirling et al., 1996; Veeh and Chappell, 1970; Vucetich and Pullar, 1969; Ward, 1967; Worthy and Grant-Mackie, 2003

**Comment 7.** *Please also note the supplement to this comment: https://essd.copernicus.org/preprints/essd-2020-288/essd-2020-288-RC2-supplement.pdf*

There are number of minor comments throughout the manuscript pdf (the supplement), such as suggestions for minor changes to wording and figures. These suggestions were followed throughout. However, there were some more substantive comments, which are copied here and for which a more detailed response is provided.

**Pdf Comment Section 1 Introduction**: *Can you expand this* [WALIS framework] *a bit more? Perhaps also include an example? As per main comments, I'd like to see this introduced a bit more so this is a stand-alone paper. What is the purpose and scope of WALIS?*

It is difficult to expand on what is already written. As stated, the WALIS database provides a framework for the cataloguing of published last interglacial RSL indicators: their descriptions, geochronological constraint, and associated metadata. To assist in clarifying the data entry character of the database the following sentences was added:

> The intuitive interface of WALIS provides a standardized template for data entry, clarifying the collection and analytical methods used in identifying and describing previously published and new paleosea-level proxies and the source of any associated uncertainties for the scientific community. This includes fields for type and description of sea-level indicator, elevation measurement method and uncertainty, geographic positioning method and uncertainty, sea-level datum, and geochronological constraint, methods, age and uncertainty; e.g. amino acid racemization and luminescence.

As for providing an example – the link in the third sentence is to the database created from this work. The link in the fourth sentence is to the database field descriptors within the database. Other than a worked example, which would be excessively long, nothing further can be added.

**Pdf Comment Figure 1**: *I don't quite agree with the way this is drawn. Most modern shore platforms in NZ form between low and high tide, so mean SL and the breaking depth should be lower*.

Figure 1 was drawn showing only the upper and lower indicative range of marine terraces, which differs from rocky shore platforms. The response to **Comment 5** addresses the distinction between marine terrace and rocky shore platform and the response to **Comment 1** provides more detail regarding indicative range. We chose not to include the indicative range of a rocky shore platform to Figure 1 because 1) they are not a common RSL indicator identified within the database, 2) it would make the Figure pretty messy, and 3) the indicative range of a rocky shore platform is included in the new Table 2 (**Comment 1**).

**Pdf Comment Figure 2**: *You can't see the Regional boundaries onshore (grey on grey) – change the colour*.

While we appreciate that you cannot see the Regional boundaries onshore, this is not of importance in this figure as the sole purpose is to show the subdivision of the coastline by Region. Furthermore, visible boundaries onshore would be complicated by the division of the tectonic regimes and numerous faults resulting in a confusing and messy figure.

**Pdf Comment Figures 3 and 4**: *Why don't you show the uncertainties? By showing all the datapoints with the same size symbols is a bit misleading*.

Due to the scale of the y-axis in most instances the uncertainty of the RSL indicator is smaller than the symbol and not visible. Scaling the symbols to the y-axis would make the majority practically invisible. For marine and terrestrial limiting points, where uncertainty is visible, overlap with the indicator arrow adds confusion as to the actual uncertainty. The numerical elevation and uncertainty are available for every indicator in the text as well as in the database. The primary message of Figures 3 and 4 is the location of indicators around the North and South Islands and to illustrate the variable elevation of RSL indicators as a reflection of displacement.

**Pdf Comment Section 4.2 South Island**. *I'm surprised not to see any mention of the marine terraces in the NW Nelson area (e.g., Farewell-Collingwood and Kahurangi areas) and the subsidence of the Marlborough Sounds. I appreciate the data is quite poor, but it might be worth mentioning this.*

We did identify some potential references for RSL indicators in the Nelson and Tasman regions – mostly through use of the references list within Beavan and Litchfield, 2012. Briefly, those references are:

- Johnston (1979) Geology of the Nelson urban area.
- Bishop (1968) Geological map of New Zealand 1:63,360 Sheet S2 Kahurangi
- Bishop (1971) Geological map of New Zealand 1:63,360 Sheet S1, S3 and pt S4 Farewell-Collingwood
- Rattenbury et al. (1998) Geology of the Nelson area
- Williams (1982) Speleothem dates, Quaternary terraces and uplift rates in New Zealand
- Berryman and Hull (2003) "Tectonic controls on Late Quaternary shorelines: a review and prospects for future research"

Bishop (1971) we were unable to get access to. Bishop (1968) only alludes to the presence of a marine deposit. The other map sheets (Johnston, 1979; Rattenbury et al., 1998) have the before state issues of no precise locations, elevations, or description adequate for the identification of a RSL indicator. However, an additional statement was added to the top of Section 4.2 to acknowledge the presence of marine terraces in the region:

No RSL indicators have been correlated to MIS 5 in the Nelson or Tasman Region; although geological mapping in the region indicates the presence of marine terraces in the area of Nelson City (Johnston, 1979; Rattenbury et al., 1998).

Williams (1982) is also briefly discussed at the top of Section 4.2. The publication was rejected for a couple of reasons. The only location information is cave names, many of which appear to be no longer in use as they could not be found on any current maps from which we could derive lat/long coordinates. Furthermore, the speleothems provide minimum ages for cave formation and there is no description or substantial discussion of LIG sea-level indicators.

Berryman and Hull (2003) is a book that we did not review.

**Pdf Comment Section 4.3 Summary of New Zealand RSL indicators**. *I think you could and should add some brief discussion in here on the RSL indicator elevation patterns (Figs. 3e and 4i) and relationship to tectonics.*

See response to **Comment 3**.

**Pdf Comments Section 5.2 Coastal deformation and GIA**. *This is a really useful section, but it would be even more useful if you could make some stronger recommendations about what should, and could, be done.*

We are not entirely sure what is being asked for here, whether it be recommendation for deriving a RSL curve or how to account for coastal deformation and GIA. (Before responding further, we should note that this section has been significantly altered in response to review comments by Prof. Tim Naish and that revision may better meet Dr. Litchfield's request.) If the request is for better recommendations to derive a RSL curve, as noted in the text,

Excessive and prevalent coastal deformation will preclude the development a sea-level reconstruction that can be registered to present day, regardless of the quality of sea-level indicators.

In other words, an RSL curve registered to present day sea level is not possible where there has been excessive deformation (see also response to **Comment 3**). This is why the Northland and SE South Island are important regions for future research to identify additional MIS 5e sea-level indicators, which will also help to constrain GIA models. However, our new revised text also notes that,

Although any sea-level reconstruction derived from these indicators may not be useful for a relative sea-level curve with relation to present sea level, more precise descriptions and age constraint can improve estimates of VLM rates, especially where there is uncertainty in correlation to the appropriate MIS 5 highstand. Furthermore, where high rates of uplift have produced marine terrace sequences recording multiple substages of MIS 5, there is opportunity to better constrain not only regional sea-level fluctuations within MIS 5 but paleoenvironmental change as well.

*Please reword as I don't think this is true for much of NZ and isn't quite what Beavan and Litchfield (2012) said. What we actually said was that in areas of large earthquakes then the long-term VLM can be obtained from the geologic data. We didn't actually talk about long-term VLM in areas of earthquakes because you have to untangle the interseismic and coseismic deformation, which in some places (e.g., the Hikurangi Margin) can be in the opposite sense. I don't think it's that important for your paper, but please remove the inference that the short-term interseismic VLM reflects long-term VLM.*

The sentence Dr. Litchfield is requesting to be reworded is,

"Although displacement by an earthquake can have dramatic effect, interseismic deformation (deformation between earthquake events) is more likely to influence long-term trends in VLM (Beavan and Litchfield, 2012)."

We certainly thank Dr. Litchfield for her comment and explanation as none of us are tectonic geomorphologists and we obviously misinterpreted the original publication. We have deleted the sentence.

**Pdf Comment Section 5.3.4 Luminescence**. *Can you be more specific here about what is missing? Is it missing in all papers? (e.g., do the Litchfield and Lian, 2004 and Oakley et al., 2017 papers have the required info? Can [we] be more specific? Is this the full list, or just some of the required data? Is there a paper that you could reference as an example of what is needed? Does including the OSL dating report as an electronic supplement cover it?*

We thank Dr. Litchfield for this comment and welcome the opportunity to further clarify what information is required. In any scientific experiment, the methods section ought to set out what has been done to your samples so that the reader, in theory, should be able to replicate the results. For the literature that was reviewed as part of this paper, these methods sections were, for the most part, lacking in that detail – it should be noted that the two papers in this comment cited (Litchfield and Lian, 2004 and Oakley et al., 2017) were among the most complete with the required information. We have provided a detailed response that is divisible into three sections: 1) equivalent dose evaluation, procedures, and equipment; 2) dose rate determination; and, 3) presentation of the results. Our response is quite long and will be added to the publication as Supplementary Material, "Recommendations for Reporting Luminescence Method". It is referred to in the manuscript with the addition of the following sentence:

The WALIS database interface allows for the reporting and archiving of all critical luminescence data and a full recommendation list is provided in the Supplementary Materials.

Our detailed response, to be included in Supplementary Materials, is as follows:

**Recommendations for Reporting Luminescence Method**

1. **Equivalent dose**

Basic information required includes: mineral type; pre-treatment techniques including whether or not the samples were etched in hydrofluoric acid (HF) and, if so, at what concentration and for how long; machine and photomultiplier tube type; detection filters used; single grain or single aliquots used; if aliquots were used, the diameter size; De determination method used (e.g., single aliquot regenerative dose, single aliquot additive dose, multiple aliquot additive dose, etc.); and an outline of the rejection criteria used to screen the data.

It is well documented that luminescence dating of New Zealand sediments and determination of the equivalent dose (De) is not straight forward for either quartz or feldspar. Focusing on feldspar dating, the De measurement procedure used is particularly important as it comes in several variants (e.g., IRSL50, pIRIR270, pIRIR290, etc.), each with their own strengths and weaknesses. This procedure needs to be spelt out in full either in the main text or in the supplementary information and include the magnitude and duration of preheating, light stimulation duration, and sample temperature. Hand in hand with feldspar dating is an assessment of the rate of luminescence signal lost, known as anomalous fading. The method of assessment should be noted and the results presented fully (%/decade ± standard error), not just a statement saying that it was 'not significant' or similar. There should also be an assessment of the magnitude of the residual dose remaining after a period of either solar or artificial bleaching. Again, the methods should be written in full with the magnitude presented ± standard error. A dose recovery test should also be performed using a subsample of grains that have been bleached and given a laboratory dose to determine the most appropriate preheating parameters used during measurement. For quartz, reporting should include all of the above, except for the fading and residual dose as these are not applicable. This information can be reported in the main text; however, it is more common to include in a supplementary information section.

2. **Dose rate determination**

Basic required information includes: method/technique used to assess external alpha, beta and gamma dose rates (where applicable); measurement results and/or assumption made about internal uranium and thorium content for quartz, or both of these and internal potassium for feldspars; method/calculations used for assessing the cosmic ray dose rate; water content evaluation and an estimate of the long-term water content used in age calculations; dose rate conversion factors used to convert concentrations to Gy/ka; and attenuation factors associated with external beta dose rate and any consideration of the impact that HF-etching might have on this value.

Ideally, an assessment of whether or not there is any disequilibrium in the uranium and thorium decay chains should also be conducted. Although this is not essential, some consideration of disequilibrium should be mentioned.

**3. Presentation of results**

A summary of the De, dose rate, and age calculation results for each dated sample should be presented in a table in the main text. This table should include: Sample name; external dose rate values and their associated uncertainties (alpha (where applicable), beta, gamma, and cosmic); total dose rate; sample De; an estimate of overdispersion (the amount of spread within the data after all known and assumed sources of uncertainty have been considered); the age model or method used in combining individual De estimates in each sample; and the age estimate. Other optional columns would include: sample depth; 'as measured' water content; and the number of grains/aliquots used in final De determination. Ideally, a table should also be included in the supplementary information section showing the total number of grains/aliquots measured for each sample and where they failed to pass the established rejection criteria.

Most importantly, the spread within the De values for each sample needs to be displayed graphically. Although our preference is for the use of radial plots; abianco plots, and probability density function could also be used. It is on the basis of the distribution of De values in each sample that the sample's De is determined and used in the age calculation. In many cases, the patterns observed in De distributions directly relate to the syn- and post-depositional histories of the samples and, by extension, the appropriateness of the age models used to combine them. We would recommend that an example of the observed De distributions be included in the main text, with all distributions reproduced in the supplementary information.

o   Inclusion of dating reports

Although it is not recommended that you simply cut and paste a dating report into the supplementary information section, they are certainly a good place to start. Of the commercial firms that authors have used, most of these laboratory reports quickly off reference critical pieces of methodological information to sources that are either outdated or buried in hard-to-access journals or grey literature. However, any reputable dating laboratory should be able to provide you with the basic information outlined in our above response.